# Systematic Assessment of Tabular Data Synthesis

## Abstract

Data synthesis has been advocated as an important approach for utilizing data while protecting data privacy. In recent years, a plethora of tabular data synthesis algorithms (*i.e.,* synthesizers) have been proposed. A comprehensive understanding of these synthesizers' strengths and weaknesses remains elusive due to the absence of principled evaluation metrics and head-to-head comparisons between state-of-the-art deep generative approaches and statistical methods. In this paper, we examine and critique existing evaluation metrics, and introduce a set of new metrics in terms of fidelity, privacy, and utility to address their limitations. Based on the proposed evaluation metrics, we also devise a unified objective for tuning, which can consistently improve the quality of synthetic data for all methods. We conducted extensive evaluations of 8 different types of synthesizers on 12 real-world datasets and identified some interesting findings, which offer new directions for privacy-preserving data synthesis.

## 1 Introduction

Data-driven decision-making has emerged as the prevailing approach to advance science, industrial applications, and governance, creating the necessity to share and publish tabular data. At the same time, growing concerns about the privacy breaches caused by data disclosure call for data publishing approaches that preserve privacy. One increasingly advocated and adopted approach to reduce privacy risks while sharing data is to release synthetic data. Ideally, synthetic data can effectively fit any data processing workflow designed for the original data without privacy concerns. Data synthesis initiatives have been promoted not only by the research community (Tao et al., 2021) but also among non-profit organizations (OECD, 2023) and government agencies (Benedetto et al., 2018).

In this paper, we study data synthesis algorithms for tabular data, which we call **synthesizers**. In recent years, a plethora of synthesizers have been proposed, which can be roughly categorized into two groups: statistical and deep generative. Statistical synthesizers use low-order marginals to create synthetic datasets that match real data distributions. They were the best-performing algorithms in NIST competitions (NIST, 2018; 2020). Deep generative synthesizers, on the other hand, learn the data distribution from real data and generate synthetic instances by sampling from the learned distribution. With the recent development in deep generative models (*e.g.,* diffusion models (Ho et al., 2020) and large language models (LLMs) (Vaswani et al., 2017; Radford et al., 2019)), new synthesizers are proposed to extend these successes to the realm of tabular data synthesis.

While recent state-of-the-art approaches achieve compelling results in synthesizing authentic tabular data, a comprehensive understanding of the strengths and weaknesses of different synthesizers remains elusive. In addition, there is a lack of principled and widely accepted evaluation metrics for data synthesis. It is known that evaluating synthesizers is inherently difficult (Theis et al., 2016), and qualitative evaluation of tabular data through visual inspection is also infeasible.

The above concerns motivate us to design a systematic evaluation framework for data synthesis to elucidate the current advancements in this field. Specifically, we examine, characterize, and critique the commonly used evaluation metrics, and propose a set of new metrics for data synthesis evaluation. Our assessments unfold along three main axes:

- *Fidelity*. To address the heterogeneity and high dimensionality of tabular data, we present a new fidelity metric based on Wasserstein distance. This metric offers a unified way to evaluate numerical, discrete, and mixed data distributions under the same criteria.
- *Privacy*. We identify the inadequacy of existing syntactic privacy evaluation metrics and the ineffectiveness of membership inference attacks by conducting comparison studies. We also propose a new privacy evaluation metric to gauge the empirical privacy risks of synthesizers.
- *Utility*. We advocate two tasks for assessing the utility of synthesizers: machine learning prediction and range (point) query. To eliminate the inconsistent performance caused by the choice of different machine learning models, we present a utility metric that quantifies the distributional shift between real and synthetic data.

**SynMeter.** We implement a systematic evaluation framework called SynMeter to support the assessment of data synthesis algorithms with the proposed evaluation metrics. Differing from the existing evaluations, SynMeter incorporates the model tuning phase, which eases hyperparameter selection and consistently improves the performance of synthesizers for fair comparison. Our code is publicly available, facilitating researchers to tune, assess, or benchmark new synthesis algorithms.

## 2 DATA SYNTHESIS EVALUATION

Given a dataset $D$ sampled from an underlying distribution $\mathbb{D}$, $\mathsf{A} \leftarrow \mathcal{T}(D)$ denotes that the synthesizer $\mathsf{A}$ is learned by running the training algorithm $\mathcal{T}$ on $D$. The synthesizer $\mathsf{A}$ generates a synthetic dataset $S$ to replace $D$ for publishing. We consider three classes of properties for synthesizers:

- **Fidelity**. As the substitute for real data, the distribution of the synthetic dataset should be close to $\mathbb{D}$. Since $\mathbb{D}$ is often unknown, fidelity is measured by the similarity between the input dataset $D$ and the synthetic dataset $S$. If one partitions the input dataset $D$ into a training set $D_{\text{train}}$ and a test set $D_{\text{test}}$, one can measure fidelity as closeness to either $D_{\text{train}}$ or $D_{\text{test}}$.
- **Privacy**. Using synthetic data is usually motivated by the desire to protect the input dataset. Some training algorithms $\mathcal{T}$ are designed to satisfy Differential Privacy (DP) (Dwork, 2006), we refer to these as DP synthesizers. (See Appendix F for the formal definition). However, satisfying DP under reasonable parameters may result in poor performance. Some synthesizers do not satisfy DP, and aim to protect privacy empirically. We call these Heuristically Private (HP) synthesizers. As a result, privacy evaluation metrics are essential for evaluating the privacy of HP synthesizers.
- **Utility**. Synthetic data is often used to replace real datasets for downstream tasks. Thus, high fidelity may not necessarily be needed if it achieves good utility for these tasks. Hence, utility evaluation is useful to measure the effectiveness of synthesizers for common tasks.

## 3 EVALUATION METRICS FOR DATA SYNTHESIS ALGORITHMS

### 3.1 FIDELITY EVALUATION

**Existing Metrics and Limitations.** Existing fidelity metrics can be categorized into three groups: low-order statistics (McKenna et al., 2019), likelihood fitness (Xu et al., 2019), and evaluator-dependent metrics (Snoke et al., 2018). The main issue with low-order statistics is the lack of versatility. Each type of marginal distribution requires a specific statistical measure, complicating comprehensive comparisons across different attribute types. Likelihood fitness assesses how well synthetic data aligns with a known prior distribution. Although this is a natural approach for assessing fidelity, it becomes problematic when the prior distribution is unknown or complex, as is often the case in real-world datasets. Evaluator-dependent metrics, on the other hand, rely heavily on auxiliary evaluators (*e.g.,* thresholds or discriminators), which require careful calibration to ensure meaningful comparisons across diverse datasets and synthesizers. A more detailed discussion of existing fidelity metrics can be found in Appendix G.1.

**Proposed Metric: Wasserstein Distance.** We opt for Wasserstein distance to measure the distribution discrepancies between synthetic data and real data. Originating from optimal transport theory (Peyré & Cuturi, 2019), the Wasserstein distance provides a structure-aware measure of the minimal amount of work required to transform one distribution into another. Formally, Let $\mathbf{P} = (p_1, p_2, \ldots p_n)$ and $\mathbf{Q} = (q_1, q_2, \ldots q_n)$ be the two probability distributions, and $\mathbf{C}$ be a matrix of size $n \times n$ in which $\mathbf{C}_{ij} \geq 0$ is the cost of moving an element $i$ of $\mathbf{P}$ to the element $j$ of $\mathbf{Q}$

($\mathbf{C}_{ii} = 0$ for all element $i$). The optimal transport plan $\mathbf{A}$ is:

$$
\begin{aligned}
\min_{\mathbf{A}} \quad & \langle \mathbf{C}, \mathbf{A} \rangle \\
\text{s.t. } \mathbf{A}\mathbb{1} = \mathbf{P}, \quad & \mathbf{A}^\top \mathbb{1} = \mathbf{Q},
\end{aligned}
\tag{1}
$$

where $\langle \cdot, \cdot \rangle$ is inner product between two matrices, $\mathbb{1}$ denotes a vector of all ones. Let $\mathbf{A}^*$ be the solution to the above optimization problem, Wasserstein distance is defined as:

$$
\mathcal{W}(\mathbf{P}, \mathbf{Q}) = \langle \mathbf{C}, \mathbf{A}^* \rangle.
\tag{2}
$$

Now we can use Wasserstein distance to define the fidelity:

**Definition 1** (Wasserstein-based Fidelity Metric). *Let $v$ be a set of marginal variables, and $V = \{v\}$ is the collection of marginal variable sets. $f(v, D)$ is the marginal extraction function that derives the corresponding marginal distribution of $v$ from distribution $D$. Let $D$ and $S$ be the empirical distribution of real and synthetic data, respectively. The fidelity of synthesis algorithm A is:*

$$
\text{Fidelity(A)} \triangleq \frac{1}{|V|} \sum_{v \in V} \mathcal{W}(f(v, D), f(v, S)),
\tag{3}
$$

The smaller Wasserstein distance indicates the higher fidelity of the synthesizer A.

**Determining Cost Matrix.** The Wasserstein distance requires the predefined cost matrix $\mathbf{C}$, which encapsulates the "cost" of transitioning from one distribution element to another. For $k$-way marginal distributions $\mathbf{P}$ and $\mathbf{Q}$, the cost matrix is formulated by summing the pairwise distances between corresponding elements:

$$
\mathbf{C}_{ij} = \sum_{r=1}^{k} d(v_i^r, v_j^r).
\tag{4}
$$

Here, $v_i, v_j \in \mathbb{R}^k$ are the element located in $i$ and $j$ in $k$-way probability distributions. The distance $d(\cdot, \cdot)$ is tailored to the nature of the attributes, differing for numerical and categorical values:

$$
d(v_i^r, v_j^r) = \begin{cases} ||v_i^r - v_j^r||_1, & \text{if numerical} \\ \infty \text{ (if } v_i^r \neq v_j^r), 1 \text{ (if } v_i^r = v_j^r), & \text{if categorical} \end{cases}
\tag{5}
$$

**Wasserstein Distance for Categorical Attributes.** Wasserstein distance is typically defined for metric spaces and is well-suited for numerical attributes. However, the cost function for categorical attributes, as defined in Equation 5, represents an atypical usage of Wasserstein distance. We acknowledge this is a slight misuse of terminology to maintain consistency throughout the paper. We also note that the above definition for categorical attributes is equivalent to the computation of total variation distance (Kotelnikov et al., 2023) and contingency similarity (Patki et al., 2016), as used in previous studies. Additionally, it is also feasible to assign semantic distance for categorical attributes (Li et al., 2021), we omit it because it depends on the specific context and most synthesizers do not model the semantics in tabular data. Finally, while summing up distances for categorical and numerical attributes is a conventional approach in tabular data evaluation, we note that it may not be the optimal approach to capture similarities across heterogeneous data types.

**Merits of Wasserstein-based Fidelity Metric.** Wasserstein distance offers several advantages for evaluations: (i) Faithfulness. It is a natural and structure-aware statistic measure for analyzing distribution discrepancies and generalizing existing metrics like total variation distance. (ii) Universality. It accommodates both numerical and categorical attributes and extends to any multivariate marginals under the same criterion, facilitating the evaluation of heterogeneous types of marginals.

### 3.2 PRIVACY EVALUATION

**Existing Metrics and Limitations.** A popular approach to assess privacy risk for HP synthesizers is to compare the similarity between input dataset and synthetic data, with higher similarity suggesting greater information leakage. We call these metrics *syntactic* because they consider only the input and synthetic datasets, and not the algorithm used to generate the synthetic data. The most

popular syntactic metric is Distance to Closest Records (DCR) (Zhao et al., 2021), which looks at the distribution of the distances from each synthetic data point to its nearest real one and uses the 5th percentile of this distribution as the privacy score. DCR and other similar metrics are widely used in academia (Walia et al., 2020; Yale et al., 2019) and industry (AWS, 2022; Gretel, 2023), and have become the conventional evaluation metric for HP synthesizers (Ganev & De Cristofaro, 2023).

We point out that syntactic privacy evaluation notions that are independent of the underlying algorithm are fundamentally flawed. For example, a synthesis algorithm that applies the same fixed perturbation to every record could produce a synthetic dataset that is quite different from the input dataset, resulting in a good privacy score under a syntactic metric, even though the input dataset could be easily reconstructed from the synthetic dataset.

Membership inference attacks (MIAs) have been widely used for empirical privacy evaluation in machine learning (especially classification models) (Shokri et al., 2017). A few MIAs against tabular data synthesis algorithms have been proposed: Groundhog (Stadler et al., 2022), TAPAS (Houssiau et al., 2022) and MODIAS (van Breugel et al., 2023). Our comparison studies in Section 5.2 demonstrate that these MIA algorithms are limited in effectiveness: they fail to distinguish different levels of privacy leakage in some situations. We also observe that the standard metrics in MIA literature (*i.e.,* TPR@lowFPR) still do not capture the maximum leakage among all records in the input. The detailed analysis of the existing privacy evaluation metrics is in Appendix G.2.

**Proposed Metric: Membership Disclosure Score (MDS).** We propose a new privacy evaluation metric to assess the membership disclosure risks of synthesizers, which is inspired by both DCR and MIAs. The intuition behind MDS is that the inclusion or exclusion of each record $x \in D$ during training may lead to different behaviors of the synthesizer A, which can be measured as a function of $x$, $D$, and A. We use the maximum value for any $x$ as the measure of privacy leakage of applying A to $D$. Specifically, we first define the disclosure risk of one record as follows.

**Definition 2** (Disclosure Risk of One Record). *Let $O_D$ be the synthesizer A's output distribution when trained with dataset $D$, $\mathcal{M}$ is a distribution distance measurement, which is non-negativity and symmetric. The disclosure risk of record $x \in D$ is given by:*

$$\mathrm{DS}(x, \mathsf{A}, D) \triangleq \mathbb{E}_{H \subset D \setminus x}\big[\mathcal{M}(O_H \| O_{H \cup \{x\}})\big], \tag{6}$$

*where $H$ is the subset of training instances that are i.i.d sampled from $D \setminus x$. The expectation is taken with respect to the i.i.d sampling of $H$ and the randomness in the synthesis algorithm A.*

Our privacy definition compares the difference between two expected output distributions for a given record $x$. Unfortunately, the above computation is intractable: even the synthesizer's output distribution is not analytically known. To simplify the situation, we instead instantiate $\mathcal{M}$ to measure the closeness between $x$ and the empirical distribution of the synthetic data:

$$\widehat{\mathrm{DS}}(x, \mathsf{A}, D) \triangleq \mathbb{E}_{H \subset D \setminus x, S \sim O_H, S' \sim O_{H \cup \{x\}}}\big[|\mathrm{dist}(x, S) - \mathrm{dist}(x, S')|\big]. \tag{7}$$

Here, $S$ is the synthetic dataset generated from $O_H$, $\mathrm{dist}(x, S)$ denotes the nearest distance (under $l_1$ norm) between record $x$ and synthetic dataset $S$. (Empirically we find that the difference between using $l_1$ and $l_2$ distance is negligible.) However, directly computing Equation (7) is computationally expensive because it requires training models on paired subsets $H$ and $H \cup \{x\}$ for every record $x$. To address this, we employ the shadow training technique commonly used in MIAs. Specifically, we train $m$ synthesizers on independently sampled subsets $H_1, ..., H_m$ of equal size $|H_i| = \lfloor \frac{1}{2}|D| \rfloor$. To calculate the disclosure risk of $x$, we divide these models into two groups: one trained on subsets where $x \in H$, and the other where $x \notin H$. For each model trained on these subsets, we randomly generate $n$ synthetic datasets and take the average nearest distance to $x$. By doing so, we only need to train $m$ synthesizers and sample $n$ synthetic datasets per synthesizer. Finally, we define the privacy risk of a synthesizer A on $D$ to be the *maximum* disclosure risk across all training data:

**Definition 3** (Membership Disclosure Score). *Let $S$ be the sampled synthetic data from the synthesizer's output distribution $O_H$. The membership disclosure score of A is given by:*

$$\mathrm{MDS}(\mathsf{A}) \triangleq \max_{x \in D} \Big| \underbrace{\mathbb{E}_{H \subset D, S \sim O_{H \cup \{x\}}}[\mathrm{dist}(x, S)]}_{\text{closeness of } x \text{ when trained with } x} - \underbrace{\mathbb{E}_{H \subset D \setminus x, S' \sim O_H}[\mathrm{dist}(x, S')]}_{\text{closeness of } x \text{ when \textbf{not} trained with } x} \Big|, \tag{8}$$

In practice, we train 20 models and generate 100 synthetic datasets per model to compute MDS for all synthesizers. We analyze the effectiveness and efficiency of MDS in Section 5.2.

Figure 1: Overview of SynMeter.

Table 1: Performance improvements (%) with the proposed tuning objective.

| Synthesizer | Fidelity ↑ | | Utility ↑ | |
|---|---|---|---|---|
| | $D_{\text{Train}}$ | $D_{\text{Test}}$ | MLA | Query Error |
| MST | 0.33 | 0.34 | 17.35 | 3.39 |
| PrivSyn | 1.60 | 2.92 | 12.08 | 1.12 |
| TVAE | 1.06 | 0.67 | 5.29 | 2.67 |
| CTGAN | 9.87 | 9.60 | 0.57 | 8.63 |
| PATE-GAN | 6.27 | 8.48 | 0.75 | 7.04 |
| TabDDPM | 13.62 | 13.65 | 13.67 | 11.95 |
| TableDiffusion | 11.34 | 10.95 | 8.32 | 7.86 |
| GReaT | 3.84 | 9.21 | 1.14 | 1.77 |

**Limitations of MDS.** Although we find MDS to be effective in assessing the privacy risks of the synthesizers studied, we note that it has its own limitations. For instance, MDS can be tricked by carefully designed pathological synthesizers and should not be used as the only privacy measure where privacy is paramount. In addition, it is also incapable of measuring all types of privacy risks associated with syntehsizers. We refer Appendix H for a detailed discussion about its limitations.

### 3.3 UTILITY EVALUATION

**Existing Metrics and Limitations.** Machine learning efficacy (Xu et al., 2019) has emerged as the predominant utility metric for data synthesis. It first chooses a machine learning model (*i.e.,* evaluator), then assesses the testing accuracy on real data after training the evaluator on synthetic datasets. However, there is no consensus on which evaluator should be used for evaluation. Different evaluators yield varying performance outcomes on synthetic data, and no single model consistently achieves the best performance across all datasets. (We show the case in Appendix G.3.)

**Proposed Metrics: Machine Learning Affinity (MLA) and Query Error.** To accurately reflect the performance degradation caused by the distribution shift of synthetic data (Lopes et al., 2021), we follow (Jordon et al., 2021) and measure the relative performance gap as the utility metric:

**Definition 4** (Machine Learning Affinity). *Let $\mathcal{E}$ be a set of candidate machine learning models (i.e., evaluators), let $e_{D_{train}}$ and $e_S$ be evaluators trained on real training data $D_{train}$ and synthetic data $S$, $acc(e, D_{test})$ denotes the evaluator's accuracy (F1 score or RMSE) when performed on test dataset $D_{test}$. The MLA of synthesizer A is given by:*

$$\text{MLA(A)} := \frac{1}{|\mathcal{E}|} \sum_{e \in \mathcal{E}} \left[ \frac{acc(e_{D_{train}}, D_{test}) - acc(e_S, D_{test})}{acc(e_{D_{train}}, D_{test})} \right]. \tag{9}$$

A lower MLA score indicates a higher utility of synthetic data on the prediction task.

In addition to machine learning prediction, range/point queries are workhorses of statistical data analysis. However, these tasks are often overlooked when evaluating state-of-the-art synthesizers. We follow (McKenna et al., 2019) to define the query error as below:

**Definition 5** (Query Error). *Consider a subset of $k$ attributes $a = \{a_1, ..., a_k\}$ sampled from dataset $D$. For each attribute, if $a_i$ is categorical, a value $v_i$ is randomly chosen from its domain $\mathbb{R}(a_i)$, which forms the basis for a point query condition; for numerical attributes, two values $s_i$ and $d_i$ from $\mathbb{R}(a_i)$ are randomly sampled as the start and end points, to construct a range query condition. The final query $c \in \mathcal{C}$ combines $k$ sub-queries and is executed on both real and synthetic data to obtain query frequency ratios $\mu_c^{D_{test}}$ and $\mu_c^S$. The query error of synthesis A is defined as:*

$$\text{QueryError(A)} := \frac{1}{|\mathcal{C}|} \sum_{c \in \mathcal{C}} \left[ ||\mu_c^{D_{test}} - \mu_c^S||_1 \right]. \tag{10}$$

## 4 A SYSTEMATIC EVALUATION FRAMEWORK FOR DATA SYNTHESIS

**Tuning Objective.** Most synthesizers do not provide guidelines for hyperparameter tuning. Instead, default settings are commonly used for evaluations. This practice can lead to suboptimal results and biased comparisons. To address this issue, we propose a simple tuning objective using proposed evaluation metrics to facilitate the hyperparameter selection:

$$\mathcal{L}(\text{A}) = \alpha_1 \text{Fidelity(A)} + \alpha_2 \text{MLA(A)} + \alpha_3 \text{QueryError(A)}. \tag{11}$$

Since smaller values indicate better performance for all proposed metrics, we conduct a grid search on synthesizers and select the best hyperparameters that minimize $\mathcal{L}$ for evaluation. The privacy evaluation metric is excluded from model tuning, as we find that incorporating MDS yields negligible improvements for synthesizers. We show how to set the coefficients ($\alpha_1$, $\alpha_2$, $\alpha_3$) in Section 5.2.

**SynMeter.** We introduce a modular toolkit called SynMeter to assess data synthesis algorithms with proposed evaluation metrics. As depicted in Figure 1, SynMeter comprises four modules, and each module is implemented with an abstract interface for any synthesizer. (The detailed description of the evaluation pipeline is in Appendix A). We envisage that SynMeter can be used to (i) facilitate data owners to tune, train, and select different synthesizers for data publishing; and (ii) serve as a benchmark for data synthesis, providing systematic evaluation metrics for comparative studies.

## 5 EXPERIMENTS

We present a series of comprehensive experiments to answer the following question:

- **RQ1:** How effective are our proposed privacy evaluation metric and tuning objective?
- **RQ2:** How do the various synthesizers perform under our assessment? What are the new findings?
- **RQ3:** Why do these methods work well (or not so well) on certain aspects? How can our metrics help for in-depth analysis?

### 5.1 EXPERIMENTAL SETUPS

**Datasets.** We evaluate using 12 public real-world datasets with varying sizes, types, attributes, and distributions. Table 2 summarizes their statistics, with detailed descriptions in Appendix B.2.

**Data Synthesis Algorithms.** We study a wide range of HP and DP synthesizers. Specifically, we evaluate six types of HP synthesizers: the non-private version of MST (McKenna et al., 2021), the non-private version of PrivSyn (Zhang et al., 2021), CTGAN (Xu et al., 2019), TabDDPM (Kotelnikov et al., 2023), and REaLTabFormer (Solatorio & Dupriez, 2023). For DP synthesizers, we assess four types: MST, PrivSyn, PATE-GAN (Jordon et al., 2018) and TableDiffusion (Truda, 2023). Detailed descriptions of these synthesizers are in Appendix B.3.

Note that our goal is not to benchmark all synthesizers but to focus on the best-known and broad spectrum of SOTA synthesizers. TabSyn (Zhang et al., 2024) is a recent diffusion-based model that is claimed to outperform TabDDPM. We found that once TabDDPM is tuned with SynMeter, it achieves a similar performance as TabSyn. Results of other synthesizers are in Appendix C.6.

**Implementation.** During the evaluation, we first tune the synthesizers with the proposed tuning objective. Then, synthetic data are generated by the trained synthesizer for evaluation, where we test 20 times and report the mean and standard deviation as the final score. The hyperparameter search spaces of data synthesis algorithms are shown in Appendix E and the implementation details of the proposed metrics are in Appendix B.1.

### 5.2 EFFECTIVENESS OF MDS AND TUNING OBJECTIVE (RQ1)

**Effectiveness of MDS.** We compare MDS against the popular syntactic privacy evaluation metric DCR (Zhao et al., 2021), as well as three state-of-the-art MIAs: Groundhog (Stadler et al., 2022), TAPAS (Houssiau et al., 2022) and MODIAS (van Breugel et al., 2023). For DCR, we calculate the nearest distance of each synthetic record to real data, using the 5th percentile of the distance distribution as the privacy score. For MIAs, we follow Carlini et al. (2022) and use the true positive rate at 1% false positive rate (TPR@1%FPR) to measure the attack performance.

We conduct two proof-of-concept experiments to evaluate the effectiveness of MDS. First, we train a DP synthesizer (PATE-GAN) with varying levels of privacy protection by adjusting the privacy budget, and we measure the empirical privacy risk using these privacy evaluation metrics. Second, we train an HP synthesizer (TabDDPM) with different duplication ratios while keeping the training data size unchanged. Intuitively, a higher proportion of duplicate samples in the training set increases the memorization of the model, which in turn poses higher privacy risks (Carlini et al., 2023).

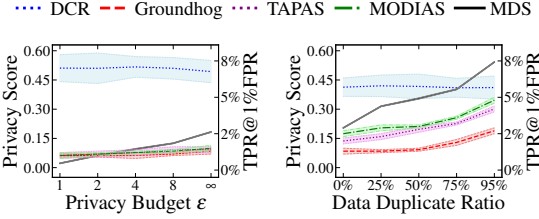

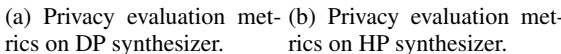

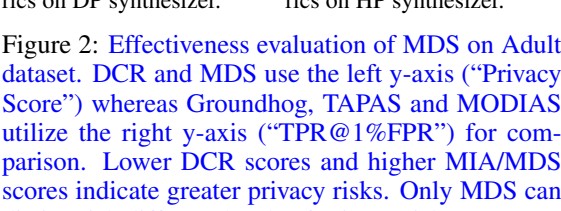

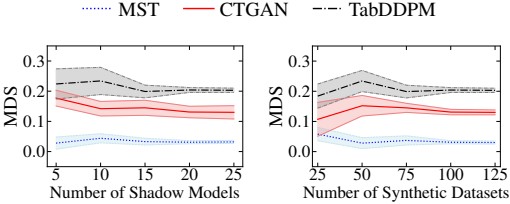

(a) Privacy evaluation metrics on DP synthesizer.    (b) Privacy evaluation metrics on HP synthesizer.

(a) Impact of the number of shadow models.    (b) Impact of the number of synthetic datasets.

Figure 2: Effectiveness evaluation of MDS on Adult dataset. DCR and MDS use the left y-axis ("Privacy Score") whereas Groundhog, TAPAS and MODIAS utilize the right y-axis ("TPR@1%FPR") for comparison. Lower DCR scores and higher MIA/MDS scores indicate greater privacy risks. Only MDS can distinguish different levels of privacy risks.

Figure 3: Stability evaluation of MDS on Adult dataset. We vary the number of shadow models and synthetic datasets used for computing MDS. The MDS of all three synthesizers can be accurately computed using 20 shadow models and 100 synthetic datasets.

The results of both experiments are presented in Figure 2. DCR fails to distinguish between different levels of privacy risk in both scenarios and exhibits significant instability (indicated by large standard deviations). For MIAs, we observe an improvement in attack performance as the proportion of duplicates in the training set increases, especially for MODIAS. However, MIAs still struggle to capture privacy nuances with DP synthesizers. In contrast, MDS effectively detects privacy risks across all scenarios and demonstrates robustness as a reliable privacy evaluation metric, as evidenced by its high standard deviation. Additional experiments on other existing metrics are in Appendix C.4.

**Stability and Efficiency of MDS.** We validate the stability of MDS by varying the number of shadow models and synthetic datasets. Specifically, we compute the membership disclosure scores for three synthesizers using different quantities of shadow models and synthetic datasets, recording the mean and variance of the results, as depicted in Figure 3. Our results indicate that the variance of MDS decreases rapidly as the number of shadow models and synthetic datasets increases, with stable results achieved using 20 shadow models and 100 synthetic datasets. Although MDS requires training more shadow models compared to existing MIAs, previous study (Zhang et al., 2024) shows that tabular synthesizers can be trained in just a few minutes, with sampling taking only a few seconds. Therefore, MDS remains a practical and efficient solution for privacy assessment.

**Effectiveness of Tuning Objective.** Although the metrics in Equation (11) are based on different measurements, empirically we observe that their values consistently fall within the same range. Consequently, in our experiments, we set all three coefficients to $1/3$, as this configuration significantly improves the quality of synthetic data, as shown in Table 1. Interestingly, the tuning phase affects two types of synthesizers differently: statistical methods gain more in utility than fidelity, while deep generative models show the opposite trend. Notably, the tuning phase proves especially beneficial for TabDDPM, with improvements in both fidelity and utility metrics. Additional experiments on the effectiveness of the proposed tuning objective are provided in Appendix C.5.

## 5.3 OVERALL EVALUATION (RQ2)

**Overview.** Figure 4 and Figure 5 report the overview ranking results for HP and DP synthesizers, respectively. For HP synthesizers, TabDDPM and REaLTabFormer exhibit superior fidelity and utility, albeit at the expense of compromising privacy. Statistical methods like PrivSyn achieve good fidelity while offering impressive privacy protection. Conversely, CTGAN, the most popular HP synthesizer, shows the least satisfactory results in synthetic data quality. For DP synthesizers, statistical methods remain effective in both fidelity and utility. The performance of deep generative models drops significantly to satisfy differential privacy. Even the strongest model (*i.e.,* TableDiffusion) underperforms statistical approaches by a large margin, which starkly contrasts with its performance in the HP context, indicating a pronounced impact of privacy constraints on deep generative models. The visualization of synthetic and real data is depicted in Figure 8 and Figure 9.

**Fidelity Evaluation.** We introduce two baselines to establish empirical lower and upper bounds for the proposed fidelity metric. The first baseline, HALF, randomly divides the real data into two equal

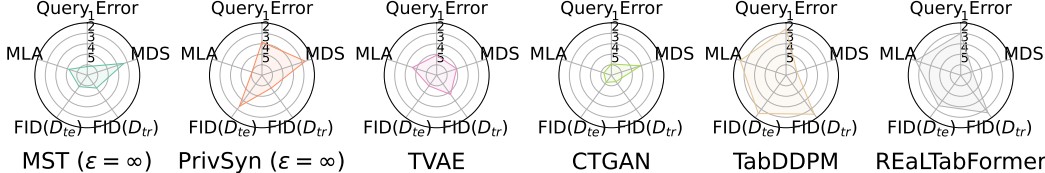

Figure 4: Average **ranking** comparison for six HP synthesizers (outer means higher rank and better performance). Each vertex is the average rank of the method across 12 datasets, and each axis is the evaluation metric. "FID($D_{\text{tr}}/D_{\text{te}}$)" denotes the fidelity evaluated on the training/test dataset. "MDS" is the proposed privacy evaluation metric, and "MLA" and "Query Error" are utility metrics.

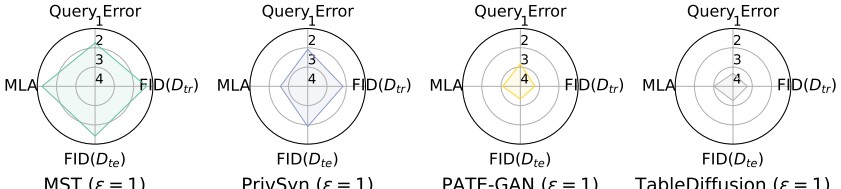

Figure 5: Average **ranking** comparison for four DP synthesizers. All methods offer provable privacy guarantees so we remove the privacy axis for comparison.

parts, using one as the training dataset $D$ and the other as the synthetic data $S$. Since both datasets are from the same distribution, this serves as the empirical upper bound of fidelity. The second baseline, HISTOGRAM, generates synthetic data using one-way marginals without accounting for correlations between attributes, making it the empirical lower bound of fidelity.

Fidelity is evaluated by applying the Wasserstein distance to both the training dataset $D_{\text{train}}$ (Table 3 and Table 4) and the test dataset $D_{\text{test}}$ (Table 5 and Table 6). The results show that TabDDPM and REaLTabFormer achieve near upper-bound fidelity, while statistical methods such as MST excel among DP synthesizers. Notably, all deep generative models experience a significant drop in fidelity when achieving differential privacy, whereas statistical methods maintain consistent performance.

**Privacy Evaluation.** We utilize SELF as the baseline to represent the lower bound of MDS. Specifically, SELF uses a direct copy of the real data as synthetic data, establishing the worst privacy protection. According to the definition of MDS, an ideal privacy-preserving synthesizer would achieve a score of 0, which is the upper bound of privacy evaluation.

Table 7 and Table 8 show the privacy assessment results for HP synthesizers. In contrast to the fidelity evaluation, CTGAN, which exhibits the lowest fidelity performance, offers impressive privacy protection against membership disclosure. Statistical methods like MST also show notable empirical privacy protections. However, the unsatisfied results of strong synthesis algorithms like TabDDPM and REaLTabFormer reveal their vulnerability to membership disclosure.

**Utility Evaluation.** The utility of data synthesis is assessed by performing downstream tasks on the synthetic datasets and measuring their performance using the proposed metrics, as shown in Table 9-12. For machine learning tasks, TabDDPM excels among HP synthesizers, contributing to its class-conditional framework that learns label dependencies during its training process. However, this advantage diminishes when adding random noise to ensure privacy, where MST takes the lead with its robust and superior performance. The outcomes for range (point) query tasks echo the results of fidelity evaluation, where TabDDPM shows superior performance in HP settings, and statistical methods (*e.g.,* MST) can surpass other methods under DP constraints.

### 5.4 In-depth Analysis (RQ3)

**Why Does CTGAN Perform Poorly?** Despite CTGAN is widely regarded as a strong synthesizer, our evaluation reveals that it produces the lowest-quality synthetic data. This discrepancy raises important questions about the reasons behind CTGAN's apparent underperformance. To investigate this, we scrutinize its learning trajectory, particularly evaluating the fidelity across different marginal types during training. As shown in Figure 6(a), both numerical and categorical marginals exhibit unexpected stagnation in improvement. This suggests that CTGAN's synthetic data quality is heavily influenced by data preprocessing. Specifically, CTGAN relies on a variational Gaussian

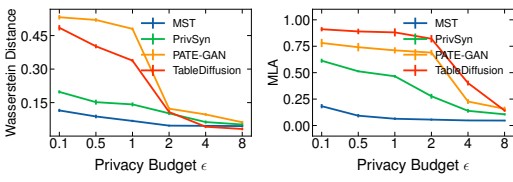

(a) CTGAN    (b) TabDDPM

Figure 6: Analyzing the learning process of CT-GAN and TabDDPM with proposed fidelity metrics on the Bean dataset.

Figure 7: Impact of privacy budget $\epsilon$ on Bean dataset. The lower score indicates higher fidelity/utility.

mixture model for numerical data and conditional sampling for categorical attributes. The model performs well when the data distribution is close to Gaussian; however, most tabular datasets are far more complex and deviate significantly from this assumption (Gorishniy et al., 2021). This mismatch largely explains CTGAN's suboptimal performance. Furthermore, this limitation may also account for CTGAN's strong empirical privacy protections. The model's difficulty in learning complex data structures results in outputs that are largely independent of any individual training sample, contributing to its good privacy protection.

**Why Does TabDDPM Excel?** One key finding of our evaluations is the TabDDPM's ability to synthesize high-quality tabular data. This challenges previous claims that deep generative models generally struggle for tabular data synthesis (Tao et al., 2021). We also use proposed fidelity metrics to analyze TabDDPM's learning process. As illustrated in Figure 6(b), the Wasserstein distance across all marginal distributions rapidly decreases, demonstrating the model's capacity to learn both numerical and categorical distributions. We attribute this success to the model's architecture: diffusion models have been shown to effectively minimize the Wasserstein distance between synthetic and real data (Kwon et al., 2022). This offers a methodological advantage over other generative models, which usually aim to minimize the Kullback-Leibler divergence. However, despite its strengths, TabDDPM presents significant privacy risks that have been largely overlooked in prior research. Directly applying differential privacy measures would severely degrade the quality of the synthetic data. Nevertheless, diffusion-based methods remain a promising frontier for tabular data synthesis.

**Large Language Models Are Semantic-aware Synthesizers.** We also notice that the recently emerged LLM-based synthesizer (*i.e.,* REaLTabFormer) also shows competitive performance, especially on datasets that consist of rich semantic attributes and complex dependence. For instance, REaLTabFormer achieves the best machine learning prediction performance on the Adult dataset, which contains detailed personal information (*e.g.,* age and relationship). Given the rapid development of LLM and the inherent rich semantics of most tabular data, LLM-based methods may become a new paradigm for realistic data synthesis.

**The Impact of Privacy Budget.** To analyze the impact of differential privacy on data synthesis, we run DP synthesizers with varying privacy budgets, and evaluate the fidelity and utility of the resulting synthetic data (see Figure 7). Our results show that statistical methods, such as MST, maintain robust performance even with a small privacy budget (*e.g.,* $\epsilon = 0.5$). In contrast, deep generative models typically require much larger privacy budgets (*e.g.,* $\epsilon = 8$) to achieve comparable results. These findings align with previous observations (Tao et al., 2021), which noted that statistical methods are more resilient to privacy constraints because they rely on estimating a small set of marginals.

## 6 RELATED WORK

**Fidelity Evaluation Metrics.** Fidelity is often evaluated based on the distributional similarities of low-order marginals with various statistical measurements. Total Variation Distance (Zhang et al., 2024) and one-dimensional Wasserstein distance (Zhao et al., 2024; Lin et al., 2020) are used to assess univariate distribution similarity for categorical and numerical attributes, respectively. Correlation differences are widely employed for bivariate distributions. Correlation statistics such as Theil's uncertainty coefficient (Zhao et al., 2021), Pearson correlation (Zhang et al., 2024), and the correlation ratio (Kotelnikov et al., 2023) are utilized to evaluate different types of two-way marginals (categorical, continuous, and mixed). The main problem with these measures is the lack of versatility. Each type of marginal requires a distinct statistical measure, which complicates the ability to

perform a comprehensive comparison across various attribute types. We refer to Appendix G.1 for a detailed discussion of the limitations of existing fidelity metrics.

**Privacy Evaluation Metrics.** Since HP synthesizers are designed without provable privacy guarantees, privacy evaluation is indispensable for these synthesizers. Syntactic privacy evaluation metrics (*e.g.*, Distance to Closest Records (Zhao et al., 2021)) are the most widely used privacy evaluation for HP synthesizer. These metrics compare the input dataset with the output dataset generated by the synthesizer, with closer distances indicating higher privacy risks. Recently, Ganev & De Cristofaro (2023) critiqued these syntactic metrics, highlighting that these ad-hoc metrics can be exploited for reconstruction attacks. However, the study did not address the fundamental flaws of these metrics (discussed in Section 3.2) and did not introduce new and effective privacy evaluation metrics. Another way to assess the empirical privacy risks of data synthesis is membership inference attack (MIA) (Shokri et al., 2017). Some studies (Stadler et al., 2022; van Breugel et al., 2023) have designed different MIA algorithms for tabular data synthesis. However, as shown in Section 5.2, existing MIA algorithms are too weak to differentiate different privacy risks across various synthesizers. Further discussion about existing privacy evaluation metrics can be found in Appendix G.2.

**Utility Evaluation Metrics.** Machine learning prediction and query errors are common downstream tasks for tabular data analysis, and many studies (Zhang et al., 2021; Xu et al., 2019; McKenna et al., 2021) have leveraged these tasks to evaluate the utility of synthetic data. In our evaluation, we also adopt these tasks for utility evaluation and present a reliable metric to address the variability in performance across different machine learning models (Jordon et al., 2021). Further discussion on utility metrics can be found in Appendix G.3.

**Benchmarking Tabular Data Synthesis.** Several studies have benchmarked tabular synthesis algorithms. However, they either only focus on DP synthesizers (Tao et al., 2021; Hu et al., 2024), or neglect the privacy evaluation for HP synthesizers (Espinosa & Figueira, 2023; Chundawat et al., 2022; Livieris et al., 2024; McLachlan et al., 2018). Additionally, existing benchmarks (Qian et al., 2024; Lautrup et al., 2024) directly leverage existing metrics for evaluation, whereas we identify the limitations of these metrics and propose a new set of evaluation metrics for systematic assessment.

# 7 DISCUSSION AND KEY TAKEAWAYS

In this paper, we examine and critique existing metrics, and introduce a systematic framework as well as a new suite of evaluation criteria for assessing data synthesizers. We also provide a unified tuning objective to ensure that evaluation results are less affected by accidental choices of hyperparameters. Our results identify several guidelines for data synthesis practitioners:

- *Model tuning is indispensable.* Tuning hyperparameters can significantly improve synthetic data quality, especially for deep generative models.
- *Statistical methods should be preferred for applications where privacy is paramount.* MST and PrivSyn achieve the best fidelity among DP synthesizers, and they also offer good empirical privacy protection even in HP settings.
- *Diffusion models provide the best fidelity and utility.* Practitioners are suggested to use diffusion models (*e.g.*, TabDDPM) for tabular synthesis when the quality of synthetic data is the priority over privacy due to their impressive ability to generate highly authentic data.
- *Deep generative models can be tailored for specific tasks.* The flexible design spaces of deep generative models make them suitable for scenarios where the applications of the synthetic data are known in advance (*e.g.*, machine learning prediction). In addition, the LLM-based synthesizer, REaLTabFormer, is particularly effective at preserving semantic information in synthetic data.

Our systematic assessment shows that recently emerged generative models achieve impressive performance on tabular data synthesis and open up new directions in this field. At the same time, several critical challenges are also revealed such as privacy issues of diffusion models and performance gaps between DP and HP synthesizers. In addition, we note that existing empirical privacy evaluation metrics (including proposed MDS) have their own limitations and DP synthesizers should be used in privacy-critical applications. Nevertheless, our evaluation metrics and framework serve a crucial role in highlighting advancements in data synthesis and represent a step toward establishing a standardized evaluation process for this field.

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

## A EVALUATION PIPELINES

The SynMeter pipeline consists of four phases: data preparation, model tuning, model training, and model evaluation.

The *data preparation* phase preprocesses data for learning algorithms[1]. In this phase, statistical methods select low-dimensional marginals to serve as compact representations for capturing data distributions. Deep generative models apply standard data processing techniques like data encoding and normalization.

The goal of *model tuning* phase is to select the optimal hyperparameters for data synthesizers. We use the proposed tuning objective in Equation (11) for hyperparameter selections.

The *model training* phase focuses on model learning with tuned hyperparameters. Various generative models implement different architectures and optimization objectives.

In the *model evaluation* phase, the trained model samples some synthetic data, which are used for evaluation. Specifically, we assess the fidelity, privacy, and utility of synthesizers via the proposed metrics.

## B DETAILS OF EXPERIMENTAL SETUPS

### B.1 IMPLEMENTATION DETAILS

**Wasserstein-based Fidelity Metric.** The computation of Wasserstein distance involves solving the linear programming problem in Equation 1 and selecting proper marginal distributions. We compute the Wasserstein distance of all the one-way and two-way marginals and use the mean as the final fidelity score. The real dataset $D$ can be designated as either $D_{train}$ or $D_{test}$ to evaluate the fidelity of synthesizers on training data or test data.

There are many open-source libraries like CVXPY (Diamond & Boyd, 2016) and POT (Flamary et al., 2021) that can be used to solve linear programming reasonably fast. However, when the cost matrix becomes rather large and dense, directly calculating the metric can be computationally expensive. Several options are provided to address this problem: (i) Sinkhorn distance (Cuturi, 2013) provides a fast approximation to the Wasserstein distance by penalizing the objective with an entropy term. (ii) Sliced-Wasserstein distance (Bonneel et al., 2015), which uses Radon transform to linearly project data into one dimension, can be efficiently computed. (iii) Reducing the size of the cost matrix by randomly sampling a small set of points from the probability densities. In practice, we find that sampling is both efficient and effective. We randomly sample half of the synthetic data when $n > 5,000$ and use the POT library to compute the Wasserstein distance as the fidelity scores.

**Membership Disclosure Score (MDS).** We follow previous work (Carlini et al., 2022) and use shadow models to compute MDS. Specifically, we trained the synthesizer using half of the dataset and kept the other half as non-members for each shadow model. Once the synthesizer was trained, we randomly generated 100 synthetic datasets with the same size of training data and calculated the average closeness difference as the disclosure score. The MDS is computed as the maximum disclosure score across all records.

**Utility Metrics.** For machine learning affinity (MLA), we utilize eight machine learning models to compute MLA: SVM, Logistic Regression (or Ridge Regression), Decision Tree, Random Forest, Multilayer Perceptron (MLP), XGBoost (Chen & Guestrin, 2016), CatBoost (Prokhorenkova et al., 2018), and Transformers (Gorishniy et al., 2021). Each model is extensively tuned on real training data to ensure optimal hyperparameters. Performance on classification and regression is evaluated by the F1 score and RMSE, respectively. For query error, we randomly construct 1,000 3-way query conditions and conduct range (point) queries for both synthetic and real data.

---

[1]Here we assume no missing values in the original data. The missing values problem has been extensively studied (Pigott, 2001), which is orthogonal to data synthesis.

Table 2: Statistics of datasets. # Num stands for the number of numerical columns, and # Cat stands for the number of categorical columns.

| Dataset | # Train | # Validation | # Test | # Num | # Cat | Task type |
|---------|---------|--------------|--------|-------|-------|-----------|
| **Adult** | 20838 | 5210 | 6513 | 6 | 9 | Binclass |
| **Shoppers** | 7891 | 1973 | 2466 | 10 | 8 | Binclass |
| **Phishing** | 7075 | 1769 | 2211 | 0 | 31 | Binclass |
| **Magic** | 12172 | 3044 | 3804 | 10 | 1 | Binclass |
| **Faults** | 1241 | 311 | 389 | 24 | 4 | Multiclass(7) |
| **Bean** | 8710 | 2178 | 2723 | 16 | 1 | Multiclass(7) |
| **Obesity** | 1350 | 338 | 423 | 8 | 9 | Multiclass(7) |
| **Robot** | 3491 | 873 | 1092 | 24 | 1 | Multiclass(4) |
| **Abalone** | 2672 | 668 | 836 | 8 | 1 | Regression |
| **News** | 25372 | 6343 | 7929 | 46 | 14 | Regression |
| **Insurance** | 856 | 214 | 268 | 3 | 4 | Regression |
| **Wine** | 3134 | 784 | 980 | 12 | 0 | Regression |

## B.2 DATASETS

We use 12 real-world datasets for evaluations. These datasets have various sizes, natures, attributes, and distributions. We explicitly divide datasets into training and test with a ratio of 8:2, then split 20% of the training dataset as the validation set, which is used for model tuning. The statistics of the datasets are presented in Table 2. Below is a detailed introduction to each dataset:

- **Adult**[2] is to predict whether income exceeds 50K/yr based on census data.
- **Shoppers**[3] is to analyze the intention of online shoppers.
- **Phishing**[4] is to predict if a webpage is a phishing site. The dataset consists of important features for predicting phishing sites, including information about webpage transactions.
- **Magic**[5] is to simulate the registration of high-energy gamma particles in the atmospheric telescope.
- **Faults**[6] is the fault detection dataset, which classified steel plates faults into 7 different types.
- **Bean**[7] predicts the type of dray bean based on form, shape, and structure.
- **Obesity**[8] is to estimate the obesity level based on eating habits and physical condition of individuals from Mexico, Peru, and Columbia.
- **Robot**[9] is a multi-class classification dataset collected as the robot moves around the room, following the wall using ultrasound sensors.
- **Abalone**[10] is to predict the age of abalone from physical measurements.
- **News**[11] is to predict the number of shares in social networks (popularity).
- **Insurance**[12] is for prediction on the yearly medical cover cost. The dataset contains a person's medical information.
- **Wine**[13] collects physicochemical tests on wine.

---

[2] https://archive.ics.uci.edu/dataset/2/adult
[3] https://archive.ics.uci.edu/dataset/468/online+shoppers+purchasing+intention+dataset
[4] https://archive.ics.uci.edu/dataset/468/online+shoppers+purchasing+intention+dataset
[5] https://archive.ics.uci.edu/dataset/159/magic+gamma+telescope
[6] https://archive.ics.uci.edu/dataset/198/steel+plates+faults
[7] https://archive.ics.uci.edu/dataset/602/dry+bean+dataset
[8] https://archive.ics.uci.edu/dataset/544/estimation+of+obesity+levels+based+on+eating+habits+and+physical+condition
[9] https://archive.ics.uci.edu/dataset/194/wall+following+robot+navigation+data
[10] https://archive.ics.uci.edu/dataset/1/abalone
[11] https://archive.ics.uci.edu/dataset/332/online+news+popularity
[12] https://www.kaggle.com/datasets/tejashvi14/medical-insurance-premium-prediction
[13] https://archive.ics.uci.edu/dataset/186/wine+quality

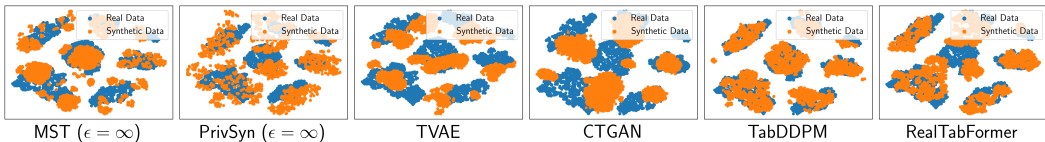

MST ($\epsilon = \infty$)  PrivSyn ($\epsilon = \infty$)  TVAE  CTGAN  TabDDPM  RealTabFormer

Figure 8: Visualization comparison of HP synthesizers on Bean dataset with t-SNE (Van der Maaten & Hinton, 2008). Real data are in blue and synthetic data are in orange.

### B.3 USED DATA SYNTHESIS ALGORITHMS

We study a wide range of state-of-the-art synthesizers, from statistical methods to deep generative models. We select them as they are either generally considered to perform best in practice (McKenna et al., 2021; Zhang et al., 2021), widely used (Xu et al., 2019; Papernot et al., 2018), or recently emerged (Kotelnikov et al., 2023; Borisov et al., 2023; Truda, 2023). These synthesizers can be categorized into two groups: heuristic private (HP) and differentially private (DP) synthesizers.

*HP Synthesizers.* Synthesizers in this category are developed without integrating DP:

- **CTGAN** (Xu et al., 2019) is one of the most widely used HP synthesis algorithms. It utilizes generative adversarial networks to learn tabular data distributions. Training techniques like conditional generation and Wasserstein loss (Gulrajani et al., 2017) are used.
- **TVAE** (Xu et al., 2019) is the state-of-the-art variational autoencoder for tabular data synthesizer, which uses mode-specific normalization to tackle the non-Gaussian problems of continuous distributions.
- **TabDDPM** (Kotelnikov et al., 2023) is the state-of-the-art diffusion model for data synthesis. It leverages the Gaussian diffusion process and the multinomial diffusion process to model continuous and discrete distributions respectively.
- **GReaT** Borisov et al. (2023) utilizes the large language model (LLM) for data synthesis. It converts records to textual representations for LLM and generates synthetic data with prompts.

*DP Synthesizers.* These methods are either inherently designed with DP or are adaptations of HP models with additional mechanisms to offer provable privacy guarantees:

- **MST** (McKenna et al., 2021) is the state-of-the-art DP synthesizer, which uses probabilistic graphical models McKenna et al. (2019) to learn the dependence of low-dimensional marginals. It won the NIST Differential Privacy Synthetic Data Challenge NIST (2018). Discrete binning is applied for numerical attributes.
- **PrivSyn** (Zhang et al., 2021) is a non-parametric DP synthesizer, which iteratively updates the synthetic dataset to make it match the target noise marginals. This method also shows strong performance in NIST competitions (NIST, 2018; 2020). Discretization is also used for modeling numerical attributes.
- **PATE-GAN** (Jordon et al., 2018) shares a similar architecture with CTGAN, but leverages the Private Aggregation of Teacher Ensembles (PATE) (Papernot et al., 2018) to offer DP guarantees.
- **TableDiffusion** (Truda, 2023) is a newly proposed diffusion model for data synthesis, which uses Differentially Private Stochastic Gradient Descent (DP-SGD) to enforce privacy.

All DP synthesizers can be adapted to the HP scenario either by using their HP counterparts[14] (*i.e.,* CTGAN for PATE-GAN, TabDDPM for TableDiffusion) or by setting the privacy budget to infinity (*i.e.,* MST and PrivSyn). However, some HP synthesizers, such as TVAE and GReaT, do not have corresponding DP variants. Thus, we only assess their performance within the context of HP models.

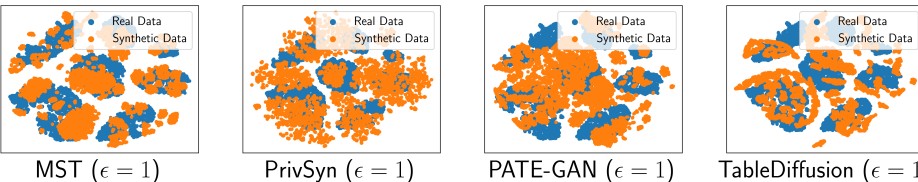

MST ($\epsilon = 1$)    PrivSyn ($\epsilon = 1$)    PATE-GAN ($\epsilon = 1$)    TableDiffusion ($\epsilon = 1$)

Figure 9: Visualization comparison of DP synthesizers on Bean dataset with t-SNE. Real data are in blue and synthetic data are in orange.

## C  ADDITIONAL EXPERIMENTS AND RESULTS

### C.1  FIDELITY RESULTS

Here we include the complete fidelity results in our evaluation. The fidelity evaluation on train data is shown in Table 5 and Table 4. The fidelity evaluation on test data is demonstrated in Table 5 and Table 6. It is observed that TabDDPM outperforms other HP synthesizers on most datasets, and statistical methods (*i.e.,* MST and PrivSyn) achieve the best performance when DP is required.

Table 3: Fidelity evaluation (lower score indicates better fidelity) of synthesizers on training data $D_{\text{train}}$ of first six datasets. The privacy budget $\epsilon$ of HP synthesizers is $\infty$ (the top part), and the budget for DP synthesizers is 1 (the middle part). HALF and HISTOGRAM are the baselines that serve as the empirical upper/lower bound of the fidelity for HP synthesizers. The best result is in bold.

|  | Adult | Shoppers | Phishing | Magic | Faults | Bean |
|---|---|---|---|---|---|---|
| MST | $0.186_{\pm.010}$ | $0.092_{\pm.002}$ | $0.019_{\pm.001}$ | $0.037_{\pm.002}$ | $0.056_{\pm.002}$ | $0.040_{\pm.002}$ |
| PrivSyn | $0.024_{\pm.001}$ | $0.030_{\pm.001}$ | $\mathbf{0.010}_{\pm.001}$ | $0.015_{\pm.003}$ | $0.064_{\pm.006}$ | $0.035_{\pm.002}$ |
| TVAE | $0.085_{\pm.002}$ | $0.156_{\pm.001}$ | $0.024_{\pm.001}$ | $0.021_{\pm.003}$ | $0.055_{\pm.007}$ | $0.047_{\pm.006}$ |
| CTGAN | $0.059_{\pm.001}$ | $0.062_{\pm.001}$ | $0.062_{\pm.002}$ | $0.157_{\pm.006}$ | $0.133_{\pm.004}$ | $0.139_{\pm.005}$ |
| TabDDPM | $\mathbf{0.020}_{\pm.001}$ | $\mathbf{0.022}_{\pm.001}$ | $0.015_{\pm.001}$ | $\mathbf{0.011}_{\pm.003}$ | $\mathbf{0.026}_{\pm.002}$ | $\mathbf{0.015}_{\pm.002}$ |
| REaLTabFormer | $0.022_{\pm.002}$ | $0.024_{\pm.003}$ | $0.012_{\pm.001}$ | $0.045_{\pm.005}$ | $0.054_{\pm.005}$ | $0.035_{\pm.007}$ |
| MST ($\epsilon = 1$) | $0.198_{\pm.013}$ | $0.103_{\pm.002}$ | $\mathbf{0.023}_{\pm.001}$ | $\mathbf{0.042}_{\pm.003}$ | $\mathbf{0.086}_{\pm.003}$ | $\mathbf{0.048}_{\pm.004}$ |
| PrivSyn ($\epsilon = 1$) | $\mathbf{0.045}_{\pm.002}$ | $\mathbf{0.077}_{\pm.005}$ | $0.033_{\pm.002}$ | $0.052_{\pm.003}$ | $0.228_{\pm.007}$ | $0.142_{\pm.007}$ |
| PATE-GAN ($\epsilon = 1$) | $0.139_{\pm.001}$ | $0.176_{\pm.002}$ | $0.173_{\pm.002}$ | $0.153_{\pm.005}$ | $0.204_{\pm.003}$ | $0.520_{\pm.006}$ |
| TableDiffusion ($\epsilon = 1$) | $0.180_{\pm.002}$ | $0.209_{\pm.002}$ | $0.123_{\pm.002}$ | $0.132_{\pm.003}$ | $0.369_{\pm.002}$ | $0.148_{\pm.005}$ |
| HALF (upper bound) | $0.020_{\pm.002}$ | $0.018_{\pm.001}$ | $0.010_{\pm.002}$ | $0.011_{\pm.004}$ | $0.017_{\pm.002}$ | $0.015_{\pm.004}$ |
| HISTOGRAM (lower bound) | $0.213_{\pm.013}$ | $0.101_{\pm.003}$ | $0.027_{\pm.001}$ | $0.051_{\pm.003}$ | $0.081_{\pm.002}$ | $0.087_{\pm.002}$ |

Table 4: Fidelity evaluation (lower score indicates better fidelity) of synthesizers on training data $D_{\text{train}}$ of last six datasets. The privacy budget $\epsilon$ of HP synthesizers is $\infty$ (the top part), and the budget for DP synthesizers is 1 (the middle part). HALF and HISTOGRAM are the baselines that serve as the empirical upper/lower bound of the fidelity for HP synthesizers. The best result is in bold.

|  | Obesity | Robot | Abalone | News | Insurance | Wine |
|---|---|---|---|---|---|---|
| MST | $0.041_{\pm.001}$ | $0.050_{\pm.002}$ | $0.037_{\pm.002}$ | $0.060_{\pm.001}$ | $0.038_{\pm.005}$ | $0.066_{\pm.001}$ |
| PrivSyn | $0.034_{\pm.002}$ | $0.065_{\pm.012}$ | $0.024_{\pm.004}$ | $\mathbf{0.018}_{\pm.001}$ | $0.033_{\pm.002}$ | $0.017_{\pm.000}$ |
| TVAE | $0.055_{\pm.004}$ | $0.053_{\pm.001}$ | $0.048_{\pm.003}$ | $0.081_{\pm.001}$ | $0.078_{\pm.007}$ | $0.039_{\pm.000}$ |
| CTGAN | $0.072_{\pm.002}$ | $0.106_{\pm.003}$ | $0.049_{\pm.004}$ | $0.040_{\pm.001}$ | $0.090_{\pm.004}$ | $0.033_{\pm.001}$ |
| TabDDPM | $\mathbf{0.017}_{\pm.001}$ | $\mathbf{0.015}_{\pm.002}$ | $0.015_{\pm.004}$ | $0.034_{\pm.001}$ | $\mathbf{0.028}_{\pm.005}$ | $0.011_{\pm.000}$ |
| REaLTabFormer | $0.031_{\pm.004}$ | $0.029_{\pm.004}$ | $\mathbf{0.013}_{\pm.004}$ | $0.038_{\pm.001}$ | $0.033_{\pm.003}$ | $\mathbf{0.008}_{\pm.001}$ |
| MST ($\epsilon = 1$) | $\mathbf{0.063}_{\pm.001}$ | $\mathbf{0.065}_{\pm.001}$ | $\mathbf{0.052}_{\pm.003}$ | $\mathbf{0.062}_{\pm.002}$ | $\mathbf{0.071}_{\pm.002}$ | $\mathbf{0.068}_{\pm.001}$ |
| PrivSyn ($\epsilon = 1$) | $0.167_{\pm.009}$ | $0.169_{\pm.021}$ | $0.127_{\pm.009}$ | $0.070_{\pm.002}$ | $0.124_{\pm.006}$ | $0.156_{\pm.004}$ |
| PATE-GAN ($\epsilon = 1$) | $0.086_{\pm.003}$ | $0.477_{\pm.002}$ | $0.331_{\pm.005}$ | $0.065_{\pm.002}$ | $0.385_{\pm.003}$ | $0.251_{\pm.000}$ |
| TableDiffusion ($\epsilon = 1$) | $0.347_{\pm.003}$ | $0.203_{\pm.001}$ | $0.232_{\pm.005}$ | $0.135_{\pm.001}$ | $0.343_{\pm.002}$ | $0.108_{\pm.001}$ |
| HALF (upper bound) | $0.017_{\pm.003}$ | $0.010_{\pm.001}$ | $0.012_{\pm.004}$ | $0.009_{\pm.001}$ | $0.026_{\pm.004}$ | $0.006_{\pm.000}$ |
| HISTOGRAM (lower bound) | $0.051_{\pm.001}$ | $0.061_{\pm.002}$ | $0.069_{\pm.001}$ | $0.063_{\pm.002}$ | $0.046_{\pm.002}$ | $0.068_{\pm.000}$ |

---

[14]Although these paired models are quite different in the numbers of neural network layers, preprocessing, and learning strategies, they belong to the same type of generative model. Thus we call them "counterparts".

Table 5: Fidelity evaluation (*i.e.,* Wasserstein distance) of data synthesis algorithms on test data $D_{\text{test}}$ of the first six datasets. `HALF` and `HISTOGRAM` are the baselines that serve as the empirical upper and lower bounds of the fidelity for HP synthesizers. The low score indicates the synthesizer can generate high-quality synthetic data. The best result is in bold.

| | Adult | Shoppers | Phishing | Magic | Faults | Bean |
|---|---|---|---|---|---|---|
| MST | $0.172_{\pm.004}$ | $0.098_{\pm.002}$ | $0.026_{\pm.001}$ | $0.039_{\pm.002}$ | $0.089_{\pm.006}$ | $0.044_{\pm.003}$ |
| PrivSyn | $0.025_{\pm.001}$ | $0.041_{\pm.003}$ | $\mathbf{0.017}_{\pm\mathbf{.002}}$ | $0.015_{\pm.002}$ | $0.079_{\pm.007}$ | $0.037_{\pm.003}$ |
| TVAE | $0.086_{\pm.002}$ | $0.154_{\pm.002}$ | $0.028_{\pm.002}$ | $0.020_{\pm.003}$ | $0.081_{\pm.016}$ | $0.050_{\pm.004}$ |
| CTGAN | $0.061_{\pm.003}$ | $0.061_{\pm.002}$ | $0.069_{\pm.001}$ | $0.150_{\pm.004}$ | $0.133_{\pm.007}$ | $0.139_{\pm.005}$ |
| TabDDPM | $\mathbf{0.021}_{\pm\mathbf{.001}}$ | $0.031_{\pm.001}$ | $0.019_{\pm.001}$ | $\mathbf{0.012}_{\pm\mathbf{.002}}$ | $\mathbf{0.058}_{\pm\mathbf{.008}}$ | $\mathbf{0.016}_{\pm\mathbf{.003}}$ |
| REaLTabFormer | $0.021_{\pm.002}$ | $\mathbf{0.030}_{\pm\mathbf{.003}}$ | $0.018_{\pm.004}$ | $0.046_{\pm.003}$ | $0.075_{\pm.005}$ | $0.028_{\pm.004}$ |
| MST ($\epsilon = 1$) | $0.179_{\pm.004}$ | $0.103_{\pm.001}$ | $\mathbf{0.028}_{\pm\mathbf{.001}}$ | $0.042_{\pm.004}$ | $\mathbf{0.112}_{\pm\mathbf{.005}}$ | $\mathbf{0.048}_{\pm\mathbf{.003}}$ |
| PrivSyn ($\epsilon = 1$) | $\mathbf{0.049}_{\pm\mathbf{.002}}$ | $\mathbf{0.084}_{\pm\mathbf{.002}}$ | $0.030_{\pm.003}$ | $\mathbf{0.031}_{\pm\mathbf{.003}}$ | $0.236_{\pm.017}$ | $0.128_{\pm.010}$ |
| PATE-GAN ($\epsilon = 1$) | $0.139_{\pm.002}$ | $0.171_{\pm.002}$ | $0.173_{\pm.002}$ | $0.155_{\pm.005}$ | $0.215_{\pm.004}$ | $0.523_{\pm.004}$ |
| TableDiffusion ($\epsilon = 1$) | $0.179_{\pm.002}$ | $0.210_{\pm.002}$ | $0.121_{\pm.002}$ | $0.132_{\pm.005}$ | $0.390_{\pm.004}$ | $0.149_{\pm.004}$ |
| `HALF` (upper bound) | $0.022_{\pm.002}$ | $0.023_{\pm.002}$ | $0.016_{\pm.003}$ | $0.011_{\pm.003}$ | $0.042_{\pm.005}$ | $0.015_{\pm.003}$ |
| `HISTOGRAM` (lower bound) | $0.199_{\pm.017}$ | $0.101_{\pm.001}$ | $0.030_{\pm.001}$ | $0.048_{\pm.002}$ | $0.113_{\pm.006}$ | $0.080_{\pm.003}$ |

Table 6: Fidelity evaluation (*i.e.,* Wasserstein distance) of data synthesis algorithms on test data $D_{\text{test}}$ of the last six datasets. `HALF` and `HISTOGRAM` are the baselines that serve as the empirical upper and lower bounds of the fidelity for HP synthesizers. The low score indicates the synthesizer can generate high-quality synthetic data. The best result is in bold.

| | Obesity | Robot | Abalone | News | Insurance | Wine |
|---|---|---|---|---|---|---|
| MST | $0.062_{\pm.003}$ | $0.055_{\pm.003}$ | $0.062_{\pm.008}$ | $0.050_{\pm.004}$ | $0.083_{\pm.009}$ | $0.075_{\pm.002}$ |
| PrivSyn | $0.053_{\pm.005}$ | $0.054_{\pm.004}$ | $\mathbf{0.032}_{\pm\mathbf{.005}}$ | $\mathbf{0.018}_{\pm\mathbf{.001}}$ | $0.074_{\pm.006}$ | $0.022_{\pm.001}$ |
| TVAE | $0.059_{\pm.003}$ | $0.059_{\pm.007}$ | $0.046_{\pm.005}$ | $0.079_{\pm.001}$ | $0.118_{\pm.009}$ | $0.045_{\pm.001}$ |
| CTGAN | $0.085_{\pm.004}$ | $0.109_{\pm.009}$ | $0.066_{\pm.005}$ | $0.040_{\pm.001}$ | $0.116_{\pm.008}$ | $0.034_{\pm.001}$ |
| TabDDPM | $\mathbf{0.043}_{\pm\mathbf{.003}}$ | $\mathbf{0.028}_{\pm\mathbf{.004}}$ | $0.034_{\pm.010}$ | $0.032_{\pm.001}$ | $\mathbf{0.070}_{\pm\mathbf{.009}}$ | $0.017_{\pm.001}$ |
| REaLTabFormer | $0.062_{\pm.006}$ | $0.036_{\pm.005}$ | $0.040_{\pm.014}$ | $0.041_{\pm.001}$ | $0.071_{\pm.010}$ | $\mathbf{0.015}_{\pm\mathbf{.001}}$ |
| MST ($\epsilon = 1$) | $\mathbf{0.075}_{\pm\mathbf{.004}}$ | $\mathbf{0.072}_{\pm\mathbf{.007}}$ | $\mathbf{0.080}_{\pm\mathbf{.010}}$ | $0.051_{\pm.002}$ | $\mathbf{0.093}_{\pm\mathbf{.006}}$ | $\mathbf{0.075}_{\pm\mathbf{.001}}$ |
| PrivSyn ($\epsilon = 1$) | $0.154_{\pm.013}$ | $0.177_{\pm.011}$ | $0.111_{\pm.011}$ | $\mathbf{0.044}_{\pm\mathbf{.001}}$ | $0.152_{\pm.011}$ | $0.130_{\pm.005}$ |
| PATE-GAN ($\epsilon = 1$) | $0.089_{\pm.004}$ | $0.478_{\pm.007}$ | $0.353_{\pm.009}$ | $0.061_{\pm.002}$ | $0.386_{\pm.011}$ | $0.250_{\pm.003}$ |
| TableDiffusion ($\epsilon = 1$) | $0.338_{\pm.005}$ | $0.203_{\pm.002}$ | $0.226_{\pm.007}$ | $0.128_{\pm.001}$ | $0.366_{\pm.008}$ | $0.098_{\pm.001}$ |
| `HALF` (upper bound) | $0.041_{\pm.006}$ | $0.023_{\pm.005}$ | $0.028_{\pm.007}$ | $0.010_{\pm.002}$ | $0.060_{\pm.007}$ | $0.014_{\pm.001}$ |
| `HISTOGRAM` (lower bound) | $0.066_{\pm.002}$ | $0.065_{\pm.002}$ | $0.094_{\pm.009}$ | $0.059_{\pm.006}$ | $0.081_{\pm.004}$ | $0.076_{\pm.001}$ |

## C.2 PRIVACY REUSLTS

The complete privacy results in our evaluation are shown in Table 7 and Table 8.

Table 7: Privacy evaluation (lower score means better empirical privacy protection) of HP synthesizers on the first six datasets. `SELF` is the baseline that serves as the empirical lower bound of MDS (the upper bound of MDS is 0 by definition). The best result is in bold.

| | Adult | Shoppers | Phishing | Magic | Faults | Bean |
|---|---|---|---|---|---|---|
| MST | $\mathbf{0.031}_{\pm\mathbf{.001}}$ | $0.012_{\pm.002}$ | $0.038_{\pm.003}$ | $0.008_{\pm.001}$ | $0.030_{\pm.002}$ | $0.015_{\pm.003}$ |
| PrivSyn | $0.046_{\pm.002}$ | $\mathbf{0.005}_{\pm\mathbf{.001}}$ | $0.017_{\pm.003}$ | $\mathbf{0.005}_{\pm\mathbf{.002}}$ | $\mathbf{0.004}_{\pm\mathbf{.001}}$ | $\mathbf{0.006}_{\pm\mathbf{.003}}$ |
| TVAE | $0.192_{\pm.003}$ | $0.050_{\pm.002}$ | $\mathbf{0.016}_{\pm\mathbf{.001}}$ | $0.016_{\pm.005}$ | $0.037_{\pm.002}$ | $0.029_{\pm.001}$ |
| CTGAN | $0.131_{\pm.002}$ | $0.018_{\pm.003}$ | $0.125_{\pm.003}$ | $0.012_{\pm.003}$ | $0.011_{\pm.003}$ | $0.028_{\pm.001}$ |
| TabDDPM | $0.204_{\pm.001}$ | $0.019_{\pm.002}$ | $0.082_{\pm.003}$ | $0.015_{\pm.001}$ | $0.092_{\pm.002}$ | $0.020_{\pm.003}$ |
| REaLTabFormer | $0.234_{\pm.001}$ | $0.047_{\pm.002}$ | $0.084_{\pm.003}$ | $0.011_{\pm.002}$ | $0.090_{\pm.002}$ | $0.018_{\pm.002}$ |
| `SELF` (lower bound) | $0.733_{\pm.000}$ | $0.094_{\pm.000}$ | $0.125_{\pm.000}$ | $0.199_{\pm.000}$ | $0.209_{\pm.000}$ | $0.273_{\pm.000}$ |

## C.3 UTILITY RESULTS

Table 9 and Table 10 present the results of MLA and Table 11 and Table 12 presents the query error results for different synthesizers. Similar to the results of fidelity evaluation, TabDDPM demon-

Table 8: Privacy evaluation (lower score means better empirical privacy protection) of HP synthesizers on the last six datasets. SELF is the baseline that serves as the empirical lower bound of MDS (the upper bound of MDS is 0 by definition). The best result is in bold.

|  | Obesity | Robot | Abalone | News | Insurance | Wine |
|---|---|---|---|---|---|---|
| MST | $\mathbf{0.013}_{\pm.001}$ | $\mathbf{0.008}_{\pm.001}$ | $0.030_{\pm.002}$ | $0.043_{\pm.003}$ | $\mathbf{0.006}_{\pm.001}$ | $0.030_{\pm.002}$ |
| PrivSyn | $0.027_{\pm.002}$ | $0.012_{\pm.001}$ | $\mathbf{0.012}_{\pm.003}$ | $0.005_{\pm.002}$ | $0.013_{\pm.001}$ | $\mathbf{0.008}_{\pm.003}$ |
| TVAE | $0.104_{\pm.003}$ | $0.039_{\pm.002}$ | $0.035_{\pm.001}$ | $\mathbf{0.004}_{\pm.003}$ | $0.036_{\pm.002}$ | $0.019_{\pm.001}$ |
| CTGAN | $0.026_{\pm.001}$ | $0.033_{\pm.003}$ | $0.024_{\pm.002}$ | $0.007_{\pm.005}$ | $0.009_{\pm.003}$ | $0.013_{\pm.001}$ |
| TabDDPM | $0.333_{\pm.001}$ | $0.113_{\pm.002}$ | $0.120_{\pm.003}$ | $0.008_{\pm.001}$ | $0.027_{\pm.002}$ | $0.075_{\pm.003}$ |
| REaLTabFormer | $0.283_{\pm.002}$ | $0.038_{\pm.001}$ | $0.150_{\pm.002}$ | $0.008_{\pm.002}$ | $0.083_{\pm.001}$ | $0.034_{\pm.001}$ |
| SELF (lower bound) | $0.671_{\pm.000}$ | $0.338_{\pm.000}$ | $0.285_{\pm.000}$ | $0.068_{\pm.000}$ | $0.078_{\pm.000}$ | $0.346_{\pm.000}$ |

strates strong performance among HP synthesizers, while statistical methods outperform other approaches among DP synthesizers.

Table 9: Utility evaluation (*i.e.,* MLA) of data synthesis on the first six datasets. The lower value means better utility. The privacy budget $\epsilon$ of HP synthesizers is set as $\infty$ (the top part), and the budget for DP synthesizers is set as 1 (the bottom part). The best result of each category is in bold.

|  | Adult | Shoppers | Phishing | Magic | Faults | Bean |
|---|---|---|---|---|---|---|
| MST | $0.086_{\pm.001}$ | $0.193_{\pm.002}$ | $0.037_{\pm.003}$ | $0.073_{\pm.001}$ | $0.255_{\pm.002}$ | $0.035_{\pm.003}$ |
| PrivSyn | $0.120_{\pm.003}$ | $0.040_{\pm.001}$ | $0.057_{\pm.002}$ | $0.085_{\pm.003}$ | $0.532_{\pm.001}$ | $0.039_{\pm.002}$ |
| TVAE | $0.035_{\pm.002}$ | $0.011_{\pm.003}$ | $0.031_{\pm.001}$ | $0.075_{\pm.002}$ | $0.217_{\pm.003}$ | $0.059_{\pm.001}$ |
| CTGAN | $0.039_{\pm.003}$ | $0.031_{\pm.002}$ | $0.068_{\pm.001}$ | $0.154_{\pm.003}$ | $0.525_{\pm.002}$ | $0.103_{\pm.001}$ |
| TabDDPM | $0.014_{\pm.001}$ | $\mathbf{0.003}_{\pm.002}$ | $0.007_{\pm.003}$ | $\mathbf{0.007}_{\pm.001}$ | $\mathbf{0.085}_{\pm.002}$ | $\mathbf{0.003}_{\pm.003}$ |
| REaLTabFormer | $\mathbf{0.004}_{\pm.001}$ | $0.004_{\pm.002}$ | $\mathbf{0.006}_{\pm.002}$ | $0.014_{\pm.001}$ | $0.101_{\pm.003}$ | $0.006_{\pm.002}$ |
| MST ($\epsilon = 1$) | $\mathbf{0.101}_{\pm.003}$ | $\mathbf{0.048}_{\pm.001}$ | $\mathbf{0.041}_{\pm.002}$ | $\mathbf{0.093}_{\pm.003}$ | $\mathbf{0.489}_{\pm.001}$ | $\mathbf{0.054}_{\pm.002}$ |
| PrivSyn ($\epsilon = 1$) | $0.120_{\pm.002}$ | $0.177_{\pm.003}$ | $0.085_{\pm.001}$ | $0.217_{\pm.002}$ | $0.753_{\pm.003}$ | $0.466_{\pm.001}$ |
| PATE-GAN ($\epsilon = 1$) | $0.126_{\pm.001}$ | $0.135_{\pm.002}$ | $0.530_{\pm.003}$ | $0.394_{\pm.001}$ | $0.781_{\pm.002}$ | $0.781_{\pm.003}$ |
| TableDiffusion ($\epsilon = 1$) | $0.198_{\pm.002}$ | $0.135_{\pm.003}$ | $0.074_{\pm.001}$ | $0.133_{\pm.002}$ | $0.904_{\pm.003}$ | $0.981_{\pm.001}$ |

Table 10: Utility evaluation (*i.e.,* MLA) of data synthesis on the last six datasets. The lower value means better utility. The privacy budget $\epsilon$ of HP synthesizers is set as $\infty$ (the top part), and the budget for DP synthesizers is set as 1 (the bottom part). The best result of each category is in bold.

|  | Obesity | Robot | Abalone | News | Insurance | Wine |
|---|---|---|---|---|---|---|
| MST | $0.332_{\pm.001}$ | $0.146_{\pm.002}$ | $0.096_{\pm.003}$ | $0.498_{\pm.001}$ | $0.270_{\pm.002}$ | $0.347_{\pm.003}$ |
| PrivSyn | $0.604_{\pm.003}$ | $0.406_{\pm.001}$ | $0.210_{\pm.002}$ | $1.992_{\pm.003}$ | $0.518_{\pm.001}$ | $0.201_{\pm.002}$ |
| TVAE | $0.294_{\pm.002}$ | $0.128_{\pm.003}$ | $0.245_{\pm.001}$ | $0.147_{\pm.002}$ | $0.336_{\pm.003}$ | $0.091_{\pm.001}$ |
| CTGAN | $0.893_{\pm.003}$ | $0.434_{\pm.002}$ | $0.282_{\pm.001}$ | $0.104_{\pm.003}$ | $1.700_{\pm.002}$ | $0.222_{\pm.001}$ |
| TabDDPM | $\mathbf{0.021}_{\pm.001}$ | $\mathbf{0.011}_{\pm.002}$ | $0.043_{\pm.003}$ | $\mathbf{0.047}_{\pm.001}$ | $0.140_{\pm.002}$ | $0.047_{\pm.003}$ |
| REaLTabFormer | $0.054_{\pm.001}$ | $0.017_{\pm.002}$ | $\mathbf{0.020}_{\pm.002}$ | $0.047_{\pm.001}$ | $\mathbf{0.039}_{\pm.002}$ | $\mathbf{0.042}_{\pm.003}$ |
| MST ($\epsilon = 1$) | $\mathbf{0.531}_{\pm.003}$ | $\mathbf{0.245}_{\pm.001}$ | $\mathbf{0.241}_{\pm.002}$ | $1.072_{\pm.003}$ | $\mathbf{1.366}_{\pm.001}$ | $0.340_{\pm.002}$ |
| PrivSyn ($\epsilon = 1$) | $0.821_{\pm.002}$ | $0.608_{\pm.003}$ | $0.624_{\pm.001}$ | $4.538_{\pm.002}$ | $1.878_{\pm.003}$ | $\mathbf{0.302}_{\pm.001}$ |
| PATE-GAN ($\epsilon = 1$) | $0.877_{\pm.001}$ | $0.755_{\pm.002}$ | $2.119_{\pm.003}$ | $0.259_{\pm.001}$ | $2.325_{\pm.002}$ | $0.405_{\pm.003}$ |
| TableDiffusion ($\epsilon = 1$) | $0.968_{\pm.002}$ | $0.439_{\pm.003}$ | $0.287_{\pm.001}$ | $\mathbf{0.781}_{\pm.002}$ | $2.503_{\pm.003}$ | $0.489_{\pm.001}$ |

## C.4 COMPARISON OF DIFFERENT PRIVACY METRICS

**Comparsion with Syntactic Privacy Evaluation Metrics and MIAs.** We compare the efficacy of different privacy evaluation metrics by conducting a series of proof-of-concept experiments. Specifically, we consider the following popular metrics:

- **DCR** (Zhao et al., 2021) measures the distance between the synthetic record and its closest real neighbor. The 5th percentile of the distance distribution represents the privacy score (a higher score means better privacy). We also utilize the worst-case (nearest distance) of DCR for comparison.

Table 11: Utility evaluation (*i.e.,* query error) of data synthesis on the first six datasets. A lower value means a smaller query error. The privacy budget $\epsilon$ of HP synthesizers is $\infty$ (the top part), and the budget for DP synthesizers is 1 (the bottom part). The best result of each category is in bold.

| | Adult | Shoppers | Phishing | Magic | Faults | Bean |
|---|---|---|---|---|---|---|
| MST | $0.056_{\pm.018}$ | $0.044_{\pm.005}$ | $\mathbf{0.009_{\pm.001}}$ | $0.035_{\pm.004}$ | $0.041_{\pm.003}$ | $0.036_{\pm.003}$ |
| PrivSyn | $0.009_{\pm.002}$ | $0.011_{\pm.006}$ | $0.011_{\pm.002}$ | $0.011_{\pm.002}$ | $0.027_{\pm.004}$ | $0.034_{\pm.002}$ |
| TVAE | $0.025_{\pm.005}$ | $0.034_{\pm.006}$ | $0.018_{\pm.000}$ | $0.014_{\pm.002}$ | $0.026_{\pm.003}$ | $0.019_{\pm.001}$ |
| CTGAN | $0.015_{\pm.001}$ | $0.017_{\pm.001}$ | $0.051_{\pm.002}$ | $0.037_{\pm.002}$ | $0.047_{\pm.006}$ | $0.030_{\pm.003}$ |
| TabDDPM | $0.006_{\pm.001}$ | $0.008_{\pm.001}$ | $0.012_{\pm.001}$ | $\mathbf{0.006_{\pm.001}}$ | $\mathbf{0.021_{\pm.002}}$ | $\mathbf{0.006_{\pm.001}}$ |
| REaLTabFormer | $\mathbf{0.004_{\pm.001}}$ | $\mathbf{0.007_{\pm.001}}$ | $0.011_{\pm.003}$ | $0.012_{\pm.001}$ | $0.024_{\pm.002}$ | $\mathbf{0.006_{\pm.001}}$ |
| MST ($\epsilon = 1$) | $0.071_{\pm.014}$ | $0.052_{\pm.017}$ | $\mathbf{0.012_{\pm.001}}$ | $0.036_{\pm.003}$ | $\mathbf{0.045_{\pm.002}}$ | $\mathbf{0.037_{\pm.002}}$ |
| PrivSyn ($\epsilon = 1$) | $\mathbf{0.010_{\pm.001}}$ | $0.027_{\pm.007}$ | $0.016_{\pm.002}$ | $\mathbf{0.025_{\pm.003}}$ | $0.100_{\pm.006}$ | $0.048_{\pm.004}$ |
| PATE-GAN ($\epsilon = 1$) | $0.028_{\pm.004}$ | $\mathbf{0.024_{\pm.002}}$ | $0.117_{\pm.009}$ | $0.058_{\pm.005}$ | $0.088_{\pm.009}$ | $0.191_{\pm.017}$ |
| TableDiffusion ($\epsilon = 1$) | $0.057_{\pm.006}$ | $0.054_{\pm.005}$ | $0.071_{\pm.007}$ | $0.074_{\pm.011}$ | $0.119_{\pm.009}$ | $0.052_{\pm.007}$ |

Table 12: Utility evaluation (*i.e.,* query error) of data synthesis on the last six datasets. A lower value means a smaller query error. The privacy budget $\epsilon$ of HP synthesizers is $\infty$ (the top part), and the budget for DP synthesizers is 1 (the bottom part). The best result of each category is in bold.

| | Obesity | Robot | Abalone | News | Insurance | Wine |
|---|---|---|---|---|---|---|
| MST | $0.035_{\pm.007}$ | $0.049_{\pm.005}$ | $0.040_{\pm.004}$ | $0.033_{\pm.005}$ | $0.039_{\pm.004}$ | $0.042_{\pm.005}$ |
| PrivSyn | $0.027_{\pm.006}$ | $0.029_{\pm.003}$ | $0.014_{\pm.001}$ | $\mathbf{0.010_{\pm.005}}$ | $0.035_{\pm.007}$ | $0.013_{\pm.002}$ |
| TVAE | $0.027_{\pm.003}$ | $0.020_{\pm.001}$ | $0.016_{\pm.002}$ | $0.030_{\pm.006}$ | $0.050_{\pm.009}$ | $0.028_{\pm.004}$ |
| CTGAN | $0.037_{\pm.004}$ | $0.033_{\pm.004}$ | $0.036_{\pm.005}$ | $0.018_{\pm.003}$ | $0.055_{\pm.006}$ | $0.016_{\pm.003}$ |
| TabDDPM | $\mathbf{0.017_{\pm.003}}$ | $\mathbf{0.008_{\pm.001}}$ | $\mathbf{0.011_{\pm.003}}$ | $0.017_{\pm.002}$ | $\mathbf{0.027_{\pm.007}}$ | $0.010_{\pm.001}$ |
| REaLTabFormer | $0.027_{\pm.004}$ | $0.009_{\pm.001}$ | $0.015_{\pm.002}$ | $0.019_{\pm.002}$ | $0.032_{\pm.006}$ | $\mathbf{0.007_{\pm.001}}$ |
| MST ($\epsilon = 1$) | $0.043_{\pm.005}$ | $\mathbf{0.050_{\pm.008}}$ | $\mathbf{0.041_{\pm.003}}$ | $0.043_{\pm.004}$ | $\mathbf{0.033_{\pm.004}}$ | $\mathbf{0.045_{\pm.004}}$ |
| PrivSyn ($\epsilon = 1$) | $0.060_{\pm.004}$ | $0.095_{\pm.002}$ | $0.051_{\pm.003}$ | $\mathbf{0.027_{\pm.005}}$ | $0.062_{\pm.007}$ | $0.064_{\pm.008}$ |
| PATE-GAN ($\epsilon = 1$) | $\mathbf{0.037_{\pm.001}}$ | $0.150_{\pm.023}$ | $0.223_{\pm.032}$ | $0.029_{\pm.003}$ | $0.138_{\pm.011}$ | $0.158_{\pm.013}$ |
| TableDiffusion ($\epsilon = 1$) | $0.108_{\pm.010}$ | $0.071_{\pm.012}$ | $0.085_{\pm.010}$ | $0.050_{\pm.003}$ | $0.195_{\pm.011}$ | $0.048_{\pm.006}$ |

- **NNDR** (Zhao et al., 2021) calculates the distance ratio between the closest and second closest real neighbor to synthetic data. The 5th percentile (or nearest distance) determines the privacy score, where higher values indicate better privacy.
- **Groundhog** (Stadler et al., 2022) first calculates statistics (*e.g.,* histogram, correlations, *etc.*) from synthetic data as features. It then uses these features to train shadow models to form a binary classification for membership attack.
- **TAPAS** (Houssiau et al., 2022) leverages the counting queries as features and trains a random forest classifier for membership attack.
- **DOMIAS** (van Breugel et al., 2023) is the state-of-the-art MIA for data synthesis, which utilizes the additional reference dataset to calibrate the density estimation of output distributions, and determines the membership via likelihood ratio hypothesis.

We randomly divide the dataset into two disjoint subsets: a training set $D_t$ and a reference set $D_r$, where $|D_t| = |D_r|$ and they share the same data distribution. Each synthesis algorithm is trained on $D_t$ and generates the synthetic data $D$, while the reference data $D_r$ remains unused during the synthesis process. Different treatments are applied for different metrics: (i) For syntactic metrics (*i.e.,* DCR and NNDR), we compute the privacy score by treating either $D_t$ or $D_r$ as the real data. Unless a synthesizer provides very good privacy, it is expected that the privacy leakage on $D_r$ is significantly smaller than that on $D_t$, since the synthetic data is generated using $D_t$ and is independent of $D_r$. (ii) For MIAs and MDS, training dataset $D_t$ and synthesis algorithm A are utilized to compute the privacy leakage. Table 13 presents the scores of different privacy evaluation metrics for various HP synthesizers on the Adult dataset. We have the following observations:

- *Syntactic metrics are not stable.* The standard deviations of syntactic metrics are quite large compared to their mean values. This instability is pronounced when using the nearest distance as the score, representing the worst-case assessment. This arises because syntactic metrics fail to account for the inherent randomness of the synthesis process.

Table 13: Comparison of privacy evaluation metrics for HP synthesizers on Adult dataset. $D_t$, $D_r$, and $S$ are the training, reference, and synthetic data. A is the synthesis algorithm. Syntactic metrics (DCR and NNDR) are highly unstable and are unable to provide meaningful privacy measures. MIAs (Groundhog, TAPAS, and MODIAS) fail to distinguish the different levels of privacy risks of synthesizers.

| Privacy Evaluation Metric | Metric Input | MST ($\epsilon = \infty$) | PrivSyn ($\epsilon = \infty$) | TVAE | CTGAN | TabDDPM | GReaT |
|---|---|---|---|---|---|---|---|
| DCR (5th percentile distance) | $D_t, S$ | $0.535_{\pm.121}$ | $0.520_{\pm.182}$ | $0.493_{\pm.116}$ | $0.533_{\pm.103}$ | $0.409_{\pm.181}$ | $0.437_{\pm.122}$ |
| | $D_r, S$ | $0.527_{\pm.146}$ | $0.531_{\pm.194}$ | $0.487_{\pm.158}$ | $0.479_{\pm.146}$ | $0.446_{\pm.175}$ | $0.502_{\pm.201}$ |
| DCR (Nearest distance) | $D_t, S$ | $0.102_{\pm.078}$ | $0.110_{\pm.084}$ | $0.124_{\pm.109}$ | $0.105_{\pm.883}$ | $0.081_{\pm.077}$ | $0.082_{\pm.069}$ |
| | $D_r, S$ | $0.117_{\pm.096}$ | $0.104_{\pm.083}$ | $0.132_{\pm.105}$ | $0.129_{\pm.094}$ | $0.102_{\pm.080}$ | $0.094_{\pm.067}$ |
| NNDR (5th percentile distance) | $D_t, S$ | $0.753_{\pm.226}$ | $0.737_{\pm.204}$ | $0.740_{\pm.218}$ | $0.733_{\pm.135}$ | $0.834_{\pm.129}$ | $0.835_{\pm.105}$ |
| | $D_r, S$ | $0.750_{\pm.223}$ | $0.703_{\pm.205}$ | $0.714_{\pm.187}$ | $0.802_{\pm.103}$ | $0.881_{\pm.101}$ | $0.795_{\pm.117}$ |
| NNDR (Nearest distance) | $D_t, S$ | $0.532_{\pm.274}$ | $0.508_{\pm.315}$ | $0.496_{\pm.229}$ | $0.517_{\pm.284}$ | $0.542_{\pm.247}$ | $0.522_{\pm.203}$ |
| | $D_r, S$ | $0.530_{\pm.298}$ | $0.498_{\pm.304}$ | $0.504_{\pm.209}$ | $0.539_{\pm.263}$ | $0.547_{\pm.229}$ | $0.512_{\pm.255}$ |
| Groundhog (TPR@1%FPR) | $D_t, A$ | $0.010_{\pm.002}$ | $0.011_{\pm.001}$ | $0.010_{\pm.003}$ | $0.010_{\pm.002}$ | $0.015_{\pm.003}$ | $0.013_{\pm.002}$ |
| TAPAS (TPR@1%FPR) | $D_t, A$ | $0.012_{\pm.001}$ | $0.013_{\pm.001}$ | $0.011_{\pm.002}$ | $0.009_{\pm.001}$ | $0.030_{\pm.002}$ | $0.020_{\pm.001}$ |
| MODIAS (TPR@1%FPR) | $D_t, A$ | $0.011_{\pm.001}$ | $0.011_{\pm.001}$ | $0.010_{\pm.002}$ | $0.008_{\pm.001}$ | $0.035_{\pm.002}$ | $0.022_{\pm.001}$ |
| MDS (ours) | $D_t, A$ | $0.031_{\pm.001}$ | $0.046_{\pm.002}$ | $0.192_{\pm.003}$ | $0.131_{\pm.002}$ | $0.204_{\pm.001}$ | $0.199_{\pm.001}$ |

- *Syntactic metrics are improper privacy measurements.* When using training data $D_t$ or reference data $D_r$ as real data to compute DCR and NNDR, the score differences are very small compared to their standard deviations. We note that for a good privacy evaluation metric, only when a synthesizer provides a very strong privacy guarantee, would we expect the two scores to be very similar. Since it is impossible that all HP synthesizers can provide such a high level of strong privacy guarantee, we assert this is because these syntactic metrics do not provide a good measure of privacy.
- *MIAs fail to distinguish different levels of privacy.* Experimental results show that the performance of MI attacks is relatively low for most synthesizers. We attribute the failure to the inherent randomness of synthesizers and synthetic datasets, which make it difficult to capture reliable signals to determine the membership.
- *MDS is a reliable privacy evaluation metric.* It is observed that the variance of MDS is very small, indicating its robustness for assessing data synthesizers. Additionally, MDS can also detect subtle differences in privacy leakage across various HP synthesizers.

**Comparison with Meeus et al.** We also notice that Meeus et al. (2023) proposed a new approach to evaluate the empirical privacy risks of synthesizers. It first identifies vulnerable samples by examining their closeness and then conducts a shadow model-based membership inference attack for the vulnerable sample for evaluation. (We follow the original paper and use the most vulnerable 10 records in our experiments.) While this approach (we call it vMIA) does not align with the standard setting for membership inference attacks, it may serve as a viable tool for empirical privacy evaluation. Thus, we conduct the following experiments to compare the effectiveness of vMIA with our proposed MDS.

Specifically, we train two DP synthesizers (*i.e.,* MST and PATE-GAN) with varying levels of privacy protection by adjusting the privacy budget, and we measure the empirical privacy risk using vMIA and MDS. We follow Meeus et al. (2023) and use the area under the curve (AUC) as the evaluation metric. The results of both experiments are presented in Figure 10. We observe an improvement in attack performance for MST, whereas the performance of PATE-GAN remains relatively low (below 60% AUC) across all levels of privacy budgets. We attribute this to the design of vulnerability scores in vMIA where the extracted vulnerable samples are determined by their closeness within datasets, which is independent of the underlying synthesizers. Additionally, since deep generative models are not designed to model marginal distributions, using marginal queries as features may not provide reliable performance signals for membership inference. Furthermore, vMIA suffers from relatively high variance. In contrast, MDS reliably detects different privacy risks across both marginal-based methods and deep generative models.

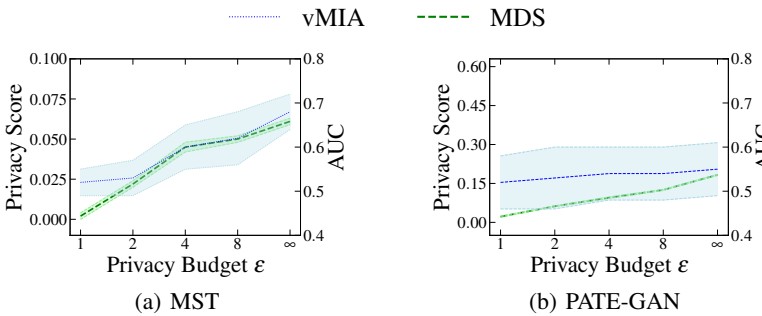

Figure 10: Privacy evaluation comparison between vMIA (Meeus et al., 2023) and MDS. MDS uses the left y-axis ("Privacy Score") whereas vMIA uses right y-axis ("AUC").

### C.5   IMPACT OF MODEL TUNING PHASE

To demonstrate the effectiveness of the proposed tuning objective, we conduct a series of comparative experiments. We leverage existing tuning approaches and evaluate the performance using both the proposed and existing evaluation metrics. Specifically, we consider the following tuning objectives and metrics:

- Existing Tuning Objectives. We note that many synthesizers (Zhang et al., 2021; McKenna et al., 2019; Zhang et al., 2024; Borisov et al., 2023) do not provide guidelines for hyperparameter tuning and some (Xu et al., 2019) are notoriously difficult to tune. However, a few synthesis algorithms, such as TabDDPM (Kotelnikov et al., 2023), describe a tuning process for their synthesizers. For comparison, we adopt the original tuning method of TabDDPM, which uses the machine learning efficiency of synthetic data on CatBoost as its tuning objective (we call it $MLE_{obj}$ for short).
- Existing Evaluation Metrics. We evaluated the results using five widely used fidelity metrics, including Total Variation Distance (TVD) and Kolmogorov-Smirnov Test (KST), Theil's uncertainty coefficient, Pearson correlation, and the correlation ratio. For existing utility metrics, we included machine learning efficiency on CatBoost and query errors. Note that we do not include the existing privacy metric (*i.e.,* DCR) because, as argued in our paper, it is flawed as a proper privacy metric and is unrelated to the existing tuning objectives. Detailed discussion about these metrics is shown in Appendix G.

**Comparsion with Existing Tuning Objective.**   We compare the performance improvements of the existing tuning objective (*i.e.,* $MLE_{obj}$) and the proposed method (*i.e.,* SynMeter) across various evaluation metrics on TabDDPM, as shown in Table 14. The results indicate that our proposed tuning objective significantly enhances performance on both the proposed and existing metrics. Additionally, while $MLE_{obj}$ effectively improves machine learning efficiency (which is also their optimization objective), it shows limited improvement in other aspects, such as all the fidelity metrics and query errors.

Table 14: Comparison the performance verage performance improvements (%) of existing tuning objective (*i.e.,* $MLE_{obj}$) and the proposed one (*i.e.,* SynMeter) with various evaluation metrics on TabDDPM. The best result is in bold.

| Tuning Objective | Fidelity ↑ | | | | | | Utility ↑ | | |
|---|---|---|---|---|---|---|---|---|---|
| | TVD | KST | Theil | Pearson | Correlation Ratio | Wasserstein (Ours) | Query Errors | MLE | MLA (Ours) |
| $MLE_{obj}$ | 2.45 | 1.52 | 2.26 | 2.47 | 2.61 | 2.18 | 2.63 | 10.58 | 7.34 |
| SynMeter | 10.15 | 14.83 | 11.46 | 12.47 | 13.83 | 13.62 | 11.95 | 13.06 | 13.67 |

**Impact of Different Coefficient Configurations.**  We also present the results of various coefficient combinations during the tuning phase, as shown in Table 15. The results demonstrate that our tuning objective is highly robust to different coefficient assignments, with all combinations showing a significant improvement over the default settings. Additionally, we note that practitioners can adjust these coefficients based on specific application needs to enhance certain characteristics of the

Table 15: Average performance improvements (%) on fidelity and utility for TabDDPM when training with the proposed tuning objective in Equation (11).

| $\alpha_1$ | $\alpha_2$ | $\alpha_3$ | Fidelity ↑ | | Utility ↑ | |
|---|---|---|---|---|---|---|
| | | | $D_{\text{Train}}$ | $D_{\text{Test}}$ | MLA | Query Error |
| 0 | 1/2 | 1/2 | 10.57 | 10.01 | 8.45 | 7.90 |
| 1/4 | 1/2 | 1/4 | 11.17 | 10.48 | 8.30 | 7.21 |
| 1/4 | 1/4 | 1/2 | 11.24 | 10.33 | 8.08 | 7.91 |
| 1/3 | 1/3 | 1/3 | 11.34 | 10.95 | 8.32 | 7.86 |
| 1/2 | 1/4 | 1/4 | 12.16 | 10.98 | 7.64 | 7.06 |
| 1/2 | 0 | 1/2 | 11.34 | 10.23 | 7.15 | 7.65 |
| 1/2 | 1/2 | 0 | 10.38 | 9.97 | 8.62 | 7.17 |

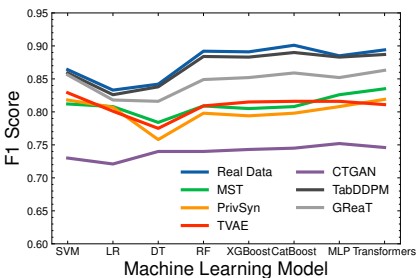

Figure 11: Performance behaviors of HP synthesizers on Magic dataset. "LR" denotes Linear Regression, "DT" is Decision Tree, and "RF" means Random Forest.

synthetic data. For example, one may want to increase $\alpha_2$ to improve the quality of synthetic data for model selection tasks. However, we also observed that no single coefficient configuration maximizes model performance across all three metrics. We believe this is because each metric emphasizes a different aspect of synthetic data quality. For instance, MLA is designed to maximize machine learning performance, specifically focusing on the correlation with label columns. In contrast, the fidelity metric evaluates the overall distributional similarity between real and synthetic data, which is independent of downstream tasks.

## C.6 PERFORMANCE OF TABSYN AND GREAT

Here we include TabSyn (Zhang et al., 2024) and GReaT (Borisov et al., 2023) for comparison. TabSyn first trains an autoencoder to capture inter-column relations and then employs a latent diffusion model for tabular data synthesis. GReaT leverage utilizes the large language model (LLM) for data synthesis. It converts records to textual representations for LLM and generates synthetic data with prompts. We compare it with TabDDPM (Kotelnikov et al., 2023), as TabDDPM has demonstrated impressive performance in our assessments.

All synthesizers are tuned using SynMeter and evaluated with our proposed metrics. As shown in Table 16 and Table 17, TabSyn and TabDDPM exhibit comparable performance across fidelity, privacy, and utility metrics, with neither emerging as a clear winner in any category. However, they all outperform GReaT on most fidelity and utility measures. It is worth noting that Zhang et al. (2024) reported superior performance for TabSyn over TabDDPM. We attribute this to the possibility that TabDDPM was not optimally tuned in previous evaluations.

Table 16: Performance comparison between TabDDPM, TabSyn, and GReaT on the first six datasets. The best result is in bold.

| | Synthesizer | Adult | Shoppers | Phishing | Magic | Faults | Bean |
|---|---|---|---|---|---|---|---|
| Fidelity ($D_{\text{train}}$) | TabDDPM | $\mathbf{0.020}_{\pm.001}$ | $\mathbf{0.022}_{\pm.001}$ | $\mathbf{0.015}_{\pm.001}$ | $\mathbf{0.011}_{\pm.003}$ | $\mathbf{0.026}_{\pm.002}$ | $\mathbf{0.015}_{\pm.002}$ |
| | TabSyn | $0.025_{\pm.003}$ | $0.030_{\pm.005}$ | $0.018_{\pm.003}$ | $0.012_{\pm.004}$ | $0.034_{\pm.003}$ | $0.033_{\pm.008}$ |
| | GReaT | $0.050_{\pm.002}$ | $0.049_{\pm.003}$ | $0.076_{\pm.002}$ | $0.037_{\pm.003}$ | $0.050_{\pm.006}$ | $0.020_{\pm.001}$ |
| Fidelity ($D_{\text{test}}$) | TabDDPM | $\mathbf{0.021}_{\pm.001}$ | $\mathbf{0.031}_{\pm.001}$ | $\mathbf{0.019}_{\pm.001}$ | $\mathbf{0.012}_{\pm.001}$ | $0.058_{\pm.008}$ | $\mathbf{0.016}_{\pm.003}$ |
| | TabSyn | $0.028_{\pm.002}$ | $0.035_{\pm.001}$ | $0.023_{\pm.001}$ | $0.014_{\pm.002}$ | $\mathbf{0.057}_{\pm.012}$ | $0.033_{\pm.007}$ |
| | GReaT | $0.052_{\pm.002}$ | $0.056_{\pm.004}$ | $0.072_{\pm.002}$ | $0.039_{\pm.003}$ | $0.063_{\pm.007}$ | $0.021_{\pm.004}$ |
| Privacy (MDS) | TabDDPM | $0.204_{\pm.001}$ | $0.019_{\pm.002}$ | $\mathbf{0.082}_{\pm.003}$ | $0.015_{\pm.001}$ | $\mathbf{0.092}_{\pm.002}$ | $0.020_{\pm.003}$ |
| | TabSyn | $0.202_{\pm.001}$ | $\mathbf{0.017}_{\pm.003}$ | $0.088_{\pm.002}$ | $0.029_{\pm.001}$ | $0.100_{\pm.003}$ | $0.021_{\pm.003}$ |
| | GReaT | $\mathbf{0.199}_{\pm.002}$ | $0.044_{\pm.003}$ | $0.091_{\pm.001}$ | $\mathbf{0.011}_{\pm.002}$ | $0.099_{\pm.003}$ | $\mathbf{0.016}_{\pm.004}$ |
| Utility (MLA) | TabDDPM | $0.014_{\pm.001}$ | $\mathbf{0.006}_{\pm.002}$ | $\mathbf{0.007}_{\pm.003}$ | $0.007_{\pm.001}$ | $\mathbf{0.085}_{\pm.002}$ | $\mathbf{0.003}_{\pm.003}$ |
| | TabSyn | $0.014_{\pm.001}$ | $\mathbf{0.006}_{\pm.002}$ | $0.025_{\pm.003}$ | $\mathbf{0.005}_{\pm.001}$ | $0.118_{\pm.002}$ | $0.005_{\pm.001}$ |
| | GReaT | $\mathbf{0.009}_{\pm.002}$ | $0.009_{\pm.003}$ | $0.020_{\pm.001}$ | $0.033_{\pm.002}$ | $0.183_{\pm.003}$ | $0.017_{\pm.001}$ |
| Utility (QueryError) | TabDDPM | $0.006_{\pm.001}$ | $\mathbf{0.008}_{\pm.001}$ | $\mathbf{0.012}_{\pm.001}$ | $\mathbf{0.006}_{\pm.001}$ | $0.021_{\pm.002}$ | $\mathbf{0.006}_{\pm.001}$ |
| | TabSyn | $\mathbf{0.005}_{\pm.001}$ | $0.009_{\pm.001}$ | $0.016_{\pm.001}$ | $0.007_{\pm.001}$ | $\mathbf{0.018}_{\pm.003}$ | $0.009_{\pm.001}$ |
| | GReaT | $0.014_{\pm.002}$ | $0.014_{\pm.004}$ | $0.049_{\pm.002}$ | $0.029_{\pm.003}$ | $0.028_{\pm.003}$ | $0.011_{\pm.001}$ |

Table 17: Performance comparison between TabDDPM, TabSyn, and GReaT on the last six datasets. The best result is in bold.

| | Synthesizer | Obesity | Robot | Abalone | News | Insurance | Wine |
|---|---|---|---|---|---|---|---|
| Fidelity ($D_{\text{train}}$) | TabDDPM | $0.017_{\pm.001}$ | $0.015_{\pm.002}$ | $0.015_{\pm.004}$ | $0.034_{\pm.001}$ | $0.028_{\pm.005}$ | $0.011_{\pm.000}$ |
| | TabSyn | $0.028_{\pm.003}$ | $0.045_{\pm.002}$ | $0.020_{\pm.005}$ | $0.012_{\pm.002}$ | $0.026_{\pm.003}$ | $0.021_{\pm.000}$ |
| | GReaT[1] | $0.055_{\pm.005}$ | $0.055_{\pm.003}$ | $0.022_{\pm.005}$ | - | $0.094_{\pm.004}$ | $0.019_{\pm.001}$ |
| Fidelity ($D_{\text{test}}$) | TabDDPM | $0.043_{\pm.003}$ | $0.028_{\pm.004}$ | $0.034_{\pm.010}$ | $0.032_{\pm.001}$ | $0.070_{\pm.009}$ | $0.017_{\pm.001}$ |
| | TabSyn | $0.047_{\pm.006}$ | $0.050_{\pm.004}$ | $0.020_{\pm.006}$ | $0.012_{\pm.001}$ | $0.067_{\pm.008}$ | $0.028_{\pm.000}$ |
| | GReaT | $0.062_{\pm.008}$ | $0.058_{\pm.006}$ | $0.037_{\pm.004}$ | - | $0.107_{\pm.010}$ | $0.024_{\pm.001}$ |
| Privacy (MDS) | TabDDPM | $0.333_{\pm.001}$ | $0.113_{\pm.002}$ | $0.120_{\pm.003}$ | $0.008_{\pm.001}$ | $0.027_{\pm.002}$ | $0.075_{\pm.003}$ |
| | TabSyn | $0.183_{\pm.002}$ | $0.062_{\pm.001}$ | $0.102_{\pm.002}$ | $0.026_{\pm.003}$ | $0.019_{\pm.002}$ | $0.124_{\pm.002}$ |
| | GReaT | $0.263_{\pm.002}$ | $0.039_{\pm.003}$ | $0.130_{\pm.001}$ | - | $0.072_{\pm.002}$ | $0.034_{\pm.003}$ |
| Utility (MLA) | TabDDPM | $0.021_{\pm.001}$ | $0.011_{\pm.002}$ | $0.043_{\pm.003}$ | $0.047_{\pm.001}$ | $0.140_{\pm.002}$ | $0.047_{\pm.003}$ |
| | TabSyn | $0.075_{\pm.001}$ | $0.086_{\pm.002}$ | $0.017_{\pm.001}$ | $0.009_{\pm.003}$ | $0.033_{\pm.001}$ | $0.082_{\pm.002}$ |
| | GReaT | $0.117_{\pm.002}$ | $0.050_{\pm.003}$ | $0.038_{\pm.001}$ | - | $0.292_{\pm.002}$ | $0.083_{\pm.003}$ |
| Utility (QueryError) | TabDDPM | $0.017_{\pm.003}$ | $0.008_{\pm.001}$ | $0.011_{\pm.003}$ | $0.017_{\pm.002}$ | $0.027_{\pm.007}$ | $0.010_{\pm.001}$ |
| | TabSyn | $0.020_{\pm.002}$ | $0.017_{\pm.003}$ | $0.009_{\pm.002}$ | $0.005_{\pm.001}$ | $0.027_{\pm.006}$ | $0.016_{\pm.001}$ |
| | GReaT | $0.030_{\pm.003}$ | $0.014_{\pm.001}$ | $0.019_{\pm.003}$ | - | $0.041_{\pm.007}$ | $0.013_{\pm.001}$ |

[1]GReaT cannot be applied to the News dataset because of the maximum length limit of large language models.

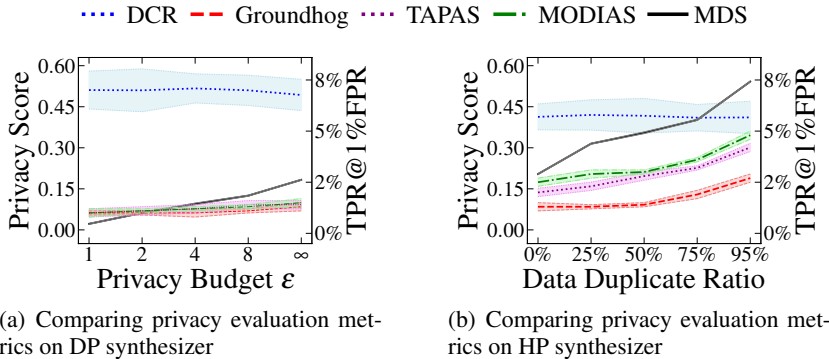

(a) Comparing privacy evaluation metrics on DP synthesizer

(b) Comparing privacy evaluation metrics on HP synthesizer

Figure 12: Effectiveness evaluation of MDS on Adult dataset. This figure is an enlarged version of Figure 2 presented in the main text.

## D   ENLARGED FIGURES

Due to the page limit, some figures in the main text may not be clear to all readers. Therefore, we have included enlarged versions of each figure from the main text, as shown in Figures 12 to Figure 15.

## E   HYPERPARAMETER SEARCH SPACES

In this paper, we evaluate the following synthesizers: MST (McKenna et al., 2019), PrivSyn (Zhang et al., 2021), TVAE (Xu et al., 2019), CTGAN (Xu et al., 2019), TabDDPM (Kotelnikov et al., 2023), REaLTabFormer (Solatorio & Dupriez, 2023), GReaT (Borisov et al., 2023), PATE-GAN (Jordon et al., 2018), and TableDiffusion (Truda, 2023). The hyperparameter search spaces of these synthesizers are shown in Table 18 to Table 26.

## F   DIFFERENTIALLY PRIVATE DATA SYNTHESIS

**Definition 6** (Differential Privacy (Dwork, 2006)). *A randomized mechanism $\mathcal{M} : D \rightarrow \mathcal{R}$ is $(\varepsilon, \delta)$-differentially private if for any two neighboring datasets $D, D' \in D$ and $S \subseteq \mathcal{R}$, it holds:*

$$\Pr[\mathcal{M}(D) \in S] \leq e^{\varepsilon} \Pr[\mathcal{M}(D') \in S] + \delta \tag{12}$$

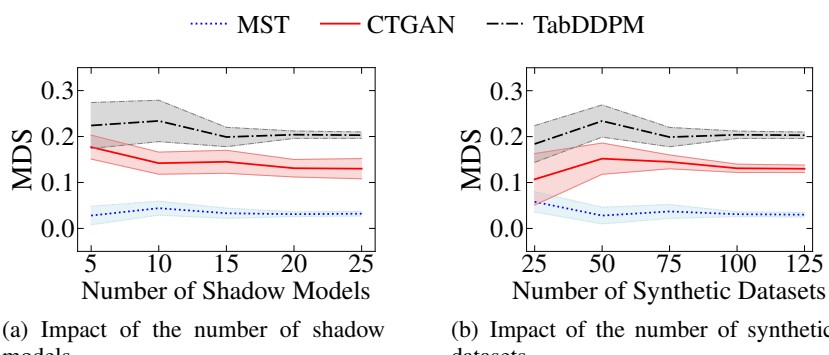

(a) Impact of the number of shadow models.

(b) Impact of the number of synthetic datasets.

Figure 13: Stability evaluation of MDS on Adult dataset. This figure is an enlarged version of Figure 3 presented in the main text.

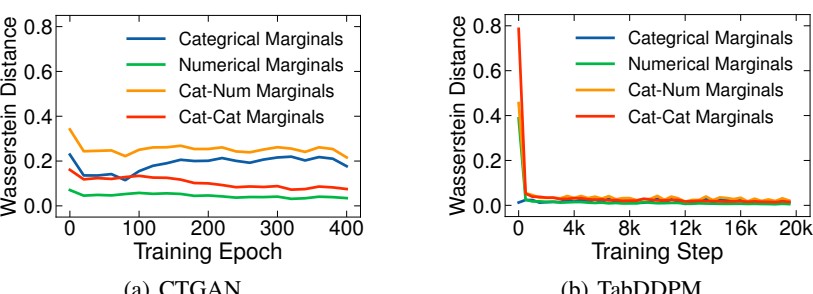

(a) CTGAN

(b) TabDDPM

Figure 14: Analyzing the learning process of CTGAN and TabDDPM with proposed fidelity metrics on the Bean dataset. This figure is an enlarged version of Figure 6 presented in the main text.

This definition requires that, on any two neighboring input databases, the difference in the output distributions of the randomized algorithm $\mathcal{M}$ is bounded by $e^\epsilon$ (*i.e.*, $\epsilon$ is the privacy budget), except with a small failure probability $\delta$. This failure probability $\delta$ is usually assumed to be cryptographically small: in this paper, it is set to $\delta = 1 \cdot 10^{-9}$.

An important property of DP is given by the *post-processing* theorem, which lets us use the output of DP mechanisms freely without worrying about further privacy leakage.

**Theorem 1** (Post-Processing). *Let $\mathcal{M} : D \to \mathcal{R}$ be an $(\varepsilon, \delta)$-DP mechanism and $f : \mathcal{R} \to \mathcal{R}'$. Then $f \circ \mathcal{M} : D \to \mathcal{R}'$ also satisfies $(\varepsilon, \delta)$-DP.*

Now we can use DP and the post-processing theorem to define differentially private data synthesis:

**Definition 7** (Differentially Privacy Data Synthesis). *Given a dataset $D$ sampled from some underlying distribution $\mathbb{D}$, we write $\mathsf{A} \leftarrow \mathcal{T}(D)$ to denote that the synthesizer $\mathsf{A}$ is learned by running the training algorithm $\mathcal{T}$ on the training set $D$. If the training algorithm $\mathcal{T}$ satisfies DP, then we call it differentially private data synthesis.*

That is, the probability that the adversary can infer if a given synthesizer $\mathsf{A}$ was fit on $D$ or $D'$, *i.e.*, $\mathsf{A} \sim \mathcal{T}(D)$ or $\mathsf{A} \sim \mathcal{T}(D')$, is bounded by the $\epsilon$ parameter. The guarantees of the overall data synthesis algorithm then follow from the post-processing theorem, as the synthetic dataset is simply sampled from the fitted synthesizer.

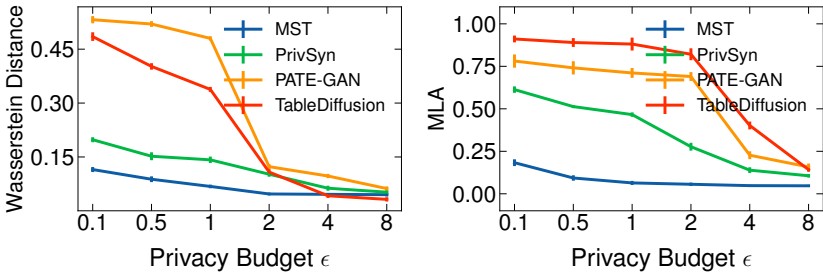

Figure 15: Impact of privacy budget $\epsilon$ on Bean dataset. This figure is an enlarged version of Figure 7 presented in the main text.

Table 18: MST (McKenna et al., 2019) hyperparameters search space.

| Parameter | Distribution |
|---|---|
| Number of two-way marginals | $\text{Int}[10, 50]$ |
| Number of three-way marginals | $\text{Int}[5, 20]$ |
| Number of bins | $\text{Int}[5, 20]$ |
| Maximum number of iterations | $\text{Int}[3000, 5000]$ |
| Number of tuning trials | 50 |

# G  DISCUSSION OF EXISTING EVALUATION METRICS

## G.1  EXISTING FIDELITY EVALUATION METRICS AND LIMITATIONS

**Low-order Statistics.**  Marginals are the workhorses of statistical data analysis and well-established statistics for one(two)-way marginals have been used to assess the quality of synthetic data.

*Distribution Measurements.* Total Variation Distance (TVD) and Kolmogorov-Smirnov Test (KST) are used to measure the univariate distribution similarity for categorical and numerical attributes, respectively. The main problem with this approach is the lack of versatility. Each type of marginal requires a distinct statistical measure, which complicates the ability to perform a comprehensive comparison across various attribute types.

*Correlation Statistics.* Some researchers use correlation difference, *i.e.,* the difference of correlation scores on synthetic and real data, to measure the pairwise distribution similarity. Popular correlation statistics like Theil's uncertainty coefficient (Zhao et al., 2021), Pearson correlation (Zhang et al., 2024), and the correlation ratio (Kotelnikov et al., 2023) are applied for different types of two-way marginals (categorical, continuous, and mixed). In addition to the lack of universality, this approach also suffers from the problem that correlation scores capture only limited information about the data distribution. Two attributes may have the same correlation score both in the real data and in the synthetic data, yet their underlying distributions diverge significantly—a phenomenon known as the scale invariance of correlation statistics[15].

**Likelihood Fitness.**  Xu et al. (2019) assume the input data are generated from some known probabilistic models (*e.g.,* Bayesian networks), thus the likelihood of synthetic data can be derived by fitting them to the priors. While likelihood fitness can naturally reflect the closeness of synthetic data to the assumed prior distribution, it is only feasible for data whose priors are known, which is inaccessible for most real-world complex datasets.

**Evaluator-dependent Metrics.**  Probabilistic mean squared error (pMSE) (Snoke et al., 2018) employs a logistic regression discriminator to distinguish between synthetic and real data, using relative prediction confidence as the fidelity metric. The effectiveness of pMSE highly relies on the choice of auxiliary discriminator, which requires careful calibration to ensure meaningful comparisons across

---

[15]https://en.wikipedia.org/wiki/Pearson_correlation_coefficient#Mathematical_properties

Table 19: PrivSyn (Zhang et al., 2021) hyperparameters search space.

| Parameter | Distribution |
|---|---|
| Number of bins | $\text{Int}[5, 20]$ |
| Maximum number of iterations | $\text{Int}[10, 100]$ |
| Number of tuning trials | 50 |

Table 20: TVAE (Xu et al., 2019) hyperparameters search space.

| Parameter | Distribution |
|---|---|
| Number of epochs | $\text{Int}[100, 500]$ |
| Batch size | $\text{Int}[500, 5000]$ |
| Loss factor | $\text{Float}[1, 5]$ |
| Embedding dimension | $\text{Int}[128, 512]$ |
| Compression dimension | $\text{Int}[128, 512]$ |
| Decompression dimension | $\text{Int}[128, 512]$ |
| $L_2$ regularization | $\text{LogUniform}[1e\text{-}6, 1e\text{-}3]$ |
| Number of tuning trials | 50 |

different datasets and synthesizers. Alaa et al. (2022) propose $\alpha$-Precision and $\beta$-Recall to quantify how faithful the synthetic data is. Specifically, $\alpha$-Precision defines fidelity as the proportion that the synthetic samples are covered by real data, and $\beta$-Recall evaluates the coverage of the synthetic data. However, previous studies (Zhang et al., 2024) find that $\alpha$-Precision and $\beta$-Recall exhibit a predominantly negative correlation, and it's unclear which one should be used for fidelity evaluation.

### G.2 EXISTING PRIVACY EVALUATION METRICS AND LIMITATIONS

**Syntactic Privacy Evaluation Metrics.** Researchers propose to measure the empirical privacy risk of synthetic data by comparing an input dataset with the output dataset generated by the synthesizer, typically using the distances between data records. For example, the Distance to Closest Records (DCR) (Zhao et al., 2021) metric looks at the distribution of the distances from each synthetic data point to its nearest real one, and uses the 5th percentile (or the mean) of this distribution as the privacy score. A small score is interpreted as indicating that the synthetic dataset is too close (similar) to real data, signaling a high risk of information leakage. There are other variations of DCR, *e.g.,* using the minimum distance instead of the 5th percentile, or using, for each record, the ratio of the closest distance and the second closest distance. However, these variations result in highly unstable measurements because of the inherent randomness of synthetic data. DCR and/or other similar metrics are widely used both in academia (Yale et al., 2019) and industry (AWS, 2022; Gretel, 2023), and have become the conventional privacy evaluation metrics for HP synthesizers (Jordon et al., 2021).

We note that metrics such as DCR are computed based on a pair of datasets: the input real dataset, and the output synthetic dataset. They do not depend on the synthesis algorithm at all. We call such metrics **syntactic**. We also note that when researchers were studying privacy properties of data anonymizers, syntactic privacy metrics such as $k$-anonymity (Sweeney, 2002), $\ell$-diversity (Machanavajjhala et al., 2006), and $t$-closeness (Li et al., 2007) were introduced. Similarly, these metrics consider only the anonymized dataset (and not the algorithm generating the dataset) when measuring privacy. Over the last decade and a half, the community gradually recognized the limitations of such syntactic privacy evaluation metrics and adopted privacy notions such as differential privacy (Dwork, 2006), which defines privacy as a property of the data processing algorithm, instead of the property of a particular output.

**Limitations of DCR.** We use the DCR as an example to show the limitations of such syntactic metrics as it's the most widely-used metric in the literature. First, DCR *overestimates* the privacy risks when data points are naturally clustered close together. As illustrated by discussions about dif-

Table 21: CTGAN (Xu et al., 2019) hyperparameters search space.

| Parameter | Distribution |
|---|---|
| Number of epochs | Int$[100, 500]$ |
| Batch size | Int$[500, 5000]$ |
| Embedding dimension | Int$[128, 512]$ |
| Generator dimension | Int$[128, 512]$ |
| Discriminator dimension | Int$[128, 512]$ |
| Learning rate of generator | LogUniform$[1e\text{-}5, 1e\text{-}3]$ |
| Learning rate of discriminator | LogUniform$[1e\text{-}5, 1e\text{-}3]$ |
| Number of tuning trials | 50 |

Table 22: TabDDPM (Kotelnikov et al., 2023) hyperparameters search space.

| Parameter | Distribution |
|---|---|
| Number of layers | Int$[2, 8]$ |
| Embedding dimension | Int$[128, 512]$ |
| Number of diffusion timesteps | Int$[100, 10000]$ |
| Number of training iterations | Int$[5000, 30000]$ |
| Learning rate of discriminator | LogUniform$[1e\text{-}5, 3e\text{-}3]$ |
| Number of tuning trials | 50 |

ferential privacy (Dwork & Roth, 2014; Li et al., 2016), leaking information regarding an individual should not be considered a privacy violation if the leakage can occur even if the individual's data is not used. Analogously, having some synthetic data very close to real ones does *not* mean worse privacy if this situation can occur even if each data point is removed. Consider, for example, a dataset that is a mixture of two Gaussians with small standard deviations. A good synthetic dataset is likely to follow the same distribution, and has many data points very close to the real ones. DCR interprets this closeness as a high privacy risk, overlooking the fact that the influence of any individual training instance on synthetic data is insignificant.

Second, DCR measures privacy loss using the 5th percentile (or mean) proximity to real data, which fails to bound the *worst-case* privacy leakage among all records. When measuring the privacy leakage across different individuals, one needs to ensure that the worst-case leakage is bounded, so that every individual's privacy is protected. It is unacceptable to use a mechanism that sacrifices the privacy of some individuals even though the protection averaged over the population is good. This point is illustrated by the fact that the re-identification of one or a few individuals is commonly accepted as privacy breaches (Li et al., 2013).

**Membership Inference Attack on Data Synthesis.** MIA has been widely used as an empirical privacy evaluation metric in machine learning, which has been extensively studied on discriminative models (Shokri et al., 2017; Carlini et al., 2022). For generative models like diffusion models (Duan et al., 2023) and LLM (Duan et al., 2024), studies mainly focus both on the *white-box* setting (where an adversary has full access to the trained model) and on the *black-box* setting (where an adversary has exact knowledge of the specifications of the generative model). In the realm of data synthesis, Annamalai et al. (2024) claim that the *non-box* setting should be considered in practice: the adversary has access to the synthetic dataset but no information about the underlying generative model or even the specifications of the synthetic data generation algorithm. Stadler et al. (2022) perform the first non-box membership inference attack called Groundhog, which utilizes handcrafted features extracted from synthetic data distribution to train shadow models. While the attack against a small minority of records can be useful to measure theoretical risks, they may not be necessarily relevant in practice especially if the adversary does not have a precise way to recognize vulnerable outliers. TAPAS (Houssiau et al., 2022) utilizes target counting queries as features and trains a random forest classifier to perform the attack and achieve better performance than Stadler et al. (2022). DOMIAS (van Breugel et al., 2023) utilizes the additional reference dataset to calibrate the density estimation of output distributions and achieve state-of-the-art performance for data synthesis. How-

Table 23: REaLTabFormer (Solatorio & Dupriez, 2023) hyperparameters search space.

| Parameter | Distribution |
|---|---|
| Number of epochs | $\text{Int}[100, 1000]$ |
| Batch size | $\text{Int}[8, 32]$ |
| Number of tuning trials | 20 |

Table 24: GReaT (Borisov et al., 2023) hyperparameters search space.

| Parameter | Distribution |
|---|---|
| Temperature | $\text{Float}[0.6, 0.9]$ |
| Number of fine-tuning epochs | $\text{Int}[100, 300]$ |
| Number of training iterations | $\text{Int}[5000, 30000]$ |
| Batch size | $\text{Int}[8, 32]$ |
| Number of tuning trials | 20 |

ever, as shown in Section 5.2, TAPAS and DOMIAS are still insufficient to distinguish nuances of privacy risks in all scenarios. As a result, current research on data synthesis rarely uses MIA for privacy evaluation (Qian et al., 2024).

### G.3 LIMITATIONS OF MACHINE LEARNING EFFICACY

To show the instability issue of machine learning efficacy, we compare the performance of different machine learning models on the Adult dataset, as illustrated in Figure 11. We can see the performance of various data synthesizers fluctuates significantly across different machine learning models, and such variations in performance underscore the impact of the choice of evaluation models. For instance, while PrivSyn is ranked third when evaluated using linear regression, it falls to fifth when assessed with decision trees. Such variations indicate that machine learning efficacy fails to provide a stable and consistent measure for evaluating the utility of synthetic data in prediction tasks. Moreover, directly averaging the performance across all models also fails to capture nuanced performance differences. For instance, the mean performance of PrivSyn and TVAE appears nearly identical (0.8 vs. 0.802), whereas MLA more effectively differentiates their relative performance degradation (0.085 vs. 0.075), providing a more reliable assessment.

## H DISCUSSION OF PROPOSED PRIVACY EVALUATION METRIC

**Comparison with Syntactic Privacy Evaluation Metrics.** We use DCR as an example to show how the proposed membership disclosure score (MDS) addresses the drawbacks of syntactic metrics. First, MDS addresses DCR's over-estimating leakage issue by quantifying how much including each record $x$ *changes* the distance between $x$ and the closest synthetic data. If including $x$ results in records much closer to $x$ to be generated, then the disclosure risk is high. Conversely, if records close to $x$ are generated regardless of whether $x$ is included, then the disclosure risk for $x$ is low. Therefore, MDS follows a distinguishing game designed to mirror the DP definition, rather than relying on the density of data points. Additionally, MDS uses the maximum disclosure risk among all records, providing a stable *worst-case* privacy measurement.

**Comparison with MIAs.** Both membership disclosure score (MDS) and membership inference attacks (MIAs) measure privacy risks by assessing the influence of discrepancies observed in the synthesizer when trained with or without certain records. Additionally, MDS incorporates shadow model techniques (Shokri et al., 2017) to estimate the influence for all data records, which is the standard approach in MIAs. However, unlike MIAs, MDS directly assesses the privacy risks of training data without relying on the construction of the membership inference security game (Carlini et al., 2022). Consequently, MDS's privacy estimation does not depend on the effectiveness of one specific attack algorithm, offering greater flexibility in evaluating various types of data synthesizers.

Table 25: PATE-GAN (Jordon et al., 2018) hyperparameters search space.

| Parameter | Distribution |
|---|---|
| Number of teachers | $\text{Int}[5, 20]$ |
| Number of generator layers | $\text{Int}[1, 3]$ |
| Number of discriminator layers | $\text{Int}[1, 3]$ |
| Generator dimension | $\text{Int}[50, 200]$ |
| Discriminator dimension | $\text{Int}[50, 200]$ |
| Number of iterations | $\text{Int}[1000, 5000]$ |
| Learning rate | $\text{LogUniform}[1e\text{-}5, 1e\text{-}3]$ |
| Number of tuning trials | 50 |

Table 26: TableDiffusion (Truda, 2023) hyperparameters search space.

| Parameter | Distribution |
|---|---|
| Number of layers | $\text{Int}[1, 6]$ |
| Number of diffusion timesteps | $\text{Int}[3, 20]$ |
| Number of epochs | $\text{Int}[5, 20]$ |
| Batch size | $\text{Int}[128, 1024]$ |
| Noise prediction | $\{\text{True, False}\}$ |
| Learning rate | $\text{LogUniform}[1e\text{-}4, 1e\text{-}2]$ |
| Number of tuning trials | 50 |

**Connections to Related Work.** The definition of the proposed MDS aligns closely with concepts of memorization in neural networks (Feldman, 2020; Zhang et al., 2023) and the leave-one-out notion of stability in machine learning (Bousquet & Elisseeff, 2002). However, it diverges in three crucial ways: (i) We measure the worst-case discourse risk as the privacy evaluation metric, whereas other studies focus on the difference of individuals or average cases. (ii) Our work specifically addresses privacy concerns in data synthesis, as opposed to other studies that explore discriminative models like classification. (iii) Our approach emphasizes the discrepancy caused by the presence or absence of target data in training, in contrast to other works that highlight performance gains from adding samples to the training set.

**Limitations of MDS.** Although MDS provides a straightforward way to assess the privacy risk of data synthesis, it may not apply to all synthesizers. Pathological synthesizers exist for which MDS is inappropriate. One such example is a synthesizer that maps all data points $x \in D$ to their opposites: $x \mapsto -x$. Suppose the nearest neighbor to $x$ in the real dataset is $x + \varepsilon$. In this case, MDS would be proportional to $|d(x, s(x)) - d(x, s(x + \varepsilon))|$, which can be tricked arbitrarily small with $\varepsilon$. However, this synthesizer completely reveals the dataset and MDS would suggest a false sense of privacy. Therefore, while we find MDS to be effective in assessing privacy risks for the synthesizers we tested, caution should be exercised when applying it in practice. For scenarios where privacy is paramount, we highly recommend using DP synthesizers instead of HP synthesizers.

We also note that MDS focuses specifically on membership privacy (Li et al., 2013) and does not address all potential privacy risks associated with synthetic datasets. For instance, attribute inference attacks (Annamalai et al., 2024) and reconstruction attacks (Jayaraman & Evans, 2022) pose serious privacy threats to synthetic data, which MDS is not designed to capture.

In addition, MDS requires training multiple shadow models to estimate the disclosure risks. This can pose a challenge when assessing large-scale tabular synthesis models like GReaT, which involve fine-tuning entire LLMs. However, existing MIAs (Stadler et al., 2022; van Breugel et al., 2023) also rely on shadow modeling to compute privacy scores. Thus, MDS remains a practical and feasible solution for privacy assessment in most tabular datasets and synthesis algorithms.

## I  MISLEADING STATEMENTS FROM PREVIOUS WORK

We find that some statements in the literature may be misleading or even incorrect due to limitations of evaluation metrics or methodologies. We highlight some of them below:

- Extensive studies (Zhao et al., 2021; Lee et al., 2023; Zhang et al., 2024) use Distance to Closest Records (DCR) to evaluate the privacy of synthetic data and assert their models are safe. However, in this paper, we show that DCR fails to serve as an adequate measure of privacy. We also show that many recently introduced HP methods exhibit significant privacy risks, which are often ignored by the community.
- Kotelnikov et al. (2023) show that the machine learning performance on TabDDPM is even better than that on real data, which implies that synthetic data can be a perfect (even better) substitute for real data. However, this statement may be incorrect due to inadequate model tuning and improper data shuffling practices. Our evaluations show that even simple models, such as linear regression, can achieve better performance on real data than on high-quality synthetic data.
- Some studies (Kim et al., 2022; Jordon et al., 2018) prioritize machine learning efficacy as the primary (if not only) fidelity evaluation metric. This approach is problematic because data synthesis can be *biased* to label attributes, and a high machine learning efficacy score does not necessarily equate to high fidelity in synthetic data.

