# OpenReview forum: "Systematic Assessment of Tabular Data Synthesis"
_ICLR.cc/2025/Conference — Submitted to ICLR 2025_

### Official Review · Reviewer_y2um · 2024-10-23

**Soundness:** 2
**Presentation:** 3
**Contribution:** 2
**Rating:** 6
**Confidence:** 3

**Summary:**

This paper studies the problem of evaluating tabular synthetic data generation techniques. To this end, it critiques existing metrics for fidelity, privacy, and utility, and proposes new ones. Then, it evaluates these metrics on a set of 12 datasets. The authors find that diffusion models generally outperform other model types in terms of utility and fidelity, whereas statistical methods are better suited for privacy. They also evaluate differentially-private synthesizers and evaluate the effect of DP budget.

**Strengths:**

This paper is interesting, I think these kinds of benchmarking studies can be very useful.

The paper tackles an important problem, there is a lot of interest in generating synthetic tabular data right now. I thought some of the metrics were interesting, particularly query error, which seems to address a common use case for synthetic data. I appreciated that the MDS score was taken as a maximum over x in D, in line with best practices in the membership inference literature.

The head-to-head comparison between models is very practically useful, and could have real-world impact.

**Weaknesses:**

I’m a little worried that the level of the technical contribution and the findings may not rise to the level of a top-tier conference. This is particularly true given that there exist other surveys on the quality of synthetic data generators.

1)	I don’t understand why the Wasserstein metric makes sense as a fidelity metric for categorical variables. You have defined the cost matrix as infinite between classes. So how can you find a meaningful transport map? This doesn’t seem like the right metric for categorical variables. By the way, Wasserstein distance has already been used as a fidelity metric for synthetic data over a metric space (e.g., CTAB-GAN+ (Zhao et al), DoppelGANger (Lin et al)), with total variation distance being used for categorial variables.

2)	The paper seems not to mention a number of related works, including:

-	On the Inadequacy of Similarity-based Privacy Metrics: Reconstruction Attacks against Truly Anonymous Synthetic Data (Ganev and De Cristofaro)
-	On the Quality of Synthetic Generated Tabular Data (Espinosa and Figueira)
-	A universal metric for the evaluation of synthetic tabular data (Chundawat et al)

For a survey paper, these omissions worried me--I’m concerned that there may be others I’m not aware of. I would definitely want to see a more in-depth literature review. How does your work relate to these, particularly the surveys on synthetic tabular data? What does your survey add to the discussion?

3)	The evaluation seems incomplete in terms of baselines. I thought there should at least be a tabular transformer (e.g., RealTabFormer or a successor) in the mix. GReaT is a transformer, but the tokenization is not really tailored to tabular data, and RealTabFormer is (a) more widely used in practice, and (b) not compared against in the GReaT paper. Also, the tables in the evaluation are illegible—it would be nice to make them bigger.

**Questions:**

1) How does your work relate to existing surveys on tabular synthetic data?

2) Why is Wasserstein distance an appropriate metric for categorical fields?

3) How do tabular transformers like RealTabFormer compare to the baselines you have evaluated?

---

> ### Author Response · Authors · 2024-11-20
>
> We thank the reviewer for their insightful feedback and address the raised questions below.
>
> **Q7.1: How does your work relate to existing surveys on tabular synthetic data?**
>
> We have added a new section (Section 6 in the revised paper) to provide a more systematic presentation of existing studies.
> We highlight the key difference between our work and the work mentioned by reviewers as follows:
>
> - [1] also critiqued existing privacy evaluation metrics (e.g., DCR), highlighting their vulnerability to reconstruction attacks.
> However, it did not thoroughly address the underlying flaws of these metrics nor propose new and effective privacy metrics. In contrast, we provide both experimental results and detailed analyses of the limitations of these syntactic privacy metrics and introduce a new metric, MDS, specifically designed to address these shortcomings.
> - Some previous evaluation papers [2,3] overlook privacy evaluation, which we consider critical for assessing HP synthesizers. As a result, their evaluations are less comprehensive and fail to provide deeper insights into the strengths and weaknesses of different synthesis algorithms.
> - [2] directly uses a range of existing fidelity metrics to evaluate each type of marginal distribution, while [3] combines these metrics into a single score for fidelity assessment. In contrast, we propose a Wasserstein-based fidelity metric that accommodates both categorical and numerical attributes for any $k$-way marginals. This approach provides a more unified, robust, and reliable method for fidelity evaluation.
>
> We kindly refer the reviewer to CQ1 in the common section for a detailed comparison between our work and more related literature.
>
> \
> [1] Ganev, Georgi, and Emiliano De Cristofaro. ``On the Inadequacy of Similarity-based Privacy Metrics: Reconstruction Attacks against'' Truly Anonymous Synthetic Data''." arXiv preprint arXiv:2312.05114 (2023).
>
> [2] Espinosa, Erica, and Alvaro Figueira. ``On the quality of synthetic generated tabular data.'' Mathematics 11, no. 15 (2023): 3278.
>
> [3] Lahoti, Mukund, and Pratik Narang. ``A Universal Metric for Robust Evaluation of Synthetic Tabular Data.'' (2024).
>
>
> **Q7.2: Why is Wasserstein distance an appropriate metric for categorical fields?**
>
> In the cost matrix of the Wasserstein distance, we assign an infinite cost to mismatches between different categories and a cost of 1 for matches within the same category. (We have reformulated the computation of the cost matrix in Equation 5 of the revised paper to make this clearer.) For one-way and two-way categorical attributes, this approach simplifies the sum of probability differences for each attribute, directly aligning with the calculations used in Total Variation Distance and Contingency Similarity [4].
>
> We agree with the reviewer that many synthesizers utilize the Wasserstein distance for optimization. In fact, we believe one possible reason diffusion models outperform other deep generative models is their ability to effectively minimize the Wasserstein distance [5].
> However, to the best of our knowledge, leveraging the Wasserstein distance as a unified fidelity evaluation metric for tabular data synthesis has been less explored.
> We kindly refer the reviewer to CQ2 in the common section for detailed explanations of the Wasserstein distance and to Q2.1 for a discussion of the novelty of this paper.
>
> [4] https://docs.sdv.dev/sdmetrics/metrics/metrics-glossary/contingencysimilarity
>
> [5] Dohyun Kwon, Ying Fan, and Kangwook Lee. ``Score-based generative modeling secretly minimizes the wasserstein distance''. In Proceedings of the 36th International Conference on Neural Information Processing Systems, 2022.
>
> **Q7.3: How do tabular transformers like RealTabFormer compare to the baselines you have evaluated?**
>
> We have added the REaLTabFormer paper as a reference and will include it in our benchmarks under our evaluation framework.
> We apologize for not having the results ready at this time, as we have received many valuable comments received from seven reviewers.
> However, we will make every effort to share the results before the rebuttal period closes and ensure that REaLTabFormer is included in the final version of our paper.
> Thank you for your patience and understanding.
>
>
> **Q7.4: The tables in the evaluation are illegible.**
>
> We have reorganized all the tables in the paper, splitting the detailed results (fidelity, privacy, and utility) into two tables.
> Each table now includes results for six datasets
> (instead of all 12 datasets) and has been moved to the Appendix. Additionally, we have enlarged all figures in the paper to improve readability.

---

> > ### Comment · Reviewer_y2um · 2024-11-23
> >
> > Dear authors, thank you for your updates. I have a few additional questions/comments:
> >
> > 1) In your answer about why Wasserstein distance makes sense for categorical variables, you argue that your choice of cost function makes the computation simplify to the same computation used for TV distance. So why not just call it (and compute) TV distance? That seems a lot more straightforward (to me) than trying to frame it as a Wasserstein distance under the indicator distance function. This is also an atypical usage of Wasserstein distance; the space of categorical variables is not a metric space, which is usually required in the definition of Wasserstein distance.
> >
> > 2) I didn't understand the organization of the new related work, and it is still missing some references. First, the choice of fidelity and utility metrics has nothing to do with whether a synthesizer is DP or not, so splitting this new section by DP vs HP doesn't seem like the right organization. Second, the discussion still doesn't seem to acknowledge that other papers have used Wasserstein distance and TV distance to measure fidelity for synthetic data, as I mentioned in my review. Third, there are statements in the related work that are not backed up by citations (eg "For example, Total Variation Distance (TVD) and the Kolmogorov-Smirnov Test (KST) are applied to assess univariate distribution similarity for categorical and numerical attributes, respectively." Finally, the related work reads like a laundry list of what people have done without really explaining much context, and how it relates to your work.
> >
> > 3) Thanks, I look forward to seeing the results from RealTabFormer.
> >
> > 4) The updated figures are still very difficult to read---the font size is tiny relative to the main text.

---

> > > ### Author Response · Authors · 2024-11-25
> > >
> > > We thank the reviewer for their insightful feedback and address the raised questions below.
> > >
> > > **Q7.5: Why not just use TV distance?**
> > >
> > > There are two main reasons why we chose Wasserstein distance over Total Variation (TV) distance:
> > >
> > > * Flexibilty.  While we have designed the cost function in our paper to align with TV distance, the Wasserstein distance allows for customization of the cost function, which can be critical for meaningful comparisons in practice. For example, in the NIST ``A Better Meter Stick for Differential Privacy'' competition [6], it is required to assess the quality of temporal or geographic attributes of synthetic data. By customizing the domain-specific cost functions, Wasserstein distance can incorporate the semantic meanings of categorical attributes (e.g., Los Angeles is closer to San Jose than Chicago) and provide a more meaningful evaluation, making Wasserstein distance more versatile than TV distance in real-world scenarios. We have briefly mentioned it in Section 3.1.
> > > * Generalization.  As elaborated in the paper,  Wasserstein distance can accommodate both categorical and numerical attributes and extends to any $k$-way marginals under the same criterion. This enables the evaluation of heterogeneous types of marginals, a capability beyond what TV distance offers.
> > >
> > > In summary, we believe the Wasserstein distance provides a more comprehensive and flexible measure for fidelity evaluation in tabular data synthesis.
> > >
> > > [6] https://www.herox.com/bettermeterstick/teams
> > >
> > >
> > > **Q7.6: The related work is not well organized.**
> > >
> > > We have reorganized the related work section in the revised paper to improve clarity and structure.
> > > Specifically, we have elaborated on three evaluation metrics—fidelity, privacy, and utility—and explained how these metrics are used to evaluate tabular data synthesis.
> > >
> > > We also have briefly highlighted their main limitations and referred to the corresponding sections for detailed discussions.
> > > Additionally, we have explicitly mentioned the use of Wasserstein distance and TV distance for one-way categorical marginals, supported by appropriate citations. Furthermore, we have discussed existing evaluation and benchmark studies, emphasizing how our work differs from them.
> > > We have also ensured that all the references you mentioned are properly cited.
> > > We hope the revised version provides clearer guidance on existing works and their relation to our work.
> > >
> > >
> > > **Q7.7: Performance of RealTabFormer.**
> > >
> > > We have implemented RealTabFormer in our evaluation framework, SynMeter [7], and included the results in our revised paper.
> > > Our findings show that RealTabFormer significantly outperforms GReaT and achieves comparable performance to TabDDPM.
> > >
> > > As a result, we have replaced GReaT with RealTabFormer as the representative LLM-based synthesizer in the main text (Figure 4, Table 3-10) and moved GReaT to Appendix C.6 for additional comparison.
> > > We also ranked the average performance of RealTabFormer across six types of synthesizers and summarized its rankings (higher is better) for each evaluation metric as follows:
> > >
> > > |               | Fidelity ($D_\text{train}$) | Fidelity ($D_\text{test}$) | Privacy (MDS) | Utility (MLA) | Utility (Query Error) |
> > > |---------------|-----------------------------|----------------------------|---------------|---------------|-----------------------|
> > > | TabDDPM       | 1.417                       | 1.500                      | 4.916         | 1.416         | 1.583                 |
> > > | REaLTabFormer | 1.583                       | 2.416                      | 4.818         | 1.500         | 2.000                 |
> > >
> > > From the table, we observe that RealTabFormer achieves performance very close to TabDDPM in terms of Fidelity and Utility.
> > > However, both synthesizers suffer from significant privacy leakage risks, with rankings of 4.8/4.9 out of 6 synthesizers on the proposed privacy metric.
> > > We believe that diffusion-based and LLM-based approaches represent promising directions for realistic tabular data synthesis. However, addressing their potential privacy risks will be crucial for practical use.
> > > We plan to include a more detailed discussion about RealTabFormer in the final version of our paper.
> > >
> > > Thank you again for highlighting this important baseline.
> > >
> > > [7] https://anonymous.4open.science/r/SynMeter

---

> > > > ### Author Response · Authors · 2024-11-25
> > > >
> > > > **Q7.8: Figures are too tiny to read.**
> > > >
> > > > Thanks for your feedback. We have made every effort to ensure that the text in all figures is readable:
> > > > * Increased Text Size. We have enlarged the text size by at least two font sizes for all figures.
> > > > * Font Adjustments. We have updated the fonts for Figures 4–5 and Figures 8–9 to ensure they are easier to read, with a font size similar to the main text.
> > > > * Enlarged Version in Appendix. We have included the enlarged versions of Figures 2-3, and Figures 6-7 in the Appendix as Figures 12-15 to further enhance readability.
> > > > * Optimized Figure Layout. We have reduced the spacing between figures, allowing for larger and more readable visuals.
> > > >
> > > > We hope these modifications significantly improve the clarity of our paper.
> > > > If any figure still appears too small or difficult to read for the reviewers, we would be grateful if you could point it out so we can address it further.

---

> > > > > ### Comment · Reviewer_y2um · 2024-11-25
> > > > >
> > > > > Thanks to the authors for your changes. They mostly address my concerns, and I would be inclined to upgrade my score, with the exception that I'm still concerned about the fact that Wasserstein distance is defined over metric spaces, which you do not have for categorical variables. This is a terminology issue, but I would like to see at least a discussion of this point in the main paper, after eq 5, that Wasserstein distance is typically defined for metric spaces, whereas categorical variables are not. I think it would also be important to state explicitly in that section that the metric you propose is equivalent to total variation distance, but you use the terminology Wasserstein distance (with the proposed cost matrix) for consistency throughout the paper. I understand why you did it, but it's a little strange to sum up distances using different cost matrices for categorical and numerical variables (other papers have done similar things, though, in fairness).
> > > > >
> > > > > One other question--which subsets of variables did you use in Figure 4 and Table 3 for instance? Did you compute the power set of attributes?
> > > > >
> > > > > In Figure 4, why does the Fidelity metric look high for TabDDPM and RealTabFormer, when lower is better for Wasserstein distance?
> > > > >
> > > > > The figures are much easier to read now, thank you for updating them.

---

> > > > > > ### Author Response · Authors · 2024-11-25
> > > > > >
> > > > > > We thank the reviewer for their valuable feedback and address the raised questions below.
> > > > > >
> > > > > > **Q7.9 Terminology Issue of Wasserstein Distance.**
> > > > > >
> > > > > > We have followed your advice and added a new paragraph in Section 3.1 (Page 3 of the revised paper) to address the terminology issues of Wasserstein distance for categorical attributes.
> > > > > > We also have explicitly mentioned that, under the defined cost functions, this approach is equivalent to total variation distance and contingency similarity, as used in previous studies.
> > > > > > We believe this added paragraph makes the definition easier for readers to understand.
> > > > > >
> > > > > >
> > > > > > **Q7.10 Implementation Details of Wasserstein Distance.**
> > > > > >
> > > > > > We compute the Wasserstein distance for all one-way and two-way marginals (categorical, numerical, and mixed) between real and synthetic data and use the mean as the final fidelity score.
> > > > > > The implementation details of all proposed metrics are provided in Appendix B.1. (We apologize for not including them in the main text due to space constraints.)
> > > > > > This approach aligns with existing fidelity metrics, which typically evaluate one-way and two-way marginals.
> > > > > >
> > > > > > For Figures 4 and 5, we note that these represent the averaged performance **rankings** for each evaluation metric among synthesizers, rather than the actual values of each metric. (This is the same with the REaLTabFormer table above).
> > > > > > We believe this is a more intuitive way to present the performance of different synthesizers across various evaluation metrics.
> > > > > > To avoid any misunderstanding, we have emphasized this information in the Figure 4 and Figure 5 captions by using bold text.
> > > > > >
> > > > > >
> > > > > > Thank you again for your valuable comments and for considering raising your scores; this means a lot to us. We are also happy to address any further questions the reviewers may have.

---

> > > > > > > ### Comment · Reviewer_y2um · 2024-11-26
> > > > > > >
> > > > > > > Thanks to the authors for addressing my comments (and for proactively dealing with the requests of many reviewers). I have updated my score to a 6.

---

### Official Review · Reviewer_SYTt · 2024-10-31

**Soundness:** 2
**Presentation:** 3
**Contribution:** 2
**Rating:** 3
**Confidence:** 4

**Summary:**

The paper proposes a new framework, called SynMeter, to assess (tabular) synthetic data generators. They focus on three dimensions:

- Fidelity:
	- Authors argue the need for a faithful and universal metric
	- They propose a Wasserstein distance-based metric to evaluate complex, high-dimensional tabular data distributions
- Privacy
	- Authors argue syntactic privacy scores to not be adequate
	- Authors argue that existing MIAs are ineffective, as they are not well understood and no MIA is effective against all synthesizers.
	- They propose a new metric called membership disclosure.
- Utility
	- Authors state that the traditionally used ML efficacy is not adequate, as they argue that there is no consensus on which evaluator should be used.
		○ They propose two new metrics: ML affinity and query error.

The paper then includes a holistic tuning objective as a combination of all metrics, to be used for hyperparameter selection.

Finally, the paper includes comprehensive experiments evaluating all metric across datasets and generators.

**Strengths:**

1. Comprehensively evaluating synthetic data generators is an important problem, and the paper provides a systematic, multi-dimensional evaluation framework to do so.
2. The paper includes considers many datasets and synthetic data generators, and comparing them across metrics is valuable for the research domain as a whole.
3. Proposes a way to pick hyperparameters across a multiple dimensions.
4. Authors make the framework publicly available as a tool for people generating synthetic data

**Weaknesses:**

While I understand the need for a holistic and widely agreed upon evaluation framework for synthetic data  generators, as a reader, I am not convinced that the metrics proposed by the authors are novel, or particularly better than previously proposed ones. I elaborate on each of the dimensions:

**1. Fidelity.**

While I find the notion of using Wasserstein distance to compute fidelity interesting, I remain to be convinced why this would be better than existing methods.
- Could you come up with an experimental setup and results which would compellingly show why the Wasserstein-based fidelity metric is strictly better than other deployed methods?

**2. Privacy.**

 I agree with the authors on the shortcomings of syntactic metrics, and like the example given for DCR. However:
- I do not follow the arguments made for why MIAs are not sufficient.
	- Why are MIAs against tabular data synthesis not well understood?
	- There indeed does not exist one MIA effective across all synthesizers, but this does not seem like a justification why MIAs are not useful? The ineffectiveness of the MIA might also just reflect limited privacy leakage?
- I do not understand what the difference is between the MDS metric and an MIA. If I understand it correctly, you are building a shadow model setup to then compute an MIA scoring function (which you then not evaluate as an MIA). You then pick the record for which you get the best distinction for this scoring function.  To me this basically comes down to compute MIA performance for all records, and use the highest MIA performance as the privacy metric.
	- How does this resolve your previously raised concerns regarding MIAs?
	- Moreover, with this, it is not clear whether this is the state-of-the-art MIA.
- Finally, in this entire discussion, I believe authors fail to mention (and implement) important related work. Houssiau et al [1] propose a new MIA which beats the one proposed by Stadler et al, and Meeus et al [2] propose a principled way to identify most at-risk records.

**3. Utility.**

I agree with the authors that there is no consensus in the literature on which metric should be used to evaluate the utility of the synthetic data. My thoughts:
	- While the exact formulation of the MLA score is, at least to my knowledge, new, I believe its novelty to be very limited. For instance, Stadler et al (in Sec. 6.3) measure utility as a decrease in ML accuracy of a model trained on real compared to a model trained on synthetic data. The only difference with the MLA metric would be the averaging across ML models and the normalization.
	- Similarly, the query error seems very similar to the k-way marginals fidelity approach, which has also been studied in for instance Annamalai et al. [3]
	- Could authors clarify why the query error should be part of the utility and not part of the fidelity evaluation?

**References**

[1] Houssiau, F., Jordon, J., Cohen, S. N., Daniel, O., Elliott, A., Geddes, J., ... & Szpruch, L. (2022). Tapas: a toolbox for adversarial privacy auditing of synthetic data. arXiv preprint arXiv:2211.06550.

[2] Meeus, M., Guepin, F., Creţu, A. M., & de Montjoye, Y. A. (2023, September). Achilles’ heels: vulnerable record identification in synthetic data publishing. In European Symposium on Research in Computer Security (pp. 380-399). Cham: Springer Nature Switzerland.

[3] Annamalai, M. S. M. S., Gadotti, A., & Rocher, L. (2024). A linear reconstruction approach for attribute inference attacks against synthetic data.

**Questions:**

(also see weaknesses)

- Could you come up with an experimental setup and results which would compellingly show why the Wasserstein-based fidelity metric is strictly better than other deployed methods?
- Why are MIAs against tabular data synthesis not well understood?
- There indeed does not exist one MIA effective across all synthesizers, but this does not seem like a justification why MIAs are not useful? The ineffectiveness of the MIA might also just reflect little privacy leakage?
- How does the MDS metric resolve your previously raised concerns regarding MIAs? To my understanding, you are in fact proposing a new MIA, but not evaluating it as such.
- Could you implement the MIA developed by Houssiau et al, and explain why the MDS metric is superior to compute the MIA performance for records identified by Meeus et al?
- Could authors clarify why the query error should be part of the utility and not part of the fidelity evaluation?

---

> ### Author Response · Authors · 2024-11-20
>
> We thank the reviewer for their insightful feedback and address the raised questions below.
>
> ***Question about the fidelity metric***
>
> **Q6.1: Could you come up with an experimental setup and results that would compellingly show why the Wasserstein-based fidelity metric is strictly better than other deployed methods?**
>
> Certainly! We have provided a simple example that demonstrates the superiority of the Wasserstein-based fidelity metric over commonly used correlation-based metrics.
> Additionally, we show how it generalizes existing metrics such as Total Variation Distance and Contingency Similarity.
> we kindly refer the reviewer to our response to CQ2 in the common section for detailed explanations and the experimental results.
>
> ***Question about the privacy metric.***
>
> **Q6.2: Why MIAs are not useful and not well understood?**
>
> We note that MIAs against classifiers have been extensively studied in recent years.
> We also note that using membership inference for privacy evaluation is well-established and our proposed method is also based on the principles.
> However, MIAs against tabular data synthesis remain relatively underexplored, with only a few MIA algorithms proposed for this domain.
> Furthermore, the effectiveness of MIA depends on the performance of specific MIA algorithms.
> Unfortunately, as shown in Section 5.2, current SOTA MIAs achieve near-random guessing performance (less than 2\%TPR@1\%FPR) for DP synthesizers across a wide range of privacy budgets, from $\epsilon=1$ to infinity (non-private one).
> Similar results have also been reported in related studies such as [1], which demonstrate that the AUC and ACC scores of all MIA algorithms are below 0.6 and are nearly identical across different synthesizers.
> Clearly, this cannot reflect the actual differences in privacy risks across varying privacy budgets and synthesizers.
> Given the above concerns, we believe that research on MIAs against tabular synthesis is still in an early stage and existing MIAs may not be able to provide a reliable measure of privacy risks across various synthesizers.
>
> However, we note the reviewer mentioned a new MIA setting by assessing only the most vulnerable samples.
> Although this setting is different from conventional MIA, it may result in stronger MIA performance for privacy evaluation. Please see Q6.5 for discussion.
>
> \
> [1] Houssiau, Florimond, James Jordon, Samuel N. Cohen, Owen Daniel, Andrew Elliott, James Geddes, Callum Mole, Camila Rangel-Smith, and Lukasz Szpruch. ``TAPAS: a Toolbox for Adversarial Privacy Auditing of Synthetic Data.'' In NeurIPS 2022 Workshop on Synthetic Data for Empowering ML Research.
>
> **Q6.3: Understanding of MDS.**
>
> We would like to emphasize that MDS is *not* a MIA.
> Rather than assessing privacy risks from the attackers' perspectives, MDS is an analytical framework that directly quantifies the disclosure risks of training data.
> Specifically, MDS utilizes shadow modeling techniques to estimate the disclosure risk of each training sample in a leave-one-out setting and selects the maximum risk as the privacy score.
> Therefore, it does not rely on the effectiveness of attacks for privacy assessment.
> This approach is more aligned with recent works on analyzing data memorization [2] and information leakage [3], offering a different lens for understanding the privacy risks of data synthesis.
>
>
> [2] Zhang, Chiyuan, Daphne Ippolito, Katherine Lee, Matthew Jagielski, Florian Tramèr, and Nicholas Carlini. ``Counterfactual memorization in neural language models.'' Advances in Neural Information Processing Systems 36 (2023): 39321-39362.
>
> [3] Ye, Jiayuan, Anastasia Borovykh, Soufiane Hayou, and Reza Shokri. ``Leave-one-out Distinguishability in Machine Learning.'' In The Twelfth International Conference on Learning Representations.

---

> > ### Comment · Reviewer_SYTt · 2024-11-29
> > **Understanding MDS**
> >
> > Thanks for clarifying the MDS metric, and linking it back to counterfactual memorization.
> >
> > I believe my issue lies in the fact that the metric still relies on one aspect of the synthetic dataset and the target record, i.e. closeness, hypothesizing that this captures all meaningful information to estimate the privacy risk of a record. For counterfactual memorization, prior work has used model loss in the case of ML models. In this case, it is well understood that ML model loss captures the privacy risk of a record relatively well. In contrast, for synthetic tabular data, I agree with the authors that MIAs are not equally well developed and I am not really convinced that closeness (as defined by a certain distance metric) captures privacy risk better than any other state-of-the-art MIA. Could authors elaborate why they believe closeness is the holistic metric to be used, better capturing any privacy risk than state-of-the-art MIAs (which might use other things than distance)?

---

> ### Author Response · Authors · 2024-11-20
>
> **Q6.4: Could you implement the MIA developed by Houssiau et al?**
>
> Thanks for pointing it out.
>
> TAPAS is an MIA toolbox developed by Houssiau et al. [1], which includes a variety of MIA variants tailored for tabular data synthesis.
> Following Meeus et al. [4], we selected the strongest MIA implemented in TAPAS for evaluation: it utilizes target counting queries as features and trains a random forest classifier to perform the attack.
> We have implemented and included the attack results in Figure 2 in the revised paper and we observe that TAPAS indeed outperforms Groundhog and achieves comparable performance to MODIAS.
> However, it is still relatively weak and fails to distinguish between different privacy levels for DP synthesizers.
>
> \
> [4] Meeus, Matthieu, Florent Guepin, Ana-Maria Creţu, and Yves-Alexandre de Montjoye. ``Achilles’ heels: vulnerable record identification in synthetic data publishing.'' In European Symposium on Research in Computer Security, pp. 380-399. Cham: Springer Nature Switzerland, 2023.
>
> **Q6.5: Why the MDS metric is superior to computing the MIA performance for records identified by Meeus et al?**
>
> We noticed that Meeus et al. [4] designed a new (different) MIA setting by first identifying the most vulnerable samples and then assessing the privacy risks with MIA on these samples.
> This setting is different from conventional MIA yet may be comparable to MDS for privacy evaluations.
> We apologize for not having these results ready at this time, as we have received many valuable comments received from seven reviewers.
> We will make every effort to implement and share the results before the rebuttal period closes.
> Thank you for your patience and understanding.
>
> ***Question about the utility metric.***
>
> **Q6.6: Could authors clarify why the query error should be part of the utility and not part of the fidelity evaluation?**
>
> Fidelity refers to the quality of data synthesis, measured by the distributional similarity between real and synthetic datasets, and is independent of downstream tasks.
> Query error, on the other hand, reflects the utility of synthetic data for point/range queries, which is a common task for data analysis.
> Therefore, we believe it is more reasonable to treat query error as a utility metric rather than as a fidelity measure.

---

> > ### Author Response · Authors · 2024-11-25
> > **Privacy Performance Comparsion with Meeus et al.**
> >
> > We noticed that Meeus et al. [4] designed a new (different) MIA setting by first identifying the most vulnerable samples and then assessing the privacy risks with MIA on these samples (we call it vMIA for short).
> > This setting is different from conventional MIA yet may be comparable to MDS for privacy evaluations.
> > In Appendix C.4 (Page 23), we compare the performance of vMIA with MDS on two DP synthesizers: MST and PATE-GAN.
> > Specifically, we train the synthesizers with different levels of privacy protection by adjusting the privacy budget and measure the empirical privacy risk using both approaches.
> > We follow Meeus et al. and use the area under the curve (AUC) as the evaluation metric.
> >
> >
> > The experimental results, shown in Figure 10, indicate that the average AUC score for vMIA does increase as $\epsilon$ increases.  The increasing trend is clear for MST, but less so for PATE-GAN.  However, the standard deviation is fairly high (which is also observed in the original paper), perhaps because only 10 target records are selected.
> > Such high variance means that one may be unable to tell whether two different scores are due to randomness or privacy levels.
> > In contrast, the MDS score exhibits significantly lower variance and demonstrates a clearer privacy detection trend for both MST and PATE-GAN, making it a more reliable metric for assessing privacy risks.

---

> > > ### Author Response · Authors · 2024-11-28
> > >
> > > As the interactive rebuttal window will close soon, we thank all reviewers again for all their helpful feedback. We believe we have addressed the reviewer's questions in our answer and are eager to continue the conversation in case of further questions.
> > > We also kindly request the reviewers to consider adjusting their score to reflect the improvements and clarifications made in response to their input.

---

> > > ### Comment · Reviewer_SYTt · 2024-11-29
> > > **Comparison with vMIA**
> > >
> > > Many thanks for running this comparison. Could you elaborate on how the you compute the variance of both vMIA and MDS in Figure 10? And could you adjust the y-axis in Figure 10b so we see the trend more clearly? I'm trying to understand any additional insights the MDS metric might offer. Many thanks.

---

> > > > ### Author Response · Authors · 2024-12-01
> > > >
> > > > As the interactive rebuttal window is about to close, we sincerely thank the reviewer again for their valuable and constructive feedback. We are happy to continue the conversation should there be any further questions regarding MDS and vMIA. Additionally, we kindly request the reviewer to consider adjusting their scores to reflect the improvements and clarifications made in response to their suggestions. Thank you again for helping us enhance the quality of our paper.

---

> > > > > ### Comment · Reviewer_SYTt · 2024-12-01
> > > > > **Answer to rebuttal**
> > > > >
> > > > > Many thanks for the clarifications. While I appreciate how the paper examines different aspects of synthetic data generation, I remain unconvinced that the proposed metrics (for fidelity, privacy and utility) are superior to the ones considered by prior work. Therefore I will maintain my current score.

---

> ### Author Response · Authors · 2024-11-29
>
> We thank the reviewer for their insightful feedback and address the raised questions below.
>
> **Q6.7: Why closeness is used for MDS.**
>
> First, we want to clarify that MDS does not capture *all* meaningful information, and we do not intend to make such a claim.
> We aim to propose a metric that is better than what is currently used; MDS is by no means perfect, and we discuss its limitations in Section 3.2 and Appendix H in the revised paper.
>
> We now discuss why we choose to use distances between real records and synthetic datasets in MDS:
>
> * For many generative models, the distance between a target record $x$ and its closest synthetic data record is related to the probability density of $x$ (which is related to the loss of the model on $x$). Usually, the density is smooth. Thus when $x$ has higher density, it is more likely that data records closer to $x$ are generated. Using distance instead of density in MDS has the advantage that it does not require a way to compute the density of a given data record, which is difficult for some synthesizers (e.g., GAN).  Instead, MDS requires only the synthesizers to output synthetic datasets (similar to the no-box setting in MIAs [1]), which all synthesizers must do.
> * The distance between real and synthetic data points is used in DCR [2], a widely adopted privacy evaluation metric for tabular data synthesis. DCR has been employed in many (if not all) SOTA HP synthesizers [3–5].  We think one reason that DCR and other syntactic privacy metrics are so popular is their intuitive nature: they quantify the similarity between synthetic datasets and real datasets as privacy risks. Therefore, when we design MDS, we aim to come up with something conceptually similar (namely using distance to the closest data point), yet more aligned with the spirit of MIAs to avoid the pitfalls of DCR.
>  * Empirical results on both HP and DP synthesizers demonstrate that MDS outperforms both DCR and existing MIAs in effectively capturing privacy risks. We thus believe that MDS represents an advancement in the state of the art for privacy evaluation metrics in tabular data synthesis.
>
> We hope the above discussion provides the reviewer with a better understanding of our choice for MDS, and we will include a more detailed discussion on the use of closeness in the final version of the paper.
>
>
>
> [1] Houssiau, Florimond, James Jordon, Samuel N. Cohen, Owen Daniel, Andrew Elliott, James Geddes, Callum Mole, Camila Rangel-Smith, and Lukasz Szpruch. ``TAPAS: a Toolbox for Adversarial Privacy Auditing of Synthetic Data.'' In NeurIPS 2022 Workshop on Synthetic Data for Empowering ML Research.
>
>
> [2] Zhao, Zilong, Aditya Kunar, Robert Birke, and Lydia Y. Chen. ``Ctab-gan: Effective table data synthesizing.'' In Asian Conference on Machine Learning, pp. 97-112. PMLR, 2021.
>
> [3] Hengrui Zhang, Jiani Zhang, Balasubramaniam Srinivasan, Zhengyuan Shen, Xiao Qin, Christos Faloutsos, Huzefa Rangwala, and George Karypis. ``Mixed-type tabular data synthesis with score-based diffusion in latent space''. In International Conference on Learning Representations, 2024.
>
> [4] Vadim Borisov, Kathrin Sessler, Tobias Leemann, Martin Pawelczyk, and Gjergji Kasneci. ``Language models are realistic tabular data generators''. In International Conference on Learning Representations, 2023.
>
> [5] Kotelnikov, Akim, Dmitry Baranchuk, Ivan Rubachev, and Artem Babenko. ``Tabddpm: Modelling tabular data with diffusion models.'' In International Conference on Machine Learning, pp. 17564-17579. PMLR, 2023.

---

> ### Author Response · Authors · 2024-11-29
>
> **Q6.8: Details about vMIA and MDS.**
>
> We first detail the computation of vMIA and MDS as follows:
>
> * **vMIA.** We first identify the most vulnerable 10 samples via the vulnerability score defined by Meeus et al. For each trial, we perform a query-based attack from Houssiau et al [1] on these selected samples (as detailed in Section 5.1 of Meeus et al.) and record the AUC score. We perform the attack 20 times and report the mean and standard deviation as the final score.
> * **MDS.** For each trial, we train 20 shadow models and generate 100 synthetic datasets per model to estimate the expected disclosure scores for each sample. The highest disclosure score among all samples in the dataset is selected as the privacy score for that trial. This procedure is repeated 20 times, and we report the mean and standard deviation as the final score.
>
>
> Since the paper revision period has ended and we can no longer make adjustments to the paper, we present the results for Figure 10(b) in the following table:
>
> | $\epsilon$ | 1                                               | 2                                               | 4                                               | 8                                               | $\infty$                                        |
> |------------|-------------------------------------------------|-------------------------------------------------|-------------------------------------------------|-------------------------------------------------|-------------------------------------------------|
> | vMIA       | $0.52\scriptscriptstyle \pm \scriptstyle .06$   | $0.53\scriptscriptstyle \pm \scriptstyle .07$   | $0.54\scriptscriptstyle \pm \scriptstyle .08$   | $0.54\scriptscriptstyle \pm \scriptstyle .06$   | $0.55 \scriptscriptstyle \pm \scriptstyle .07$  |
> | MDS        | $0.022\scriptscriptstyle \pm \scriptstyle .002$ | $0.062\scriptscriptstyle \pm \scriptstyle .002$ | $0.095\scriptscriptstyle \pm \scriptstyle .003$ | $0.125\scriptscriptstyle \pm \scriptstyle .003$ | $0.183\scriptscriptstyle \pm \scriptstyle .002$ |
>
>
>
>
> We are not sure about the exact reasons for the relatively large variance of vMIA, but conjecture the following two possible reasons:
>
> * vMIA evaluates MIA performance on only a small subset of samples (10 in our experiments and the original paper). This may result in higher variance, as the AUC score can fluctuate significantly with changes in the prediction of even a single sample.
> * In vMIA, the query-based attack [1] collects training samples using pairs of synthetic and real datasets as features and labels, fitting them into a classifier for the attack. Specifically, for each selected real dataset, a shadow model is trained, and one synthetic dataset is generated from the shadow model to form the training sample [6]. This approach may be inefficient in capturing the inherent randomness of synthesizers during generation. In contrast, MDS generates 100 synthetic datasets per shadow model to estimate the expected disclosure scores for each sample, providing a more robust and reliable evaluation of privacy risks.
>
> We hope the above results and discussion can provide a better understanding of the results of vMIA and MDS.
>
> [6] https://tapas-privacy.readthedocs.io/en/latest/quickstart.html

---

### Official Review · Reviewer_QCck · 2024-11-04

**Soundness:** 3
**Presentation:** 3
**Contribution:** 3
**Rating:** 6
**Confidence:** 3

**Summary:**

This paper critically examines the limitations of existing evaluation metrics and introduces new metrics for fidelity, privacy, and utility, establishing a comprehensive framework for assessing tabular data synthesis. Additionally, it proposes an integrated tuning objective that consistently optimizes data quality across different synthesizers. The study demonstrates that recent advancements in generative models significantly enhance tabular data synthesis performance while also highlighting key challenges, such as privacy risks and performance disparities among synthesizers.

**Strengths:**

1. The paper effectively identifies and addresses the limitations of existing metrics for fidelity, privacy, and utility, highlighting the need for the proposed metrics.
2. By introducing fidelity, privacy, and utility as core evaluation dimensions, the paper offers a well-rounded framework for assessing synthetic data quality.
3. The proposed metrics are thoroughly validated through extensive experiments on a large number of datasets, demonstrating their robustness and applicability in various contexts.

**Weaknesses:**

1. There is a lack of experimental comparison between the proposed fidelity and utility metrics and existing metrics. For example, including case studies where the proposed metrics and existing ones yield different evaluations on the same model could have strengthened the paper's claim of improved fidelity and utility assessments.
2. While the proposed privacy metric is an innovative approach, its reliance on numerous shadow models and synthetic datasets could lead to high computational costs. This complexity might render the metric impractical for large datasets, as the evaluation process could require substantial time and resources, limiting its usability in real-world applications.

**Questions:**

Since the tuning objective includes the same metrics used for evaluation, isn’t the observed performance improvement in Table 1 simply an expected result? Can this performance improvement truly be considered a reflection of the tuning objective’s effectiveness?

---

> ### Author Response · Authors · 2024-11-20
>
> We thank the reviewer for their insightful feedback and address the raised questions below.
>
> **Q5.1: Experimental comparison between the proposed fidelity and utility metrics and existing metrics.**
>
> For the proposed fidelity metric, we provide a simple example illustrating its superiority and generalization over commonly used metrics.
> We kindly refer the reviewer to our response to CQ2 in the common section for detailed explanations.
>
> For the proposed utility metric (i.e., MLA), we demonstrate in Appendix F.3 of the revised paper that the performance of various data synthesizers fluctuates significantly across different machine learning models.
> Directly using a single machine-learning model for evaluation or averaging performance across models fails to capture the nuanced performance degradation caused by the distribution shift in synthetic data.
> In contrast, MLA effectively addresses this issue by measuring the relative performance gap across all evaluated models, providing a more robust and accurate utility assessment.
>
>
> **Q5.2: High computational cost of proposed privacy metric MDS.**
>
> We agree with the reviewer that computing MDS involves training multiple shadow models, which can be computationally expensive.
> However, as discussed in Section 5.2, tabular datasets are typically much smaller compared to image or NLP datasets, and tabular synthesizers are relatively lightweight, making their training process rather fast: most synthesizers (e.g., PrivSyn, MST, CTGAN, TabSyn) can be trained in just a few minutes, with sampling taking only a few seconds.
> Additionally, existing SOTA MIAs [1] for tabular data also rely on shadow modeling to compute privacy scores.
> Therefore, we believe MDS remains a practical solution for privacy assessment in most tabular datasets and synthesis algorithms.
> Nevertheless, we have acknowledged the efficiency concerns as a limitation of MDS and included this discussion in Appendix G.
>
> \
> [1] Stadler, Theresa, Bristena Oprisanu, and Carmela Troncoso. ``Synthetic data–anonymisation groundhog day.'' In 31st USENIX Security Symposium (USENIX Security 22), pp. 1451-1468. 2022.
>
>
> **Q5.3: Can the performance improvement truly be considerered a reflection of the tuning objective’s effectiveness?**
>
> In Appendix C.5, we explored the effectiveness of different coefficient combinations, and the results demonstrate that even when one metric (e.g., fidelity) is excluded from the tuning process (e.g., the coefficient $\alpha_1 = 0$), the fidelity of the optimized synthesizer still shows noticeable improvement.
> This indicates that tuning with this objective can indeed enhance the overall quality of the synthetic data and validate the effectiveness of our tuning objective.

---

> > ### Comment · Reviewer_QCck · 2024-11-25
> >
> > Thank you for addressing my concerns. Your answers to Q5.1 and Q5.2 were sufficient and effectively resolved my questions. However, I believe Q5.3 has not been fully addressed.
> >
> > Let me re-iterate my question regarding the tuning objective. Tuning with the proposed method and evaluating it with the proposed metrics is **naturally expected to yield favorable results**, as this aligns directly with the optimization process. To validate the generality and robustness of the proposed method, it would be beneficial to include the following comparisons:
> > 1) Tuning with existing methods: Train the synthesizer using existing tuning objectives and evaluate the resulting model with both existing metrics and proposed metrics.
> > 2) Tuning with the proposed method: Train the synthesizer using the proposed tuning objective and evaluate the resulting model with both existing metrics and proposed metrics.
> >
> > This comparison would provide a clearer understanding of the effectiveness of the proposed tuning objective and its performance relative to existing methods across different evaluation metrics.

---

> > > ### Author Response · Authors · 2024-11-25
> > > **Effectivness of Proposed Tuning Objective**
> > >
> > > Thank you for your detailed suggestions. Following your advice, we have conducted a series of experiments to demonstrate the effectiveness of the proposed tuning objective. We first describe the existing tuning objectives and metrics used for experiments.
> > >
> > > * Existing Tuning Objectives. We note that many synthesizers [1-6] do not provide guidelines for hyperparameter tuning (that's also our motivation to develop a unified tuning objective) and some are notoriously difficult to tune [6].
> > > In addition, to our best knowledge, no benchmark/evaluation studies have mentioned the importance of the tuning phase for tabular data synthesis. However, a few synthesis algorithms, such as TabDDPM [7], do describe a tuning process for their synthesizers. Therefore, we adopt the tuning approach of TabDDPM, which uses the machine learning efficiency of synthetic data on CatBoost as its tuning objective (we call it MLE$_{\text{obj}}$ for short).
> > > * Existing Evaluation Metrics. We evaluated the results using a wide range of existing fidelity metrics, including Total Variation Distance [8], Kolmogorov-Smirnov Test (KST) [5], Theil's uncertainty coefficient [8], Pearson correlation [5], and the correlation ratio [7]. For utility metrics, we included machine learning efficiency (MLE) on CatBoost [1,7] and query errors [3-4]. Note that we do not include the existing privacy metric (i.e., DCR [8]) because, as argued in our paper, it is flawed as a proper privacy evaluation metric and is also not involved in the existing tuning objectives. A detailed discussion of these metrics is in Appendix G of the revised paper.
> > >
> > > We compare the performance improvements of the existing tuning objective (i.e., MLE$_{\text{obj}}$) and the proposed approach (i.e.,SynMeter) across various evaluation metrics on TabDDPM, as shown in the following two tables.
> > >
> > > | Fidelity Improv (%) | TVD      | KST   | Theil | Pearson | Correlation Ratio | Wasserstein (Ours) |
> > > |---------------------|----------|-------|-------|---------|-------------------|--------------------|
> > > | MLE$_\text{obj}$    | 2.45     | 1.52  | 2.26  | 2.47    | 2.61              | 2.18               |
> > > | SynMeter (Ours)     | 10.15    | 14.83 | 11.46 | 12.47   | 13.83             | 13.62              |
> > >
> > >
> > > | Utlity Improv (%) | Query Errors | MLE   | MLA (Ours) |
> > > |-------------------|--------------|-------|------------|
> > > | MLE$_\text{obj}$  | 2.63         | 10.58 | 7.34       |
> > > | SynMeter (Ours)   | 11.95        | 13.06 | 13.67      |
> > >
> > > The results indicate that our proposed tuning objective significantly enhances performance on both the proposed and existing metrics. Additionally, while MLE$_{\text{obj}}$ effectively improves machine learning efficiency (which is also their optimization objective), it shows limited improvement in other aspects, such as all the fidelity metrics and query errors.
> > >
> > > We believe the above results can better demonstrate the effectiveness of the proposed tuning objective. These experiments and results have been included in Appendix C.5 of the revised paper.
> > >
> > > We will also try our best to find more existing tuning objectives and evaluate the performance of more synthesizers to further showcase the robustness of SynMeter in the final version of the paper.
> > >
> > > Thank you again for your insightful suggestions, which have helped us better illustrate the advantages of the proposed tuning phase of SynMeter.
> > >
> > > [1] Xu, Lei, Maria Skoularidou, Alfredo Cuesta-Infante, and Kalyan Veeramachaneni. ``Modeling tabular data using conditional gan.'' Advances in neural information processing systems 32 (2019).
> > >
> > > [2] Vadim Borisov, Kathrin Sessler, Tobias Leemann, Martin Pawelczyk, and Gjergji Kasneci. ``Language models are realistic tabular data generators''. In International Conference on Learning Representations, 2023.
> > >
> > > [3] McKenna, Ryan, Daniel Sheldon, and Gerome Miklau. ``Graphical-model based estimation and inference for differential privacy.'' In International Conference on Machine Learning, pp. 4435-4444. PMLR, 2019.
> > >
> > > [4] Zhang, Zhikun, Tianhao Wang, Ninghui Li, Jean Honorio, Michael Backes, Shibo He, Jiming Chen, and Yang Zhang. ``PrivSyn: Differentially private data synthesis.'' In 30th USENIX Security Symposium (USENIX Security 21), pp. 929-946. 2021.
> > >
> > > [5] Hengrui Zhang, Jiani Zhang, Balasubramaniam Srinivasan, Zhengyuan Shen, Xiao Qin, Christos Faloutsos, Huzefa Rangwala, and George Karypis. ``Mixed-type tabular data synthesis with scorebased diffusion in latent space''. In International Conference on Learning Representations, 2024.
> > >
> > > [6] https://github.com/sdv-dev/CTGAN/issues/325
> > >
> > > [7] Kotelnikov, Akim, Dmitry Baranchuk, Ivan Rubachev, and Artem Babenko. ``Tabddpm: Modelling tabular data with diffusion models.'' In International Conference on Machine Learning, pp. 17564-17579. PMLR, 2023.
> > >
> > > [8] Zhao, Zilong, Aditya Kunar, Robert Birke, and Lydia Y. Chen. ``Ctab-gan: Effective table data synthesizing.'' In Asian Conference on Machine Learning, pp. 97-112. PMLR, 2021.

---

### Official Review · Reviewer_yZJK · 2024-11-05

**Soundness:** 3
**Presentation:** 2
**Contribution:** 3
**Rating:** 6
**Confidence:** 4

**Summary:**

The authors present an evaluation framework called SynMeter to evaluate generative modeling approaches for tabular data across 3 different dimensions: i) fidelity, ii) utility, and iii) privacy. The authors introduce reasonable metrics to evaluate algorithms/datasets along these 3 dimensions. The authors then evaluate several SOTA tabular data generation algorithms using SynMeter.

**Strengths:**

1) The paper is very thorough. Given the extensive appendix, I imagine it has gone through one or more review cycles before. Nevertheless, the authors clearly have discussed in details a lot of very reasonable concerns about their approach, which I quite appreciate.
2) The paper is quite topical and there aren't that many similar papers out there.

**Weaknesses:**

1) Presentation:
a) The authors try to cram in too many things into the paper. All figures are too tiny. I recommend moving some of the figures to the appendix if make the remaining ones bigger.
b) The appendix needs better structure. I'd recommend moving the experiment details ahead of discussion of limitations.
2) Technical points:
a) The MDS metric is designed for the synthesis algorithm whereas others are for a specific synthetic dataset. This is a major inconsistency.
b) MDS only captures privacy against MIA attacks. This is not  unreasonable but please spend a few lines explaining why you chose to only focus on MIAs?
c) One of the references [1] is quite similar to this paper in scope but uses slightly different metrics. Given the similarity, I would love to see the authors discuss the key differences between the papers and areas of novelty.



[1] Qian, Zhaozhi, Rob Davis, and Mihaela van der Schaar. "Synthcity: a benchmark framework for diverse use cases of tabular synthetic data." Advances in Neural Information Processing Systems 36 (2024).

**Questions:**

See weaknesses.

---

> ### Author Response · Authors · 2024-11-20
>
> We thank the reviewer for their insightful feedback and address the raised questions below.
>
> **Q4.1: The presentation of the work should be more well-organized.**
>
> We have reorganized all the tables in the paper, splitting the detailed results (fidelity, privacy, and utility) into two tables. Each table now includes results for six datasets (instead of all 12 datasets) and has been moved to the appendix.
> Additionally, we have enlarged all figures in the paper to improve readability.
>
> To further enhance clarity, we have added a new section (Section 6 in the revised paper) that provides detailed discussions of related work, including the paper you mentioned [1].
> Moreover, we have reorganized the Appendix, placing the experimental details and additional results before the discussion of existing metrics.
> We hope these changes make the presentation of our work more structured and easier to follow.
>
>
> [1] Qian, Zhaozhi, Rob Davis, and Mihaela van der Schaar.
> ``Synthcity: a benchmark framework for diverse use cases of tabular synthetic data.'' Advances in Neural Information Processing Systems 36 (2024).
>
>
> **Q4.2: The MDS metric is designed for the synthesis algorithm whereas others are for a specific synthetic dataset.**
>
> We agree with the reviewer that the privacy metric is designed for the synthesis algorithm, whereas fidelity and utility are defined for the synthetic data itself.
> We believe this distinction is reasonable, as well-established privacy evaluation metrics such as DP and MIA are designed for the data process rather than specific outputs.
> On the other hand, fidelity and utility metrics often refer to the quality and usability of the synthetic data, aligning with evaluation practices for other data types, such as image synthesis.
>
> **Q4.3: Explaining why MDS is designed to only focus only focus on MIAs.**
>
> MDS focuses specifically on membership disclosure risks due to its simplicity and its widespread use in the privacy evaluation of machine learning models [2].
> We recognize that there are other privacy attacks for tabular data, such as attribute inference attacks [3] and reconstruction attacks [4].
> We acknowledge that MDS does not encompass all potential privacy risks associated with synthetic datasets.
> We have clarified this more clearly in the revised paper (Section 3.2 and Appendix G).
>
> \
> [2] Carlini, Nicholas, Steve Chien, Milad Nasr, Shuang Song, Andreas Terzis, and Florian Tramer. ``Membership inference attacks from first principles.'' In 2022 IEEE Symposium on Security and Privacy (SP), pp. 1897-1914. IEEE, 2022.
>
>
> [3] Jayaraman, Bargav, and David Evans. ``Are attribute inference attacks just imputation?.'' In Proceedings of the 2022 ACM SIGSAC Conference on Computer and Communications Security, pp. 1569-1582. 2022.
>
> [4] Annamalai, Meenatchi Sundaram Muthu Selva, Andrea Gadotti, and Luc Rocher. ``A linear reconstruction approach for attribute inference attacks against synthetic data.'' (2024).
>
>
> **Q4.4: Clarify the difference between this paper and Qian et al. [1].**
>
> The key differences between our work and Qian et al. [1] (as well as other similar benchmark studies) are listed as follows:
> - Qian et al. [1] directly apply a wide range of existing evaluation metrics without analyzing their limitations or providing guidance on how these metrics should be used. In contrast, our work critically examines and identifies shortcomings in these metrics, proposing a new set of evaluation metrics specifically designed to address these limitations.
> - Qian et al. [1] focus on the development of an out-of-the-box Python library and do not provide any evaluation results. By comparison, we present a systematic evaluation process that includes tuning, training, and assessment, alongside insights derived from comprehensive experiments on various types of tabular synthesizers.
> \end{itemize}
>
>
> Given the rapid development of tabular data synthesis in recent years and the availability of numerous evaluation metrics, we believe our work contributes to the community by taking a step toward a standardized evaluation process.
> This, in turn, helps researchers better understand the strengths and weaknesses of various synthesis algorithms and advances the field of data synthesis evaluation.
>
>
> We also kindly refer the reviewer to our response to CQ1 in the common section and the newly added Section 6 in the revised paper for a detailed comparison with existing literature.

---

> > ### Comment · Reviewer_yZJK · 2024-11-22
> >
> > I am happy with the authors response to all my comments. I believe my concerns are addressed.

---

### Official Review · Reviewer_65dZ · 2024-11-06

**Soundness:** 2
**Presentation:** 4
**Contribution:** 2
**Rating:** 6
**Confidence:** 4

**Summary:**

This paper offers an opinionated selection of methods to evaluate synthetic data generation methods for tabular data. In particular, they select metrics to evaluate fidelity (Wasserstein distance), empirical privacy (they introduce the Membership Disclosure Score), and utility (via Machine Learning Affinity and QueryError). The paper goes through a very extensive set of experiments, where it compares synthesizers based on these metrics.

**Strengths:**

- Well-written, and evaluates many methods against many datasets
- Wasserstein distance seems like an appropriate choice for measuring fidelity, and it is adequately justified; similarly, MLA and QueryError seem good measures for utility.
- The paper offers some interesting takeaways: statistical methods work best for privacy applications, while diffusion models offer good fidelity.

**Weaknesses:**

- The proposed privacy metric, MDS, has critical flaws that make it unsuitable (and, potentially, misleading) for evaluating privacy. [See below]
- The use of Wasserstein distance for synthesizer isn't exactly new; for example, Singh et al. [1] used it in their optimization objective. Similarly, as argued below, MLA is incremental.
- Besides evaluating synthesizers with an opinionated (and well-justified, in some cases) approach, this paper feels quite redundant and it's unclear what is the "delta" from prior work. From a quick search, there's dozens of synthetic data evaluation frameworks [2-6], and it's unclear why a new one is needed. I appreciated your comparisons between metrics, provided in the appendix, but the main question is: can you empirically demonstrate that one would be wrong in using one of the prior frameworks, and that they should use yours instead?

[1] Singh Walia, Manhar. "Synthetic Data Generation Using Wasserstein Conditional Gans With Gradient Penalty (WCGANS-GP)." (2020).

[2] https://github.com/schneiderkamplab/syntheval

[3] https://github.com/Vicomtech/STDG-evaluation-metrics?tab=readme-ov-file

[4] Qian, Zhaozhi, Rob Davis, and Mihaela van der Schaar. "Synthcity: a benchmark framework for diverse use cases of tabular synthetic data." _Advances in Neural Information Processing Systems_ 36 (2024).

[5] Livieris, Ioannis E., et al. "An evaluation framework for synthetic data generation models." _IFIP International Conference on Artificial Intelligence Applications and Innovations_. Cham: Springer Nature Switzerland, 2024.

[6] McLachlan, Scott, et al. "Realistic synthetic data generation: The ATEN framework." _Biomedical Engineering Systems and Technologies: 11th International Joint Conference, BIOSTEC 2018, Funchal, Madeira, Portugal, January 19–21, 2018, Revised Selected Papers 11_. Springer International Publishing, 2019.

**Questions:**

**MDS**

1) Def 2 isn't well defined:
- H is supposedly sampled "at random" from the dataset, but the distribution of this sampling isn't defined.
- please replace the expectation's subscript with $H \subset D \setminus \{x\}$, or clarify that the expression $x\in H$ is an additional requirement to $H \subset D$.
- what is $\mathcal{M}$ in this definition? Can it be _any_ distance? Any particular property it should have?

2) I have several reasons to think that the membership disclosure score (MDS), as defined in Eq. 8, is a very poor choice:
- It's important to note that MDS, as defined in Eq. 8, is just an estimate, and it gives no formal guarantees. For a privacy metric, this is troublesome, and I highlight one counterexample to its reliability below.
- Here's an example where MDS suggests high privacy, but where an attack is trivial. Consider a synthesizer $s(\cdot)$ that maps a point as follows $s: x \mapsto -x$ . It's trivial to see how an attacker can achieve 100% accuracy. Yet, suppose the nearest neighbor to $x$ in the real dataset is $x+\varepsilon$. Then MDS would be proportional to $|d(x, s(x)) - d(x, s(x+\varepsilon))|$, which can be made arbitrarily small with $\varepsilon$. Hence MDS is a metric which can be tricked, and this makes it unsuitable for any serious privacy application. NOTE: I noted that in Appendix C you acknowledge possible drawbacks, but seem to dismiss them. Unfortunately, these are _critical_ issues even for less contrived synthesizers: for example, privacy metrics are routinely used to ensure that synthesizer implementations are bug-free, and this is certainly something that MDS cannot be trusted to do.
- MDS' value is (potentially) unbounded, and it's unclear how its value can be matched to the risk of successful attack. Note that the two main ways of measuring privacy in this context both offer this: 1) DP (its parameters can be mathematically matched to the risk of MIA) and of course 2) running a (potentially worst-case) MIA attack directly tells us this.
- Finally, an important drawback is that MDS doesn't really capture the worst-case: it takes the average (expectation) across multiple runs of the generator. This may be fine, but it should be carefully motivated.

I strongly recommend using a conventional metric (e.g., risk against state of the art MIA attack), which is empirical (similarly to MDS), but provides a better interpretation and it is well-understood by the security community.
Together with this MIA metric, I recommend also including a metric with theoretical guarantees; DP parameters $(\varepsilon, \delta)$ would be the most standard choice for this.

**Utility**

The authors introduce "Machine Learning Affinity" (MLA) as a metric, which is defined as the average difference across various models of the performance of a model that uses training or synthetic data. This feels incremental: most works on synthetic data generation already look at the difference between the performance (e.g., (Jordon et al., 2021)), and looking at the average across models looks like the natural next-step. I would recommend downtuning the claim that this metric is novel.

---

> ### Author Response · Authors · 2024-11-20
>
> We thank the reviewer for their insightful feedback and address the raised questions below.
>
> ***Questions about the fidelity metric***
>
>
> **Q3.1: The use of Wasserstein distance for synthesizers isn’t exactly new since some synthesizers used it in the optimization objective.**
>
> We agree with the reviewer that some synthesizers utilize the Wasserstein distance for optimization, and we have added Sing et al. [1] as a reference in our revised paper.
> In fact, as discussed in Section 5.4, we think that one possible reason that diffusion models outperform other deep generative models is their ability to effectively minimize the Wasserstein distance[2].
> However, to the best of our knowledge, leveraging the Wasserstein distance as a unified fidelity evaluation for tabular data synthesis has been less explored.
> Existing methods use various statistics (e.g., correlations as used in [1]) tailored to different types of marginals (categorical, continuous, and mixed).
> In contrast, the Wasserstein distance offers a more general and reliable measure for fidelity evaluation.
> We have included a more detailed discussion of existing fidelity metrics in Appendix F.1.
>
> [1] Singh Walia, Manhar. ``Synthetic Data Generation Using Wasserstein Conditional Gans With Gradient Penalty (WCGANS-GP).'' (2020).
>
> [2] Dohyun Kwon, Ying Fan, and Kangwook Lee. ``Score-based generative modeling secretly minimizes the wasserstein distance''. In Proceedings of the 36th International Conference on Neural Information Processing Systems, 2022.
>
> ***Questions about the privacy metric***
>
> **Q3.2: Question about Definition 2.**
>
> Thank you for pointing it out. We have explicitly mentioned that $H$ is sampled i.i.d from the dataset and replaced the expectation's subscript with $H \subset D \backslash x$ in Equation 6-8 for clarity.
> We also have mentioned in definition 2 that $\mathcal{M}$ is a distribution distance measurement, which should be non-negativity and symmetric.

---

> > ### Author Response · Authors · 2024-11-20
> >
> > **Q3.3: Should use a conventional metric like MIA and include a metric with theoretical guarantees.**
> >
> > We sincerely thank the reviewer for their valuable suggestions.
> > We would like to highlight that the ``conventional metric'' used to evaluate the privacy risks of tabular data synthesis is Distance to Closest Records (DCR) [1].
> > This metric is widely adopted by almost all SOTA heuristic privacy (HP) synthesizers [1, 4-6] and is advocated by several evaluation frameworks, including those you mentioned [7-8].
> > We argued in the paper that DCR is problematic as a syntactic privacy metric and should not be used for privacy evaluation.
> > Addressing this limitation is a key motivation behind our development of MDS.
> >
> > We agree with the reviewer that MIA is a conventional way to assess the privacy risks of machine learning models.
> > However, the effectiveness of MIA depends on the performance of specific MIA algorithms.
> > Unfortunately, as shown in Section 5.2, current SOTA MIAs achieve near-random guessing performance (less than 2\%TPR@1\%FPR) for DP synthesizers across a wide range of privacy budgets, from $\epsilon=1$ to infinity (non-private one).
> > Similar results have also been reported in related studies such as [11], which demonstrate that the AUC and ACC scores of all MIAs are below 0.6 and are nearly identical across different synthesizers.
> > Clearly, this cannot reflect the actual differences in privacy risks across varying privacy budgets and synthesizers.
> > Moreover, existing MIAs exhibit significant variance, making them unreliable for privacy evaluation.
> >
> > We would like to address MDS as an analytical framework that directly quantifies the disclosure risks of training data.
> > This approach is more aligned with recent works on analyzing data memorization [9] and information leakage [10], offering a different lens for understanding the privacy risks of data synthesis.
> >
> > However, we acknowledge that the proposed MDS is not without limitations (as discussed in Q3.4) and is not intended to replace MIAs.
> > Nevertheless, in cases where syntactic privacy metrics like DCR are used as heuristic privacy measures, we believe that it is better to use MDS instead.
> > We will clarify this point more explicitly in the revised paper.
> >
> > Additionally, we indeed have included provable privacy metrics (i.e., ($\epsilon,\delta$)-DP as you mentioned) in our evaluation with DP synthesizers.
> > However, DP is tailored for data synthesis algorithms and cannot be used as an empirical evaluation metric for HP synthesizers.
> >
> > [3] Zhao, Zilong, Aditya Kunar, Robert Birke, and Lydia Y. Chen. ``Ctab-gan: Effective table data synthesizing.'' In Asian Conference on Machine Learning, pp. 97-112. PMLR, 2021.
> >
> > [4] Hengrui Zhang, Jiani Zhang, Balasubramaniam Srinivasan, Zhengyuan Shen, Xiao Qin, Christos
> > Faloutsos, Huzefa Rangwala, and George Karypis. ``Mixed-type tabular data synthesis with scorebased diffusion in latent space''. In International Conference on Learning Representations, 2024.
> >
> > [5] Vadim Borisov, Kathrin Sessler, Tobias Leemann, Martin Pawelczyk, and Gjergji Kasneci. ``Language models are realistic tabular data generators''. In International Conference on Learning Representations, 2023.
> >
> > [6] Kotelnikov, Akim, Dmitry Baranchuk, Ivan Rubachev, and Artem Babenko. ``Tabddpm: Modelling tabular data with diffusion models.'' In International Conference on Machine Learning, pp. 17564-17579. PMLR, 2023.
> >
> > [7] https://github.com/schneiderkamplab/syntheval
> >
> > [8] https://github.com/Vicomtech/STDG-evaluation-metrics?tab=readme-ov-file
> >
> >
> > [9] Zhang, Chiyuan, Daphne Ippolito, Katherine Lee, Matthew Jagielski, Florian Tramèr, and Nicholas Carlini. ``Counterfactual memorization in neural language models.'' Advances in Neural Information Processing Systems 36 (2023): 39321-39362.
> >
> > [10] Ye, Jiayuan, Anastasia Borovykh, Soufiane Hayou, and Reza Shokri. ``Leave-one-out Distinguishability in Machine Learning.'' In The Twelfth International Conference on Learning Representations.
> >
> > [11] Houssiau, Florimond, James Jordon, Samuel N. Cohen, Owen Daniel, Andrew Elliott, James Geddes, Callum Mole, Camila Rangel-Smith, and Lukasz Szpruch. ``TAPAS: a Toolbox for Adversarial Privacy Auditing of Synthetic Data.'' In NeurIPS 2022 Workshop on Synthetic Data for Empowering ML Research.
> >
> >
> > **Q3.4: Counterexamples of MDS.**
> >
> > We agree with the reviewer that there are pathological synthesizers for which MDS may not be appropriate.
> > In Appendix G, we have included an example that closely resembles the counterexamples you have described. However, we would like to emphasize that most (if not all) practical synthesizers are not pathological. These synthesizers are randomization algorithms that learn the (noise) distribution of the input records and synthesize data from this distribution.
> > Our experiments on both HP and DP synthesizers demonstrate the effectiveness of MDS in evaluating these practical cases.
> > We will include further discussion of MDS's limitations in the revised paper.

---

> > > ### Comment · Reviewer_65dZ · 2024-11-22
> > >
> > > Thank you for your response.
> > >
> > > > We would like to highlight that the ``conventional metric'' used to evaluate the privacy risks of tabular data synthesis is Distance to Closest Records (DCR) [1].
> > >
> > > Apologies if my wording raised any confusion on the meaning of "conventional metric". My recommendation is to i) use DP wherever possible (thank you for the addition), and ii) TPR@FPR or accuracy of MIAs.
> > >
> > > > current SOTA MIAs achieve near-random guessing performance (less than 2%TPR@1%FPR)
> > >
> > > In this case, the algorithm should be evaluated with worst-case data and assumptions (similar to what DP auditing methods do).
> > >
> > > > However, we acknowledge that the proposed MDS is not without limitations (as discussed in Q3.4) and is not intended to replace MIAs. Nevertheless, in cases where syntactic privacy metrics like DCR are used as heuristic privacy measures, we believe that it is better to use MDS instead.
> > >
> > > MDS, like DCR, is unrelated to the risk of an attack (e.g., MIA); further, there's a counterexample (as per my review) where MDS is misleading; I would encourage you to include this as part of your discussion on MDS.
> > > As such, I fear MDS may lead to wrong comparisons between synthetic data generation methods.
> > >
> > > > Additionally, we indeed have included provable privacy metrics (i.e., ()-DP as you mentioned) in our evaluation with DP synthesizers. However, DP is tailored for data synthesis algorithms and cannot be used as an empirical evaluation metric for HP synthesizers.
> > >
> > > It can be argued for privacy that, similarly to other security contexts, if a defense is not mathematically proven then it should not be considered to be private, regardless of what empirical analyses say. This is particularly true if the empirical analyses are carried out based on metrics that are unrelated to the risk of an attack.
> > > It is your responsibility to highlight the risks in proposing this approach as part of an evaluation framework.

---

> > ### Comment · Reviewer_65dZ · 2024-11-22
> >
> > Thank you for the added reference, as well as for clarifying Definition 2.

---

> ### Author Response · Authors · 2024-11-20
>
> **Q3.5: MDS does not have a formal guarantee, is (potentially) unbounded, and does not capture the worst-case.**
>
> We would like to clarify that MDS is a *privacy evaluation metric*, not a privacy metric.
> This distinction means that MDS is designed to estimate the empirical privacy risks associated with a synthesizer, rather than providing formal privacy guarantees like DP.
>
> Additionally, rather than relying on the theoretical lower bound (the upper bound of MDS is 0 by design), we think it is more beneficial to use the MDS of a naive baseline (directly using real data as the synthetic one) to compare the privacy risks in practice.
> We have added the baseline in Table 7-8 to indicate the privacy risks of different synthesizers.
>
> We agree with the reviewer that MDS may not capture the worst-case scenario due to its use of expectations. However, compared to commonly used metrics like DCR—which relies on the mean of the 5th percentile of the distance distribution as the privacy score—MDS provides a more reliable assessment of privacy risks.
> We have rephrased the related sentence in the revised paper to avoid any potential misunderstanding.
>
> ***Question about the utility metric***
>
> **Q3.6: MLA is defined as the average difference across various models, which is incremental.**
>
> We acknowledge that [12] mentions measuring the variability of various models' performances through ranking, which is indeed conceptually similar to MLA.
> However, some SOTA synthesizers like [6] still misused this approach, achieving better machine learning performance on synthetic data than on real data.
> This misuse underscores the importance of employing a more robust metric like MLA for machine learning evaluation.
> Nevertheless, we will tone down our claim and include a reference to [12], as you have suggested.
>
> [12] Jordon, James, Lukasz Szpruch, Florimond Houssiau, Mirko Bottarelli, Giovanni Cherubin, Carsten Maple, Samuel N. Cohen, and Adrian Weller. ``Synthetic Data--what, why and how?.'' arXiv preprint arXiv:2205.03257 (2022).
>
> ***Question about the motivations***
>
> **Q3.7: There's dozens of synthetic data evaluation frameworks, why a new evaluation framework is needed?**
>
> We thank the reviewers for pointing out these related works.
> We have included all the papers you mentioned and added a new section in the revised paper (Section 6) to systematically discuss the differences between existing evaluation studies and our approach.
> We also discuss the key differences as follows.
>
> Specifically, [13-15] focus on developing toolboxes to facilitate the use of data synthesis by directly integrating a wide range of existing evaluation metrics. In contrast, our work critically analyzes and critiques these metrics, proposing a new set of evaluation metrics for systematic assessment. In addition, we also emphasize a systematic evaluation process to make sure of fair comparison between different types of synthesizers.
> [16] focuses solely on benchmarking HP synthesizers based on fidelity and utility but overlooks privacy evaluation, which we consider a critical aspect of HP synthesis evaluation.
> [17] primarily introduces methodologies for a specific type of tabular data (Electronic Health Records) and does not include the evaluation of general DP or HP synthesizers.
>
> We kindly refer the reviewer to our response to CQ1 in the common section for a detailed comparison with more related studies.
>
>
> [13] https://github.com/schneiderkamplab/syntheval
>
> [14] https://github.com/Vicomtech/STDG-evaluation-metrics?tab=readme-ov-file
>
> [15] Qian, Zhaozhi, Rob Davis, and Mihaela van der Schaar. ``Synthcity: a benchmark framework for diverse use cases of tabular synthetic data.'' Advances in Neural Information Processing Systems 36 (2024).
>
> [16] Livieris, Ioannis E., et al. ``An evaluation framework for synthetic data generation models.'' IFIP International Conference on Artificial Intelligence Applications and Innovations. Cham: Springer Nature Switzerland, 2024.
>
> [17] McLachlan, Scott, et al. ``Realistic synthetic data generation: The ATEN framework.'' Biomedical Engineering Systems and Technologies: 11th International Joint Conference, 2018.

---

> > ### Comment · Reviewer_65dZ · 2024-11-22
> >
> > > privacy evaluation metric, not a privacy metric
> >
> > Thank you. As per my previous comment, it would be useful for readers if you could more explicitly include the drawbacks of this choice.
> >
> > > Additionally, rather than relying on the theoretical lower bound (the upper bound of MDS is 0 by design), we think it is more beneficial to use the MDS of a naive baseline (directly using real data as the synthetic one) to compare the privacy risks in practice. We have added the baseline in Table 7-8 to indicate the privacy risks of different synthesizers.
> >
> > Makes sense.

---

> ### Author Response · Authors · 2024-11-20
>
> **Q3.8: Can you empirically demonstrate the superiority of proposed metrics?**
>
> Certainly! We have shown the superiority of the proposed evaluation metrics by conducting a series of experiments:
>
> - **Fidelity Metric.** We provide a simple example to demonstrate that correlation statistics fail to faithfully capture the fidelity of bivariate distributions due to their scale invariance. In contrast, the proposed Wasserstein distance effectively indicates distribution differences. Additionally, we note that the Wasserstein distance generalizes existing metrics such as Total Variation Distance and Contingency Similarity. For more details, we kindly refer the reviewer to our response to CQ1 in the common section.
> - **Privacy Metric.** We demonstrate the advantages of the proposed MDS in multiple ways. First, we address the limitations of commonly used syntactic privacy metrics, such as DCR, by identifying their fundamental flaw: their privacy notions are independent of the underlying synthesizer. This issue is thoroughly discussed in Section 3.2, and further supported by experiments in Appendix C.4. Additionally, in Section 5.2, we show the ineffectiveness of existing MIAs in detecting varying levels of privacy risks for both DP and HP synthesizers. In contrast, MDS consistently demonstrates reliability and robustness in detecting privacy risks among synthesizers, making it a more effective privacy evaluation metric.
> - **Utility Metrics.** We show in Appendix F.3 that MLA provides a more stable measure to evaluate the machine learning performance of synthetic data. (However, we will tone down this claim as you have suggested.)
>
> We hope these mentioned experiments and explanations provide a clearer understanding of the strengths of our proposed metrics.

---

> > ### Comment · Reviewer_65dZ · 2024-11-22
> >
> > > Fidelity Metric.
> >
> > Noted, thanks for the response.
> >
> > > Privacy Metric.
> >
> > See discussion above.
> >
> > > Utility Metrics.
> >
> > Noted, thanks for tuning down.

---

> > > ### Author Response · Authors · 2024-11-25
> > >
> > > Thank you for your valuable suggestions. We would like to clarify that MDS is closely related to the concept of MIA. At a high level, MDS can be viewed as a combination of DCR and MIA.
> > >
> > > MDS can be adapted into an MIA as follows. For a data point $x$, one first obtains two sets of distances to the closest synthetic data points, sets $A$ for the case when $x$ is included as input, and sets $B$ for the case when $x$ is not included, as what we did when computing MDS.  Given a synthetic dataset $S$, one computes the distance between $x$ the the closest data point in $S$ and then checks whether this distance is likely from the same distribution as $A$ or as $B$.
> > >
> > > Since DCR is widely used and intuitive, when we design MDS, we aim to come up with something conceptually similar (namely using distance to the closest data point), yet more aligned with the spirit of MIA so as to avoid the pitfalls of DCR.
> > >
> > > We have empirically evaluated several state-of-the-art MIAs (including two newly added [1-2] in our revised paper) against data synthesis and found that they are less effective at distinguishing between different levels of privacy risks.
> > > Empirical results on both HP and DP synthesizers demonstrate that MDS outperforms both DCR and existing MIAs in effectively capturing privacy risks.
> > > We thus believe that MDS represents an advancement in the state of the art for privacy evaluation metrics in tabular data synthesis.
> > >
> > > We agree with the reviewer that MDS has its limitations and may not be suitable for all synthesizers, particularly the counterexamples you mentioned. We have made the following modifications to address the limitation of MDS more explicitly:
> > >
> > >
> > > * We have added a new paragraph discussing the limitations of MDS in Section 3.2: (i) it may be misled by carefully designed pathological synthesizers, and (ii) it cannot capture all types of privacy risks posed by synthesizers. We also have detailed these limitations and incorporated the counterexample you mentioned in Appendix H.
> > > * We explicitly have mentioned in the conclusion (Section 7) that all existing empirical privacy evaluation metrics have limitations (including proposed MDS) and advocate for using DP synthesizers for applications where privacy is critical.
> > > * We have replaced `evaluation metrics` with `privacy evaluation metrics` to describe MDS in the revised paper to avoid any misleading.
> > > * We acknowledge MIA is a conventional privacy evaluation measure and should be more explored in the context of tabular data synthesis. Therefore, we have included a new MIA algorithm, TAPAS [1] in Section 5.2. Unfortunately, while TAPAS outperforms some previous MIA algorithms, it still fails to differentiate the varying privacy levels of DP synthesizers.
> > > * We also have added a new privacy evaluation approach proposed by Meeus et al.[2], which first identifies vulnerable samples in datasets and then performs MIA on these samples. Although this approach is more effective than conventional MIAs on some synthesizers, it suffers from a large standard deviation and is not effective for all types of synthesizers. Detailed experiments and discussion can be found in Appendix C.4 in the revised paper.
> > >
> > >
> > > We are deeply grateful to the reviewers for their valuable suggestions, which have significantly helped us improve the clarity and quality of the paper.
> > > We hope our explanations have clarified the motivations and limitations of the proposed metrics, and we are committed to further refining the paper in the final version.
> > >
> > > [1] Houssiau, Florimond, James Jordon, Samuel N. Cohen, Owen Daniel, Andrew Elliott, James Geddes, Callum Mole, Camila Rangel-Smith, and Lukasz Szpruch. ``TAPAS: a Toolbox for Adversarial Privacy Auditing of Synthetic Data.'' In NeurIPS 2022 Workshop on Synthetic Data for Empowering ML Research.
> > >
> > > [2] Meeus, M., Guepin, F., Creţu, A. M., de Montjoye, Y. A. (2023, September). Achilles’ heels: vulnerable record identification in synthetic data publishing. In European Symposium on Research in Computer Security (pp. 380-399). Cham: Springer Nature Switzerland.

---

> > > > ### Author Response · Authors · 2024-11-26
> > > >
> > > > As the interactive rebuttal window will close soon, we thank all reviewers again for all their helpful feedback. We believe we have addressed the reviewer's questions in our answer and are eager to continue the conversation in case of further questions. We kindly request the reviewers to consider adjusting their score to reflect the improvements and clarifications made in response to their input.

---

> > > > > ### Comment · Reviewer_65dZ · 2024-11-27
> > > > >
> > > > > I have upgraded my score.
> > > > >
> > > > > > We have replaced evaluation metrics with privacy evaluation metrics to describe MDS in the revised paper to avoid any misleading.
> > > > >
> > > > > Please include an explanation in the paper of the distinction between the two (in a similar way you did here).

---

> > > > > > ### Author Response · Authors · 2024-11-28
> > > > > >
> > > > > > Thank you for the reminder! As the paper revision deadline has passed, we will include the explanation in the main text in the final version of the paper. Thank you again for your insightful comments!

---

### Official Review · Reviewer_NX8w · 2024-11-08

**Soundness:** 2
**Presentation:** 2
**Contribution:** 2
**Rating:** 3
**Confidence:** 3

**Summary:**

This paper investigates the performance of tabular data synthesizers w.r.t. three metrics, fidelity, privacy, and utility. The authors conduct extensive experiments to compare SOTA tabular data synthesizers.

**Strengths:**

S1. Extensive experiments are conducted to examine the performance of each tabular data synthesizer w.r.t. the three metrics adopted in this paper.

**Weaknesses:**

W1. The technical depth of this paper is limited. All the metrics are already proposed by existing work and the authors mainly conduct an empirical comparison among tabular data synthesizers. The novelty of this paper is not clear.

W2. This paper discusses tabular data synthesizers. However, it seems to me that the three metrics also fit general data synthesizers. What is the unique feature of tabular data that requires the adoption of the three metrics in model evaluation? If the authors cannot elaborate on the connection between the three metrics and tabular data, the motivation for adopting the three metrics will be unclear.

W3. Since the authors adopt three metrics, the tabular data synthesis task becomes a multi-objective optimization problem. What are the relationships among the three objectives? Do they contradict each other? Is it possible to maximize the model performance w.r.t. all three metrics? The authors should provide an in-depth analysis of these issues.

W4. For fidelity, why only consider the marginal distribution (definition 1)? If we only consider marginal distributions, the complex relationships among columns in a table may be ignored. Note that for tabular data, we may have some (approximate) functional dependencies among columns, which are very important for data integrity and challenging to capture for tabular data synthesizers.

**Questions:**

W1-W4

---

> ### Author Response · Authors · 2024-11-20
>
> We thank the reviewer for their insightful feedback and address the raised questions below.
>
>
> **Q2.1: The novelty of this paper is not clear.**
>
> We respectfully disagree with the assertion that all metrics are already proposed by existing work.  In particular, we proposed a new privacy metric MDS.  Below we would like to elaborate on the key contributions of our paper:
>
> - **New Systematic Evaluation Metrics.** In the paper we propose new evaluation metrics for fidelity, privacy, and utility.
>     For fidelity, we introduce the Wasserstein distance, which generalizes existing metrics such as total variation distance and contingency similarity [1] and accommodates both numerical and categorical attributes under a unified criterion.
>     For privacy, we propose Membership Disclosure Score (MDS), a novel privacy evaluation metric that addresses the limitations of the widely used yet problematic DCR metric [2].
>     For utility, we propose Machine Learning Affinity (MLA), which modifies the Machine Learning Efficacy (MLE) metric[3] by measuring the relative performance gap with a set of machine learning models.
>     We demonstrate through an example in Appendix F.3 in the revised paper that MLA provides a more stable measure for machine learning prediction.
> - **New Tuning Objective.** One important problem of existing synthesizers is the lack of a principled approach for hyperparameter tuning. For example, CTGAN [3] is notoriously difficult to tune, with its authors describing the tuning process as ``a bit of an art''[9]. In addition, many papers [3-7] omit tuning details. However, we found that hyperparameter tuning significantly impacts synthetic data quality across datasets and thus is indispensable for fair comparisons between different types of synthesizers. Our proposed tuning objective, based on the new evaluation metrics, provides a systematic approach to this challenge and can improve the quality of synthetic data for all methods.
> - **New Insights for Tabular Data Synthesis.** Our paper also offers new insights for tabular data synthesis.
>     For example, we show that statistical synthesizers (overlooked by many deep-generative models [4,7,8]) can outperform some complex models like CTGAN and offer strong privacy protection even without using differential privacy.
>     Moreover, we highlight that existing diffusion-based synthesizers exhibit significant privacy leakage risks, an issue that has been underestimated in prior studies due to the use of problematic privacy metrics such as DCR.
>     We believe these new insights can help the community better understand the progress and challenges in tabular data synthesis.
>
> Additionally, we publicly release SynMeter, a benchmark built on the proposed evaluation metrics, which can serve as a useful tool for the systematic evaluation of various types of HP and DP tabular data synthesizers.
>
>
> [1] https://docs.sdv.dev/sdmetrics/metrics/metrics-glossary/contingencysimilarity
>
> [2] Zhao, Zilong, Aditya Kunar, Robert Birke, and Lydia Y. Chen. ``Ctab-gan: Effective table data synthesizing.'' In Asian Conference on Machine Learning, pp. 97-112. PMLR, 2021.
>
> [3] Xu, Lei, Maria Skoularidou, Alfredo Cuesta-Infante, and Kalyan Veeramachaneni. ``Modeling tabular data using conditional gan.'' Advances in neural information processing systems 32 (2019).
>
> [4] Vadim Borisov, Kathrin Sessler, Tobias Leemann, Martin Pawelczyk, and Gjergji Kasneci. ``Language models are realistic tabular data generators''. In International Conference on Learning Representations, 2023.
>
> [5] McKenna, Ryan, Daniel Sheldon, and Gerome Miklau. ``Graphical-model based estimation and inference for differential privacy.'' In International Conference on Machine Learning, pp. 4435-4444. PMLR, 2019.
>
> [6] Zhang, Zhikun, Tianhao Wang, Ninghui Li, Jean Honorio, Michael Backes, Shibo He, Jiming Chen, and Yang Zhang. ``PrivSyn: Differentially private data synthesis.'' In 30th USENIX Security Symposium (USENIX Security 21), pp. 929-946. 2021.
>
>
> [7] Hengrui Zhang, Jiani Zhang, Balasubramaniam Srinivasan, Zhengyuan Shen, Xiao Qin, Christos
> Faloutsos, Huzefa Rangwala, and George Karypis. ``Mixed-type tabular data synthesis with scorebased diffusion in latent space''. In International Conference on Learning Representations, 2024.
>
> [8] Kotelnikov, Akim, Dmitry Baranchuk, Ivan Rubachev, and Artem Babenko. ``Tabddpm: Modelling tabular data with diffusion models.'' In International Conference on Machine Learning, pp. 17564-17579. PMLR, 2023.
>
> [9] https://github.com/sdv-dev/CTGAN/issues/325

---

> ### Author Response · Authors · 2024-11-20
>
> **Q2.2: Connections between the three metrics and tabular data synthesis.**
>
> Thanks for raising this question.  The metrics were designed for tabular data; however, it is plausible that some metrics/methodologies can be adapted to apply to other data synthesizers.   We view that as strength rather than weakness of our paper.   We now elaborate on how these metrics were designed for tabular data.
>
> - Our proposed Wasserstein distance-based fidelity metric measures distances between low-dimensional $k$-way marginals computed from the synthetic data and the real data.  The usage of marginals is because of the nature of tabular data.
> - Our proposed MDS privacy metric measures the maximum disclosure risks among all samples using distance; and it could be applied to other data types.
> - Our proposed utility metric includes two components: machine learning affinity (MLA) and query errors.
> MLA considers machine learning tasks like classification and regression, which are also natural for tabular data.
> Quer errors are calculated on a 3-way range/point query task, which is also specific to tabular data analysis.
>
>
>
> **Q2.3: In-depth analysis of three metrics for multi-objective optimization.**
>
> We have evaluated the model performance with various coefficient configurations in Appendix C.5.
> The results indicate that our optimization objective is not sensitive to specific coefficient selections, with all configurations showing robust performance improvements across all synthesizers.
> For example, even when the fidelity metric is not involved in optimization (i.e., $\alpha_1=0$), the fidelity of the synthesizers still improves.
>
> However, we also observed that no single coefficient configuration maximizes model performance across all three metrics.
> We believe this is because each metric emphasizes a different aspect of synthetic data quality.
> For instance, MLA is designed to maximize machine learning performance, specifically focusing on the correlation with label columns.
> In contrast, the fidelity metric evaluates the overall distributional similarity between real and synthetic data, which is independent of downstream tasks. We have added these discussions in Appendix C.5 in the revised paper.
>
> **Q2.4: Why only consider the marginal distribution for fidelity? How about the complex relationship among columns?**
>
> We would like to highlight that the proposed Wasserstein-based fidelity metric *indeed* captures correlations among columns.
> As detailed in Definition 1 and Equation 4, the evaluated marginal distributions can be $k$-way, enabling the measurement of similarities across any multivariate distributions.
> In our implementation, we compute the Wasserstein distance of all the one-way and two-way marginals (categorical, continuous, mixed) and use the mean as the final fidelity score, as detailed in Appendix B.1.
> In fact, as mentioned in Appendix F.1, the proposed fidelity metric addresses the scale invariance issue [10] of correlation statistics (e.g., Pearson correlation) used in previous works [2, 8], offering a more reliable and universal fidelity metric.
> Furthermore, we provide a simple example in CQ2 (in the common section above) to illustrate the superiority of the Wasserstein-based metric and its ability to capture correlations among columns effectively.
>
> [10] https://en.wikipedia.org/wiki/Pearson_correlation_coefficient#Mathematical_properties

---

> > ### Comment · Reviewer_NX8w · 2024-11-24
> >
> > Thanks for the clarification. I do not have further questions.
> >
> > Although I appreciate the authors' feedback on my concerns, I still feel the technical depth of this submission is too limited as it is purely empirical without any in-depth and rigorous analysis. Therefore, I will maintain my initial score.

---

> > > ### Author Response · Authors · 2024-11-25
> > >
> > > Thank you for your reply. We acknowledge that our work mainly focuses on empirical results (like all other evaluation studies). However, we believe our studies would be beneficial for the community for the following reasons:
> > >
> > > * Addressing Limitations of Existing Metrics. We have identified the shortcomings of widely used syntactic privacy evaluation metrics, such as DCR [2], and proposed a new, effective privacy evaluation metric for assessing the privacy risks of HP synthesizers. We think this is crucial, as reliance on flawed evaluation metrics can lead to incorrect conclusions and hinder progress in the field. We also discuss examples of such incorrect conclusions in Appendix I.
> > > * Unified Fidelity Evaluation and Systematic Frameworks. We have introduced a more flexible and comprehensive fidelity metric based on Wasserstein distance, emphasized the importance of hyperparameter tuning, and proposed a systematic evaluation framework for tabular data synthesis.
> > > * Comprehensive Evaluation and Insights. We conducted an extensive evaluation of a wide range of DP and HP synthesizers, using the proposed metrics to analyze their strengths and weaknesses in detail (as discussed in Section 5.4). This comparison is particularly important because DP synthesizers are often developed by the database and security communities, while HP synthesizers are typically introduced by the AI/ML community, and comparisons between the two have been limited. Our head-to-head comparisons are practically useful for practitioners seeking the best performance for their task and facing the daunting task of selecting and configuring appropriate synthesizers.
> > >
> > > We believe these contributions offer valuable insights and practical advancements for the field.
> > > We kindly request that you reconsider your score in light of the contributions we have outlined.
> > >
> > > Thank you again for your time and thoughtful comments.

---

### Official Review · Reviewer_mPz2 · 2024-11-09

**Soundness:** 3
**Presentation:** 3
**Contribution:** 3
**Rating:** 8
**Confidence:** 4

**Summary:**

This paper reviews the current state of tabular data synthesis, an approach that balances data utility with privacy. Despite numerous proposed algorithms, a comprehensive comparison of their performance is lacking due to the absence of standardized evaluation metrics. The authors critique existing metrics and propose new ones focusing on fidelity, privacy, and utility. They also introduce a unified tuning objective that enhances the quality of synthetic data across different methods. Extensive evaluations on eight synthesizers and twelve datasets reveal insights that guide future research on privacy-preserving data synthesis.

**Strengths:**

+. The authors introduced a new fidelity metric based on Wasserstein distance to evaluate diverse data types, addressing the heterogeneity and high dimensionality of tabular data.
+. The authors introduced the membership disclosure score as a novel privacy metric effectively addresses the limitations of existing privacy metrics and enhances the understanding of privacy risks in data synthesis.

**Weaknesses:**

-. The related work section of the paper is relatively weak and should be systematically organized to provide a more comprehensive introduction of relevant studies.
-. My major concern is whether the synthesized tabular data can maintain usability for more complex downstream applications. In other words, is this usability specific to certain downstream tasks, or can it apply to any downstream task?

**Questions:**

How can we evaluate whether the synthesized tabular data has general applicability for downstream tasks?

---

> ### Author Response · Authors · 2024-11-20
>
> We thank the reviewer for their insightful feedback and address the raised questions below.
>
> **Q1.1: The related work section of the paper is relatively weak and should be systematically organized.**
>
> We agree with the reviewer that the related work section should be more detailed and well-organized.
> We have added a new section (Section 6 in the revised paper) to ensure a more systematic presentation of existing studies.
> Specifically, we have detailed the related benchmark studies of DP and HP synthesizers and also discussed the new evaluation toolbox and metrics.
> Additionally, we have expanded the discussion to include more evaluation studies and highlighted the key difference between our approach and existing methods.
> We kindly refer the reviewer to our response to CQ1 in the common section for specific details on these improvements and differences.
>
> **Q1.2: How can we evaluate whether the synthesized tabular data has general applicability for downstream tasks?**
>
> Given that any data synthesis process loses some information, we believe that no metric can ensure that synthetic datasets with good scores apply to any downstream task. However, as both the utility metric and the fidelity metric measure in certain ways whether a synthetic dataset is useful for downstream tasks, we believe that the usability applies beyond the classification/regression tasks considered in the utility metric.
>
> In the utility score, we utilize two of the most commonly adopted downstream tasks (e.g., classification/regression and range/point query) to assess the utility of synthetic data.
> These tasks are widely recognized as benchmarks for evaluating synthetic data utility.
>
>
> The Wasserstein distance-based fidelity metric directly measures the distributional similarity between synthetic and real data. If the downstream tasks depend on such distributional similarity, then it is likely that a synthetic dataset with high fidelity is useful.
>
>
> For example, TabDDPM, which demonstrates the best fidelity in our paper, also achieves the highest utility across the two downstream tasks.
> Conversely, some synthesizers perform well on specific downstream tasks but struggle with fidelity.
> A case in point is TVAE on the Shoppers dataset: while it ranks third for machine learning prediction, it exhibits the worst fidelity among heuristic private synthesizers.
> Therefore, fidelity evaluation provides an effective approach to evaluating general applicability across varying use cases.

---

> > ### Comment · Reviewer_mPz2 · 2024-11-24
> >
> > The author's response has addressed some of my concerns, so I will maintain my current rating.

---

### Author Response · Authors · 2024-11-20
**Common Response**

We thank all reviewers for their constructive feedback, which has been instrumental in improving our work.
We have addressed the concerns raised by each reviewer individually, with detailed responses provided under their respective reviews.
Revisions made to the paper in response to the reviews are highlighted in blue on the paper for clarity.
For questions raised by multiple reviewers, we have included a dedicated common section below, and we refer to this section in the relevant individual responses.

**CQ1: The related work section of the paper is relatively weak and the difference between this work and existing literature should be addressed.**

We agree with the reviewers that the related work section should be more detailed and better organized.
In response, we have added a new section (Section 6 in the revised paper) to provide a more systematic presentation of existing studies.
Additionally, we have included all the literature mentioned by the reviewers and provided a detailed comparison with our work.

We highlight the key differences between our work and these studies as follows:
* **Fidelity Evaluation.**  To assess the different types of marginals in tabular data, existing evaluation studies typically incorporate a wide range of existing metrics [1-7] or add these metrics together to form a final fidelity score [8].
    In contrast, we propose a Wasserstein-based fidelity metric that accommodates both categorical and numerical attributes for any $k$-way marginals, offering a more unified and reliable approach for fidelity evaluation.
* **Privacy Evaluation.** Many studies rely on DCR or other syntactic metrics for privacy evaluation [3,4]. However, as shown in our paper, these metrics are flawed.
    We propose a new heuristic privacy metric that, similar to DCR, also uses closeness between real record and synthetic record for measure leakage, but uses it in a way that addresses the flaws of DCR.
    Additionally, some studies [6-8] overlook the privacy evaluation, which we believe is critical for evaluating HP synthesizers.
* **Motivations**. Existing studies often focus on benchmarking a wide range of synthesizers using existing metrics [1,2,4,6-8] or on developing user-friendly toolboxes [3,5]. In contrast, we emphasize a systematic evaluation process—encompassing tuning, training, and evaluation—using more principled metrics to provide deeper insights into the strengths and weaknesses of different synthesis algorithms.

We believe our work contributes to the community by taking a step toward a standardized evaluation process of tabular data synthesis. This, in turn, helps researchers better understand the current progress of tabular data synthesis.


[1] Tao, Yuchao, Ryan McKenna, Michael Hay, Ashwin Machanavajjhala, and Gerome Miklau. ``Benchmarking differentially private synthetic data generation algorithms.'' arXiv preprint arXiv:2112.09238 (2021).

[2] Yuzheng Hu, Fan Wu, Qinbin Li, Yunhui Long, Gonzalo Garrido, Chang Ge, Bolin Ding, David Forsyth, Bo Li, and Dawn Song. ``Sok: Privacy-preserving data synthesis.'' In 2024 IEEE Symposium on Security and Privacy (SP), pp. 2–2, 2024.

[3] https://github.com/Vicomtech/STDG-evaluation-metrics?tab=readme-ov-file

[4] Lautrup, Anton Danholt, Tobias Hyrup, Arthur Zimek, and Peter Schneider-Kamp. ``SynthEval: A Framework for Detailed Utility and Privacy Evaluation of Tabular Synthetic Data.'' arXiv preprint arXiv:2404.15821 (2024).

[5] Qian, Zhaozhi, Rob Davis, and Mihaela van der Schaar. ``Synthcity: a benchmark framework for diverse use cases of tabular synthetic data.'' Advances in Neural Information Processing Systems 36 (2024).

[6] Livieris, Ioannis E., et al. ``An evaluation framework for synthetic data generation models.'' IFIP International Conference on Artificial Intelligence Applications and Innovations. Cham: Springer Nature Switzerland, 2024.

[7] Espinosa, Erica, and Alvaro Figueira. ``On the quality of synthetic generated tabular data.'' Mathematics 11, no. 15 (2023): 3278.

[8] Lahoti, Mukund, and Pratik Narang. ``A Universal Metric for Robust Evaluation of Synthetic Tabular Data.'' (2024).

---

> ### Author Response · Authors · 2024-11-20
>
> **CQ2: Demonstration of why Wasserstein-based fidelity metric is better than other metrics.**
>
> Certainly! Here, we use a simple example to demonstrate the superiority of the Wasserstein-based metric over some commonly used metrics and also show how it generalizes existing metrics.
>
>
> *Wasserstein-based metric is more faithful than existing correlation-based metrics.*
>
> Correlation statistics are often used to measure the fidelity of bivariate marginal distributions in synthetic data.
> These metrics have been widely adopted by many SOTA synthesizers [9,10] and evaluation frameworks [1,4,7].
> The process involves computing correlation scores for both real and synthetic data and then calculating the difference between these scores, with smaller differences indicating higher fidelity.
>
> To accommodate different types of attributes (categorical, continuous, and mixed), various correlation statistics are applied: Theil’s uncertainty coefficient, Pearson correlation, and the correlation ratio.
> However, correlation statistics are scale-invariant [11], which limits their ability to faithfully capture distribution similarities between real and synthetic data. We use the following example to demonstrate this limitation.
>
> Considering a simple tabular dataset with two continuous columns $X$ and $Y$:
> - The first column $X$ is generated from a standard normal distribution: $X \sim \mathcal{N}(0,1)$.
> - The second column is generated by adding a constant $c$ from $X$: $Y=X+c$.
>
> Now assume we have a synthesizer that outputs a synthetic dataset where:
> - The first column $X^{\prime}$ comes from a different normal distribution $X^{\prime} \sim \mathcal{N}(-1,1)$.
> - The second column $Y^\prime$ is generated by adding a constant $d$ from $X^{\prime}$: $Y^{\prime}=X^{\prime}+d$.
>
> In this case, the two columns ($X$ and $Y$) in both the real and synthetic datasets are perfectly linearly correlated.
> Using Pearson correlation, both real and synthetic datasets would yield a correlation score of 1. Consequently, the computed score would be 0 (indicating high fidelity).
> However, this result is misleading because the bivariate distributions of the real and synthetic data are quite different: one is $\mathcal{N}(0,1)$ while the other is $\mathcal{N}(-1,1)$.
> Correlation-based metrics fail to capture this discrepancy because they are insensitive to shifts in the data's underlying distribution.
>
> In contrast, the proposed Wasserstein-based metric measures the minimum effort required to transform one distribution into another and produces a score of 1.35 when $c=d=0.5$ (a higher Wasserstein score indicates lower fidelity).
> This approach effectively captures the discrepancies between the above bivariate distributions, providing a more faithful and reliable measure of fidelity.
>
> We have appended the code of the above example in the Supplementary Material.
> This demonstrates how the Wasserstein-based metric can better reflect the differences between the distributions and is more suitable as a fidelity metric.
>
> *Wasserstein-based metric is the generalization of some existing metrics.*
>
> We also note that the proposed Wasserstein-based metric generalizes several existing metrics, such as Total Variation Distance[12] and Contingency Similarity[13] , which are commonly used to measure the fidelity of one-way or two-way categorical distributions.
>
> Specifically, by customizing the cost matrix in the Wasserstein distance to enforce strict matching for categorical attributes (assigning an infinite cost to mismatches between different categories and assigning cost as 1 for the same categories), the computation simplifies the sum of probability differences for each attribute.
> This approach aligns directly with how Total Variation Distance and Contingency Similarity are calculated.
>
> In summary, we believe the Wasserstein-based metric provides a more faithful and general fidelity measure for synthesis evaluation.
>
>
>
>
> [9] Hengrui Zhang, Jiani Zhang, Balasubramaniam Srinivasan, Zhengyuan Shen, Xiao Qin, Christos
> Faloutsos, Huzefa Rangwala, and George Karypis. ``Mixed-type tabular data synthesis with scorebased diffusion in latent space''. In International Conference on Learning Representations, 2024.
>
> [10] Kotelnikov, Akim, Dmitry Baranchuk, Ivan Rubachev, and Artem Babenko. ``Tabddpm: Modelling tabular data with diffusion models.'' In International Conference on Machine Learning, pp. 17564-17579. PMLR, 2023.
>
>
> [11] https://en.wikipedia.org/wiki/Pearson_correlation_coefficient#Mathematical_properties
>
> [12] https://docs.sdv.dev/sdmetrics/metrics/metrics-glossary/tvcomplement
>
> [13] https://docs.sdv.dev/sdmetrics/metrics/metrics-glossary/contingencysimilarity

---

### Meta-Review · Area_Chair_ZmNd · 2024-12-16

**Metareview:**

The tabular data synthesis is important in practical applications. Many methods have been proposed, but their evaluation protocols lack rigorousness. Therefore, this work proposes a set of new metrics  in terms of fidelity, privacy, and utility. They conducted experiments with recent methods on various datasets. However, there exist disputes on the efficacy of the proposed metric and the experimental environments. I think that they need to largely expand the scale of the experiment after including more synthesizer and tabular datasets. It is too small to be considered as a top-notch assessment paper.

**Additional Comments On Reviewer Discussion:**

The authors had left a plethora of justification in terms of more experiment and the meaning of the new metrics. However, the overall evaluation is little below the decision boundary.

---

### Decision · Program_Chairs · 2025-01-22

Reject